# GTSF1 accelerates target RNA cleavage by PIWI-clade Argonaute proteins

Amena Arif[1,2,6], Shannon Bailey[2], Natsuko Izumi[3], Todd A. Anzelon[4], Deniz M. Ozata[2,7], Cecilia Andersson[2], Ildar Gainetdinov[2], Ian J. MacRae[4], Yukihide Tomari[3,5] & Phillip D. Zamore[2✉]

Argonaute proteins use nucleic acid guides to find and bind specific DNA or RNA target sequences. Argonaute proteins have diverse biological functions and many retain their ancestral endoribonuclease activity, cleaving the phosphodiester bond between target nucleotides t10 and t11. In animals, the PIWI proteins—a specialized class of Argonaute proteins—use 21–35 nucleotide PIWI-interacting RNAs (piRNAs) to direct transposon silencing, protect the germline genome, and regulate gene expression during gametogenesis[1]. The piRNA pathway is required for fertility in one or both sexes of nearly all animals. Both piRNA production and function require RNA cleavage catalysed by PIWI proteins. Spermatogenesis in mice and other placental mammals requires three distinct, developmentally regulated PIWI proteins: MIWI (PIWIL1), MILI (PIWIL2) and MIWI2[2–4] (PIWIL4). The piRNA-guided endoribonuclease activities of MIWI and MILI are essential for the production of functional sperm[5,6]. piRNA-directed silencing in mice and insects also requires GTSF1, a PIWI-associated protein of unknown function[7–12]. Here we report that GTSF1 potentiates the weak, intrinsic, piRNA-directed RNA cleavage activities of PIWI proteins, transforming them into efficient endoribonucleases. GTSF1 is thus an example of an auxiliary protein that potentiates the catalytic activity of an Argonaute protein.

In animals, PIWI-interacting RNAs (piRNAs) 21 to 35 nucleotides in length direct PIWI proteins to silence transposons and regulate gene expression[1]. Invertebrates produce piRNAs and PIWI proteins in both the soma and the germline but mammalian piRNAs act only in the germline. Mice that lack any of their three PIWI proteins—MIWI2, MILI and MIWI—or other proteins required for piRNA production are invariably male-sterile[1–4]. As in other animals, mouse piRNA production requires catalytically active PIWI proteins, MILI and MIWI[5,6]. Transposon silencing is the ancestral function of piRNAs; uniquely, placental mammals also produce pachytene piRNAs, which first appear shortly after the onset of meiosis I[13–18] and reach a peak abundance in spermatocytes rivalling that of ribosomes[19]. Pachytene piRNAs tune the abundance of mRNAs required for spermiogenesis, the process by which round spermatids become sperm[20–23]. Pachytene piRNAs have been proposed to direct MIWI and MILI to cleave specific mRNAs, ensuring appropriate levels of their protein products[20,23–25]. Testing this idea has been thwarted by the absence of a cell-free system in which MIWI or MILI can be loaded with synthetic piRNAs of defined sequence.

## Recombinant MIWI loaded with a defined piRNA

We used lentiviral transduction to engineer a stable HEK293T cell line over-expressing epitope-tagged MIWI. Tagged MIWI was captured from cell lysate using anti-Flag antibody coupled to paramagnetic beads, incubated with a synthetic piRNA bearing a monophosphorylated 5′ terminus and 2′-O-methylated 3′ end, and eluted from the magnetic beads using 3×Flag peptide. MIWI loaded with a synthetic piRNA (MIWI piRISC) (Fig. 1a), but not unloaded apo MIWI (Extended Data Fig. 1a), cleaved a 5′ [32]P-radiolabelled target RNA fully complementary to the synthetic guide (Extended Data Fig. 1b,c and Supplementary Data Fig. 1).

Recombinant MIWI bound stably to an RNA guide bearing a 5′ monophosphate but not to a guide with a 5′ hydroxyl group (Extended Data Fig. 1d). The MID domain of Argonaute proteins contains a 5′ monophosphate-binding pocket that anchors the RNA guide to the protein[26–30]. Mutations predicted to disrupt 5′ monophosphate-binding perturb PIWI function[28,31–35]. We immobilized MIWI on paramagnetic beads via its 3×Flag tag, incubated it with guide RNA, washed the beads, eluted the MIWI piRISC with 3×Flag tag peptide, and tested its ability to cleave a fully complementary target RNA. Incubation with a 5′ monophosphorylated but not an otherwise identical 5′ hydroxy guide yielded MIWI piRISC that cleaved target RNA (Extended Data Fig. 1d). In vivo, the methyltransferase HENMT1 adds a 2′-O-methyl group to the 3′ ends of piRNAs. Terminal 2′-O-methyl modification likely stabilizes small RNAs against degradation by cellular ribonucleases rather than secures the guide to the protein[1]. Consistent with this function for 2′-O-methylation, piRNAs bearing a 3′ terminal 2′ hydroxyl

[1]Department of Biochemistry and Molecular Biotechnology Graduate Program, University of Massachusetts Chan Medical School, Worcester, MA, USA. [2]Howard Hughes Medical Institute and RNA Therapeutics Institute, University of Massachusetts Chan Medical School, Worcester, MA, USA. [3]Laboratory of RNA Function, Institute for Quantitative Biosciences, The University of Tokyo, Tokyo, Japan. [4]Department of Integrative Structural and Computational Biology, The Scripps Research Institute, La Jolla, CA, USA. [5]Department of Computational Biology and Medical Sciences, Graduate School of Frontier Sciences, The University of Tokyo, Kashiwa, Japan. [6]Present address: Beam Therapeutics, Cambridge, MA, USA. [7]Present address: Department of Molecular Biosciences, Stockholm University, Stockholm, Sweden. ✉e-mail: phillip.zamore@umassmed.edu

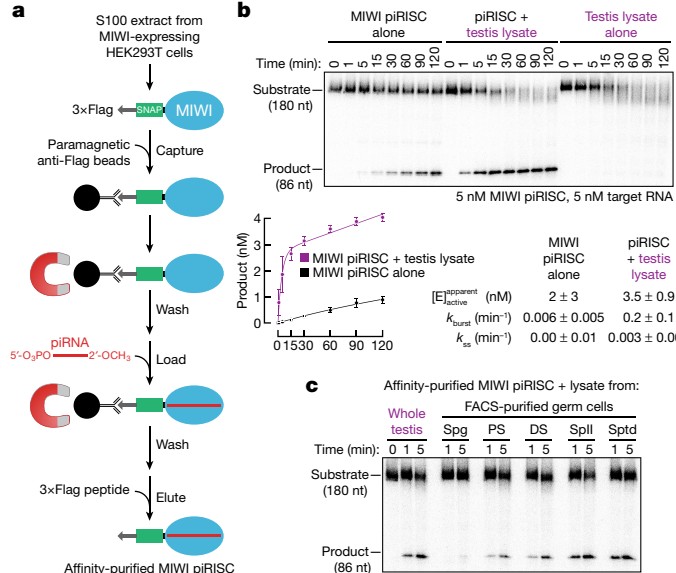

**Fig. 1 | A component of mouse testis lysate potentiates piRNA-directed target RNA cleavage by MIWI. a**, Strategy for programming recombinant MIWI with synthetic piRNA. **b**, Top, representative denaturing polyacrylamide gel electrophoresis showing target RNA cleavage by MIWI piRISC with and without added testis lysate. Bottom, product formed as a function of time by MIWI piRISC (mean ± s.d., $n$ = 3). Rate constants were determined by fitting the data to the burst-and-steady-state equation (Methods, equation (1)). $[E]_{active}^{apparent}$, the apparent concentration of active MIWI piRISC estimated from data fitting. **c**, Target RNA cleavage by MIWI piRISC in the presence of lysate from either whole testis or FACS-purified germ cells. DS, diplotene spermatocytes; PS, pachytene spermatocytes; SpII, secondary spermatocytes; Spg, spermatogonia; Sptd, spermatids. For gel source data, see Supplementary Fig. 1.

or 2′-*O*-methyl were equally functional in directing target cleavage, provided that the piRNA was 5′ monophosphorylated (Extended Data Fig. 1d).

## Purified MIWI piRISC is a slow-acting enzyme

Although affinity-purified MIWI loaded with a piRNA bearing 5′ monophosphorylated and 3′, 2′-*O*-methylated termini specifically cleaved a fully complementary target RNA at the phosphodiester bond that links target nucleotides t10 to t11, the rate of cleavage (0.01 min⁻¹) was more than 300 times slower than that catalysed by mouse AGO2 RISC[36,37] (≥3 min⁻¹). At physiological temperature (37 °C), mouse AGO2 RISC catalyses multiple rounds of target cleavage[38]. By contrast, 5 nM MIWI piRISC cleaved only around 15% of the target RNA (5 nM) after 1 h (Fig. 1b, Extended Data Fig. 1e and Supplementary Fig. 1). Inefficient target cleavage by MIWI piRISC was not caused by the presence of the amino-terminal 3×Flag–SNAP tandem tag: removing the tandem tags using tobacco etch virus protease generated an untagged protein (Extended Data Fig. 1f and Supplementary Data Fig. 1) whose target-cleavage kinetics were identical to that of piRISC assembled with the tagged MIWI (Extended Data Fig. 1g).

The ubiquitously expressed arginine methyltransferase PRMT5 modifies the amino terminus of PIWI proteins, allowing it to bind Tudor domain-containing proteins, many of which are required for piRNA biogenesis, gametogenesis and fertility[39–41]. A potential explanation for the sluggish activity of recombinant MIWI piRISC is that it lacks arginine methylation. We used mass spectrometry to map the positions of methyl arginine in our affinity-purified MIWI. All of the arginine residues previously shown to be methylated in endogenous

MIWI immunoprecipitated from mouse testis[39] were methylated in recombinant MIWI produced in HEK293T cells (Extended Data Fig. 1h).

Another possible explanation for the inefficiency of target cleavage by MIWI piRISC is that the recombinant protein, although properly arginine methylated, lacks other post-translational modifications. To test this idea, we immobilized apo MIWI, incubated it with wild-type (C57BL/6) mouse testis lysate in the presence of an ATP-regenerating system at 25 °C for 15 min, removed the testis lysate by washing, and then loaded MIWI with a synthetic piRNA and eluted the resulting piRISC. Pre-incubation of MIWI with testis lysate either before or after loading with a piRNA did not increase its target-cleaving activity (Extended Data Fig. 2a,b). We conclude that neither a missing post-translational modification nor a tightly associated protein partner is likely to explain the low activity of MIWI piRISC compared to AGO2 RISC.

## MIWI piRISC requires an auxiliary factor

Adding testis lysate increased the rate of target cleavage by affinity-purified MIWI piRISC around 20-fold (Fig. 1b and Extended Data Fig. 2b). This effect cannot be attributed to the lysate contributing additional piRISC, because testis lysate alone did not generate detectable cleavage product (Fig. 1b).

MIWI is first produced as spermatogonia enter meiosis and differentiate into spermatocytes[3,42]. To determine whether the MIWI-potentiating factor is differentially expressed during spermatogenesis, we supplemented the target-cleavage assay with lysate prepared from stage-specific germ cells purified using fluorescence-activated cell sorting (FACS). piRISC-potentiating activity was greatest in lysate from secondary spermatocytes (Fig. 1c), a cell type in which pachytene piRNA-directed target cleavage is readily detected in vivo[20,23]. Moreover, the potentiating activity was testis-specific: lysates from brain, liver or kidney did not enhance piRNA-directed target cleavage by MIWI (Extended Data Fig. 3a). Finally, the potentiating activity was specific for PIWI proteins and had no effect on the rate of target cleavage by mouse AGO2 (Extended Data Fig. 3b).

Three lines of evidence suggest that the MIWI-potentiating activity contains one or more structural $Zn^{2+}$ ions. First, pre-treating testis lysate with the sulfhydryl alkylating agent *N*-ethylmaleimide inactivated the potentiating activity (Extended Data Fig. 3c), indicating that reduced cysteine residues, which often bind divalent metal cations[43], are essential. Second, the MIWI-potentiating activity was unaltered by EGTA, which chelates $Ca^{2+}$, but was irreversibly inactivated by EDTA, which chelates many divalent metals, and by 1,10-phenanthroline, which specifically chelates $Zn^{2+}$ (Extended Data Fig. 3d). Adding additional metal ions did not rescue the activity (Extended Data Fig. 3e–g), suggesting that loss of $Zn^{2+}$ irreversibly denatures the MIWI-potentiating factor, a characteristic of zinc-finger proteins[44,45]. Third, the MIWI-potentiating activity bound more tightly to an immobilized metal affinity resin charged with $Zn^{2+}$ than to resin charged with $Ni^{2+}$ (Extended Data Fig. 3h).

To identify the MIWI-potentiating activity, we developed a chromatographic purification scheme using cation-exchange, hydrophobic-interaction and size-exclusion chromatography (Fig. 2a). Notably, the activity eluted from a Superdex 200 size-exclusion column as a single peak at approximately 17 kDa (Fig. 2b). Together, our data suggest that the MIWI-potentiating activity corresponds to a small, testis-specific, $Zn^{2+}$-binding protein abundantly expressed in meiotic and post-meiotic male germ cells (Fig. 2c).

## GTSF1 potentiates catalysis by MIWI piRISC

Gametocyte specific factor 1 (GTSF1) is a conserved, 19,083 Da, tandem CHHC-type zinc-finger protein essential for piRNA function and fertility in flies[7,8,10], silk moths[11], worms[46] and mice[9,12]. Similar to *Miwi* (also known as *Piwil1*) and *Mili* (also known as *Piwil2*), the mRNA abundance of *Gtsf1*,

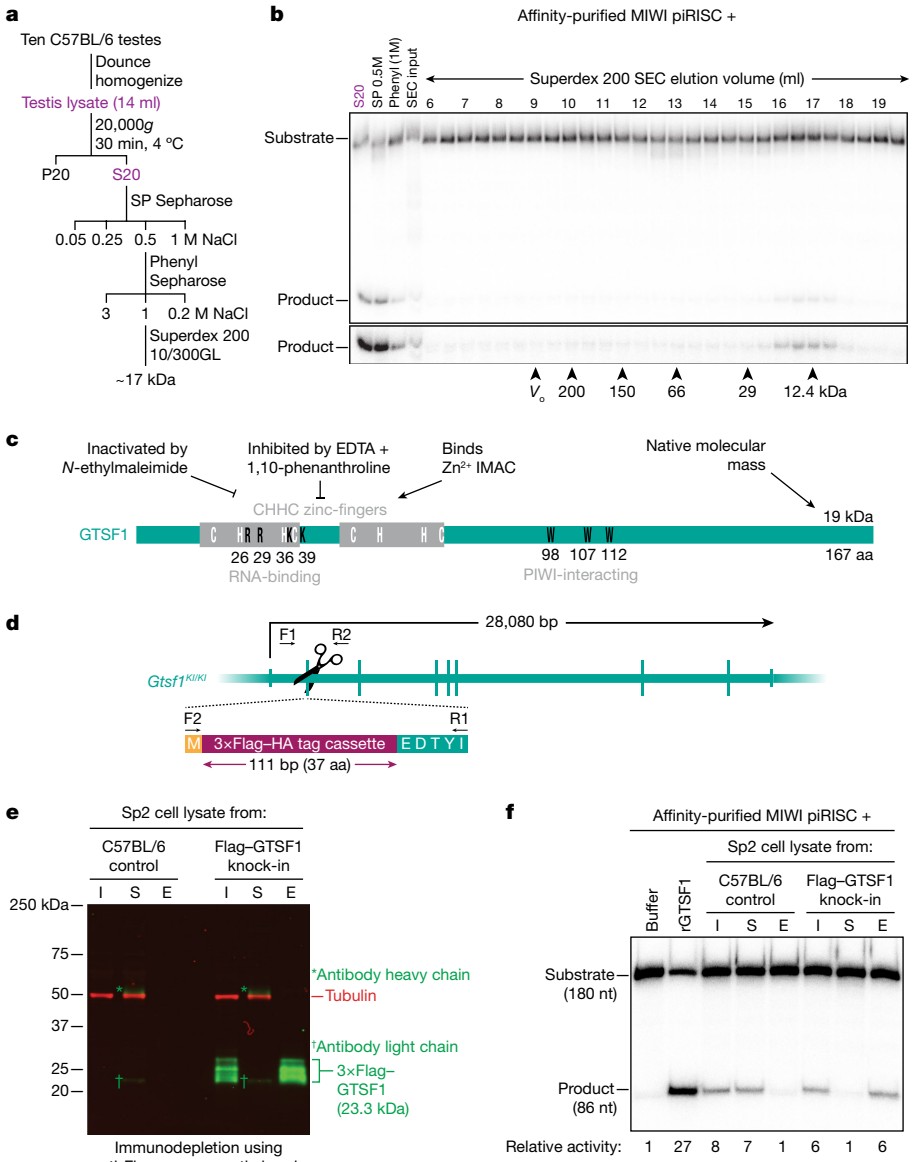

**Fig. 2 | The testis protein GTSF1 potentiates target RNA cleavage by MIWI piRISC. a**, Scheme to purify the MIWI-potentiating factor from mouse testis lysate. P20, pellet; S20, supernatant. **b**, Target-cleavage assay to estimate the apparent molecular mass of the MIWI-potentiating activity by size-exclusion chromatography (SEC). Arrowheads indicate the peak concentration of the molecular mass standards, and their peak elution volumes. $V_0$, void volume; phenyl (1M), 1 M NaCl eluate from hydrophobic-interaction chromatography; SP 0.5M, 0.5 M NaCl eluate from cation-exchange chromatography. **c**, Schematic representing known attributes of GTSF1 and the corresponding properties of the MIWI-potentiating activity in testis lysate.

IMAC, immobilized metal affinity chromatography. **d**, Strategy for creating knock-in mouse expressing 3×Flag–HA–GTSF1 ($Gtsf1^{Flag/Flag}$). F1, F2, R1 and R2 represent sequencing primers used to validate the insertion. The exon encodes an incomplete isoleucine codon, which is completed upon splicing. **e**, Western blotting showing immunodepletion of 3×Flag–HA–GTSF1 from secondary spermatocyte lysate using anti-Flag paramagnetic beads. I, input; S, supernatant; E, 3×Flag peptide eluate. **f**, Product formed by MIWI piRISC in the presence of the indicated components. Numbers below the gel indicate relative amount of product formed in each condition. For gel source data, see Supplementary Fig. 1.

as well as its paralogues *Gtsf1l* and *Gtsf2*, rises sharply as male germ cells enter meiosis I, peaks in secondary spermatocytes, and then declines in round spermatids (Extended Data Fig. 4a). Immobilized mouse GTSF1 captures all three PIWI proteins from testis lysate, but GTSF1 and PIWI proteins co-expressed in HEK293T cells do not co-immunoprecipitate, leading to the suggestion that the GTSF1–PIWI interaction requires additional protein or RNA components[9]. MIWI2-bound piRNAs are absent in *Gtsf1^{-/-}* male mice[9], which are sterile[12], probably because loss of GTSF1 causes loss of MIWI2-directed retrotransposon promoter methylation[47]. Although RNA cleavage by MIWI2 is not required for piRNA biogenesis or function, production of MIWI2 piRISC requires

MILI-dependent piRNA amplification, a process that requires the endonuclease activity of MILI[6]. In flies, piRNA-directed transcriptional silencing of transposons by Piwi, but not Piwi piRISC assembly, requires the GTSF1 orthologue Asterix[7,8].

To test whether the MIWI-potentiating activity in the testis lysate corresponded to GTSF1, we used CRISPR–Cas9 to engineer a mouse strain with a 3×Flag–HA epitope tag inserted into the endogenous *Gtsf1* coding sequence (*C57BL6/J-Gtsf1^{em1(Flag)Pdz}*, hereafter referred to as *Gtsf1^{Flag}*; Fig. 2d). Because the MIWI-potentiating activity was greatest in secondary spermatocytes, we prepared lysate from FACS-purified secondary spermatocytes from homozygous *Gtsf1^{Flag/Flag}* male mice.

Incubation with anti-Flag antibody coupled to paramagnetic beads depleted the lysate of both epitope-tagged GTSF1 (Fig. 2e) and the MIWI-potentiating activity (Fig. 2f). The activity was recovered from the beads by elution with 3×Flag peptide (Fig. 2f). By contrast, lysate from C57BL/6 secondary spermatocytes retained the MIWI-potentiating factor after incubation with anti-Flag antibody, and no activity was eluted from the beads after incubation with Flag peptide. Thus, GTSF1 is necessary to potentiate the catalytic activity of MIWI.

GTSF1 is also sufficient to potentiate the catalytic activity of MIWI. Adding purified recombinant GTSF1 (Extended Data Fig. 4b) to MIWI piRISC increased the rate of target cleavage by MIWI 19–100-fold for three different piRNA sequences (Fig. 3a–c and Extended Data Fig. 4c). First, the addition of 500 nM GTSF1 to MIWI programmed with an artificial, 30-nt piRNA (5 nM) caused the pre-steady-state rate ($k_{burst}$) of target cleavage to increase from 0.010 min$^{-1}$ to 1.1–1.5 min$^{-1}$ (Fig. 3), a rate similar to that of AGO2[36,37] (≥3 min$^{-1}$). Second, programming MIWI with an endogenous mouse piRNA sequence from the 9-qC-10667.1 locus (*pi9*) that is antisense to the L1MC transposon increased the pre-steady-state rate of cleavage of a fully complementary target RNA -19-fold: $k_{burst}$ = 0.033 min$^{-1}$ for piRISC alone and 0.62 min$^{-1}$ with GTSF1 added (Fig. 3c and Extended Data Fig. 5). Finally, for a 6-qF3-28913(−),8009(+) locus (*pi6*) piRNA that directs cleavage at a partially complementary site in the *Scpep1* mRNA in diplotene spermatocytes[20], GTSF1 increased $k_{burst}$ 80-fold, from 0.0040 min$^{-1}$ to 0.32 min$^{-1}$ (Fig. 3c and Extended Data Fig. 5).

Our data suggest that GTSF1 does not promote product release or turnover: the steady-state rate ($k_{ss}$ < 0.005 min$^{-1}$) of target cleavage under multiple-turnover conditions (that is, GTSF1 ≫ target RNA ≫ MIWI) was essentially unchanged from the rate in the absence of GTSF1 (Fig. 3b). MIWI might remain bound to the cleaved products, preventing it from catalysing multiple rounds of target cleavage. Supporting this idea, at incubation times greater than 15 min, 3′-to-5′ exonucleases present in testis lysate degrade the uncut, full-length target RNA, but the 5′ cleavage product remains stable, consistent with its being protected by MIWI bound to its 3′ end (Fig. 1b). In vivo, product release has been proposed to be facilitated in insects and mammals by the RNA-stimulated ATPase Vasa[48–50] (also known as DDX4) and the Vasa-like protein DDX43[51].

## RNA binding is essential for GTSF1 function

Only three eukaryotic proteins are known to contain CHHC zinc-fingers: the spliceosomal RNA-binding protein U11-48K, the TRM13 tRNA methyltransferase, and GTSF1 and its paralogues[52]. In vitro, the first zinc-finger of GTSF1 binds RNA directly; RNA binding requires four basic surface residues[53] (R26, R29, K36 and K39). Potentiation of target cleavage required RNA binding by GTSF1: purified mutant GTSF1(R26A/R29A/K36A/K39A) (Extended Data Fig. 4b) did not detectably increase the rate of catalysis by either MIWI or MILI (Fig. 3d–f and Extended Data Fig. 4c), suggesting that GTSF1 must interact with RNA to function.

## GTSF1 function requires PIWI binding

In flies and mice, GTSF1 interacts with PIWI proteins through conserved aromatic residues in its central region[7,9] (Fig. 2c). Mutating these residues in the recombinant protein—W98A, W107A and W112A for mouse GTSF1—reduced the stimulatory effect of mouse GTSF1 on MIWI in a concentration-dependent manner, supporting the idea that GTSF1 binds MIWI directly when potentiating target cleavage (Fig. 3g and Extended Data Fig. 4c) and suggesting that W98, W107, and W112 define a conserved surface of GTSF1 that interacts with PIWI proteins. We measured $k_{burst}$ as a function of GTSF1 concentration (Supplementary Data Fig. 2). Fitting the data to a hyperbolic function revealed that GTSF1 binds MIWI piRISC more than 60-fold more tightly than GTSF1(W98A/W107A/W112A) (GTSF1: dissociation constant ($K_d$) = 8 nM, 95%

confidence interval 6–9 nM; GTSF1(W98A/W107A/W112A): $K_d$ = 500 nM, 95% confidence interval 100–900 nM) (Fig. 3g). Moreover, PIWI binding appears to be the sole defect in GTSF1(W98A/W107A/W112A), since the first-order rates of target cleavage by MIWI at saturating concentrations of either wild-type or mutant GTSF1 were essentially indistinguishable (wild-type: 1.1 min$^{-1}$, 95% confidence interval 0.7–2 min$^{-1}$; mutant GTSF1: 0.8 min$^{-1}$, 95% confidence interval 0.6–0.9 min$^{-1}$).

## GTSF1 function is evolutionarily conserved

GTSF1 orthologues are found in many metazoan genomes[52], suggesting that GTSF1 may potentiate target cleavage by PIWI proteins in many animals. Supporting this idea, purified recombinant Gtsf1 from the arthropod *Bombyx mori* (Extended Data Fig. 4b) potentiated the catalytic activity of the *B. mori* PIWI protein, Siwi (Extended Data Fig. 6a).

The structure and kinetics of the sponge *Ephydatia fluviatilis* (freshwater sponge) Piwi (EfPiwi) were recently described[35]. Like MIWI, EfPiwi possesses inherently weak catalytic activity, a feature common to all PIWI proteins examined to date. Although the *E. fluviatilis* genome is yet to be sequenced, the closely related, fully sequenced genome of *Ephydatia muelleri* contains a readily identifiable GTSF1 orthologue. Purified recombinant EmGtsf1 (Extended Data Fig. 4c) stimulated the single-turnover catalytic rate of EfPiwi piRISC by around 28-fold (Fig. 3h). Sponges (Porifera) are the sister group to all other animals, the Eumetazoa, having separated around 900 million years ago. Thus, the last common ancestor of all animals probably required GTSF1 to potentiate target cleavage by PIWI proteins.

The GTSF1 tandem zinc-finger domains are conserved across phyla, whereas the GTSF1 central and carboxy-terminal sequence diverges substantially between mammals and arthropods (Extended Data Table 1). For example, the sequences of the mouse and rhesus macaque first and second zinc-fingers are 100% identical, whereas their C-terminal domains share 88.5% identity. The first zinc-finger of mouse GTSF1 is 37.5% and 45.8% identical to its fly and moth orthologues, but the mouse protein shares just 8% and 8.3% identity with the central and C-terminal domains of the fly and moth proteins, respectively. Consistent with the evolutionary divergence of their central and C-terminal domains, testis lysate from rat or rhesus macaque enhanced target cleavage by MIWI piRISC, whereas lysate from *Drosophila melanogaster* or *Trichoplusia ni* ovaries, *T. ni* Hi5 cells or purified recombinant *B. mori* BmGtsf1 did not (Extended Data Fig. 6b,c and Supplementary Data Fig. 1).

## GTSF1 paralogues can distinguish PIWI paralogues

Many animal genomes encode more than one GTSF protein[52] (Extended Data Fig. 6c). For example, *D. melanogaster* has four GTSF paralogues. The *D. melanogaster* OSC and OSS cell lines, which are derived from somatic follicle cells that support oogenesis, express Piwi but lack the PIWI paralogues Aub or Ago3. Piwi-mediated, piRNA-guided transposon silencing in these cells requires Asterix, a GTSF1 orthologue[7,8,10]. In vivo, *asterix* mutants phenocopy *piwi* mutants and are female sterile, even though Piwi is successfully loaded with piRNAs and transits to the nucleus in the absence of Asterix[7,8]. Whether the other fly GTSF paralogues have a function in vivo, perhaps as auxiliary factors for Aub or Ago3, remains to be tested. Similar to fly Asterix, mouse GTSF1 is essential for piRNA function and fertility. In mice, two GTSF1 paralogues, GTSF1L and GTSF2 are also expressed during spermatogenesis and interact with PIWI proteins[54]. Unlike GTSF1, single and double *Gtsf1l*- and *Gtsf2*-knockout males are fertile[54]. Genes encoding the *Gtsf* paralogues are syntenic in mammals, whereas *Gtsf2* is lost in primates (Extended Data Fig. 7).

The central and C-terminal domains of GTSF orthologues are more similar among closely related species than among GTSF paralogues within the same species (Extended Data Figs. 4b and 6c and Extended Data Table 1), further supporting the view that this domain has evolved

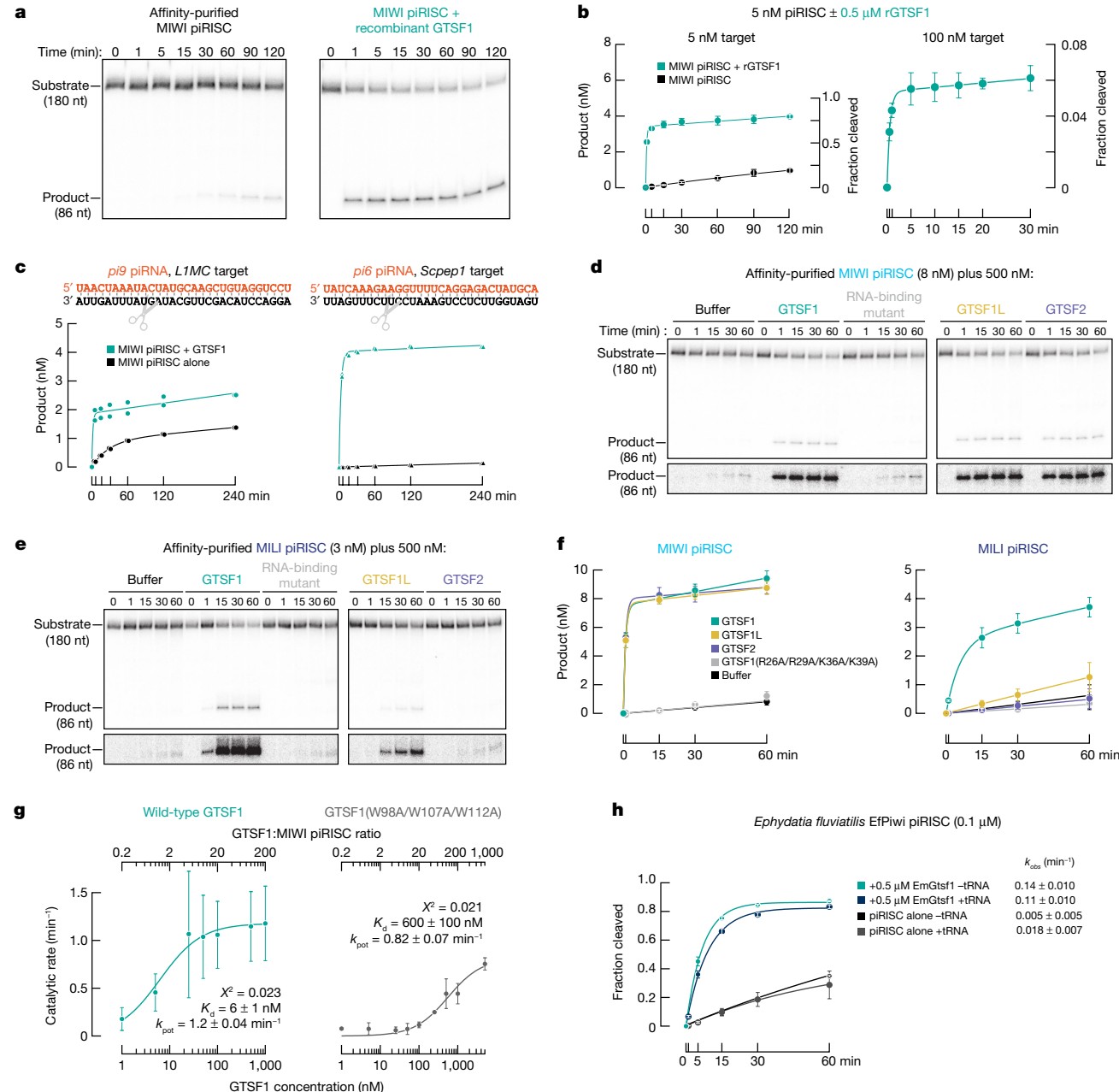

**Fig. 3 | GTSF1 paralogues can distinguish between MIWI and MILI.**
**a**, Representative denaturing polyacrylamide gel electrophoresis showing that purified GTSF1 recapitulates the effect of testis lysate on MIWI catalysis.
**b**, Amount of product generated as a function of time (mean ± s.d., n = 3). Data were fit to the burst-and-steady-state equation. Data for MIWI piRISC alone are from Fig. 1. **c**, Product formed as a function of time by MIWI programmed with *pi6* and *pi9* piRNAs for two independent trials. The mean values of the two trials were fit to the burst-and-steady-state equation. **d,e**, Representative denaturing polyacrylamide gel images of the assay to test GTSF1 mutants and paralogues

in target cleavage by MIWI (**d**) or MILI (**e**) piRISC. **f**, Product formed as a function of time (mean ± s.d., n = 3) fit to the burst-and-steady-state equation. **g**, Pre-steady-state first-order rate constants of target cleavage by MIWI piRISC in the presence of either wild-type GTSF1 or the PIWI-interacting mutant GTSF1(W98A/W107A/W112A) were plotted as a function of GTSF1 concentration. Data are mean ± s.d., n = 3. $k_{pot}$, maximum observable rate. **h**, Single-turnover rate of target cleavage by EfPiwi piRISC in the presence or absence of EmGtsf1 or yeast tRNA (mean ± s.d., n = 3). For gel source data, see Supplementary Fig. 1.

to bind specific PIWI proteins. Consistent with this idea, mouse GTSF1, GTSF1L and GTSF2 differ in their ability to potentiate target cleavage by MIWI and MILI. Whereas GTSF1 accelerated target cleavage by both MIWI and MILI, purified recombinant GTSF1L and GTSF2 (Extended Data Fig. 4b) efficiently potentiated target cleavage by MIWI but not by MILI (Fig. 3d–f and Extended Data Fig. 4c). GTSF2 was unable to detectably increase the rate of target cleavage by MILI. Although GTSF1L had a modest effect on the rate of target cleavage by MILI piRISC, this

enhancement was less than one-sixteenth that provided by GTSF1 and half that of the PIWI binding mutant GTSF1(W98A/W107A/W112A) (Fig. 3d–f and Extended Data Fig. 4c). We conclude that GTSF1L and GTSF2 are specialized to potentiate target cleavage by MIWI piRISC.

Silk moth BmGtsf1-like is more similar to mouse GTSF1 than to BmGtsf1 (Extended Data Fig. 6c and Extended Data Table 1). BmGtsf1-like increased the amount of target cleaved by MIWI piRISC by 2.6-fold (s.d. = 0.1; P = 0.0027 by ANOVA with Dunnett's post-hoc

test) but had no detectable effect on MILI (Extended Data Fig. 6a), further supporting the idea that the GTSF central domain determines the affinity of the protein for specific PIWI proteins. In vivo, BmGtsf1 interacts with BmSiwi and is required for transposon silencing and sex determination[11]. BmGtsf1 increased the rate of target cleavage by affinity-purified Siwi but not that of the other silk moth PIWI protein, BmAgo3 (Extended Data Fig. 6a).

## MIWI and MILI require extensive base pairing

piRNA–target RNA complementarity from g2–g22 (that is, 21 base pairs) is required for efficient target cleavage directed by endogenous piRNAs loaded into MIWI piRISC immunoprecipitated from adult mouse testis[5]. However, GTSF1 does not detectably co-immunoprecipitate with MIWI piRISC from mouse testis[5,39]. A requirement for 21-base-pair complementarity between target and guide is, to our knowledge, unique among Argonaute proteins: fly Ago2 slices a target with as few as 11 contiguous base pairs[55]; mammalian AGO2 requires only 11 contiguous base pairs for detectable cleavage[56]; and the eubacterial DNA-guided DNA endonuclease TtAgo[57] requires as few as 14.

Affinity-purified MIWI, programmed with either of two different synthetic 30-nt piRNAs, readily cleaved target RNA complementary to guide nucleotides g2–21 but not a target complementary to g2–g16 (Extended Data Fig. 8a,b). Under multiple-turnover conditions with saturating amounts of purified, recombinant GTSF1 ([GTSF1] ≫ [target] > [piRISC]), MIWI readily cleaved a target RNA with 19 nucleotides (g2–g20) complementary to its synthetic piRNA guide (Fig. 4a, top). The lower background of single-turnover experiments ([GTSF1] ≫ [piRISC] > [target]) enabled longer incubation times. Under these conditions, we could detect GTSF1-stimulated cleavage of a target RNA with as few as 15 complementary nucleotides (g2–g16; Fig. 4a, bottom). We note that the pachytene stage of meiosis in mouse spermatogenesis lasts about 175 h, and the pachytene piRNA pathway components are expressed until at least the round spermatid stage, a time interval spanning more than 400 h.

## Guide length limits target cleavage by MIWI

In vivo, piRNAs are trimmed to a length characteristic of the PIWI protein in which they reside: around 30 nt for MIWI and 26–27 nt for MILI[13,14,58]. An attractive hypothesis is that these piRNA lengths are optimal for target cleavage catalysed by the specific PIWI protein. Our data suggest a more complex relationship between piRNA length, PIWI protein identity and target complementarity. In the presence of GTSF1, MIWI loaded with a 30-nt piRNA and MILI loaded with a 26-nt piRNA readily cleaved a fully complementary target RNA in a 60-min reaction (Fig. 4b). By contrast, neither piRISC cleaved a target complementary to piRNA positions g2–g16 (Fig. 4a,b and Extended Data Fig. 8a,b). Similarly, both MIWI and MILI loaded with a 26- or 21-nt guide produced little cleavage for a target complementary to positions g2–g16, although both guide lengths supported cleavage of a fully complementary target (Fig. 4b). In fact, for MIWI, a 21mer was more active than a 30-nt guide (Fig. 4b). Without GTSF1, MIWI or MILI loaded with any of these guide lengths produced little cleaved target in 60 min (Fig. 4b). Of note, MIWI or MILI loaded with a 16-nt guide RNA, a piRNA length that is not present in vivo, readily cleaved the RNA target (Fig. 4b).

GTSF1 accelerates the rate of pre-steady-state target cleavage ($k_{burst}$) by MIWI and MILI but has little effect on their slow rates of steady-state cleavage ($k_{ss}$), suggesting that piRISC remains bound to its cleavage by-products (Fig. 3b). We incubated GTSF1 and MIWI—loaded with a 30-, 26-, 21- or 16-nt guide—with a target RNA fully complementary to each guide and measured $k_{burst}$ and $k_{ss}$ (Fig. 4c). As the guide length decreased, $k_{burst}$ decreased and $k_{ss}$ increased. Compared with its native 30-nt guide length, the 16-nt guide increased $k_{ss}$ approximately ninefold

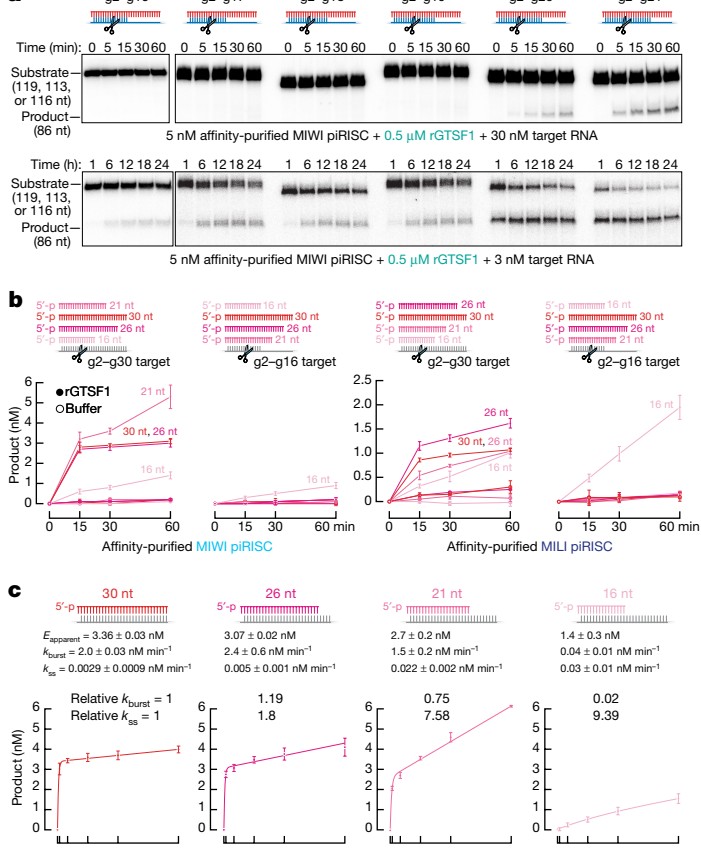

**Fig. 4 | Cleavage by MIWI piRISC is sensitive to the complementarity between the piRNA and target. a**, MIWI piRISC target cleavage in the presence of GTSF1 for targets with varying complementarity to synthetic piRNA guide. Top, multiple-turnover conditions; bottom, single-turnover conditions. All reactions contained saturating amounts of GTSF1. **b**, Target-cleavage assay using targets complementary to piRNA guide nucleotides g2–g16 or g2–g30 and MIWI or MILI loaded with piRNAs of the indicated lengths, with or without GTSF1 (mean ± s.d., $n$ = 3). **c**, Absolute and relative pre-steady-state and steady-state rates of cleavage of the g2–g30 target by MIWI loaded with piRNAs of the indicated lengths in the presence of GTSF1 (mean ± s.d., $n$ = 3), $E_{apparent}$, apparent active enzyme concentration estimated from fitting data to equation (1). For gel source data, see Supplementary Fig. 1.

and decreased $k_{burst}$ by more than 50-fold. These data suggest that as the guide was shortened, the cleaved products were released more rapidly.

We estimated the binding affinity ($\Delta G$) of the piRNA for its target using the nearest-neighbour rules for base pairing at 37 °C. As the strength of the piRNA base pairing to the target increased, the rate of pre-steady-state cleavage increased (Fig. 5a), consistent with base pairing serving to extract the 3′ end of the piRNA from the PAZ domain, facilitating the transition to a more catalytically competent piRISC conformation[59]. Conversely, decreased base-pairing strength increased the steady-state rate of target cleavage, supporting the view that for biologically relevant piRNA lengths, product release is the rate-determining step for MIWI-catalysed target cleavage (Fig. 5a).

## Discussion

In nearly all animals, both piRNA biogenesis and piRNA function require the PIWI endoribonuclease activity. Yet purified mouse MIWI and MILI are intrinsically slow to cleave complementary target RNAs. Our data show that unlike Argonaute proteins, PIWI proteins require an auxiliary factor, GTSF1, to efficiently cleave their RNA targets.

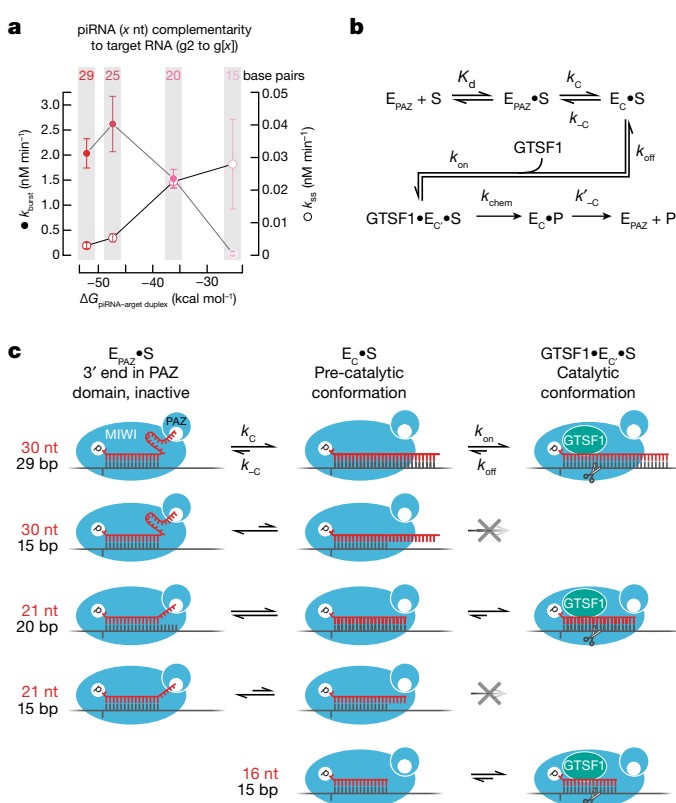

**Fig. 5 | A model for the function of guide length and GTSF1 in target cleavage by PIWI proteins. a**, The energy of base pairing, estimated by standard nearest-neighbour methods versus pre-and steady-state rates of GTSF1-potentiated target cleavage (mean ± s.d., $n = 3$), directed by piRNAs of different lengths loaded into MIWI. The same target, which was fully complementary to each piRNA, was used in all experiments. **b**, piRNA-directed, MIWI-catalysed target cleavage is envisioned to require two sequential conformational changes: (1) a target-dependent conformational change in piRISC ($E_{PAZ}$) in which the piRNA 3′ end leaves the PAZ domain, allowing extensive base pairing of the piRNA with the target substrate (S); and (2) GTSF1-dependent conversion of this piRISC pre-catalytic state ($E_C$) to the fully competent catalytic state ($E_{C'}$). P, cleaved target products. $K_d$, dissociation constant; $k_c$ and $k_{-c}$, rate constants for pre-catalytic state; $k_{on}$ and $k_{off}$, rate constants for GTSF1 binding; $k_{chem}$, rate constant for the endoribonucleolytic chemistry step; $k'_{-c}$, rate constant for piRISC (EPAZ) regeneration. **c**, Proposed effects of different piRNA lengths, extent of guide–target complementarity and GTSF1 on the forward and reverse rates of the two conformational rearrangements. The wide, central cleft, as observed in the cryo-electron microscopy structure of *E. fluviatilis* Piwi-A[35], is envisioned to allow the central region of a 30 nt piRNA to be mobile and exposed to solvent when its 3′ end is secured to the PAZ domain (top right).

The ability of GTSF1 to potentiate PIWI-catalysed target cleavage provides a biochemical explanation for the genetic requirement for this small zinc-finger protein in the piRNA pathway. GTSF1 function requires that it bind both RNA and the PIWI protein, and differences among C-terminal domains restrict individual GTSF1 paralogues to specific PIWI proteins.

We propose a testable kinetic scheme for target cleavage by MIWI (Fig. 5b,c and Extended Data Fig. 9) that incorporates the requirement for GTSF1 and the observation that a 16-nt guide changes the rate-determining step of target cleavage catalysed by MIWI. As originally proposed for fly Ago2[59], piRISC bound to a target is presumed to exist in two states: one in which the piRNA 3′ end is secured in the PAZ domain and a competing, pre-catalytic conformation in which the piRNA is fully base paired to its target. Structures of Piwi-A from the freshwater sponge *E. fluviatilis* show that extensive pairing between a piRNA and

its target induces a catalytically competent conformation in which the PAZ domain is rotated away from the piRNA 3′ end[35]. We propose that the PAZ-bound state is more favourable for guides bound to PIWI proteins than to AGOs, requiring a high degree of complementarity between guide and target to extract the piRNA 3′ terminus from the PAZ domain. Our data also suggest that a slow rate of product release after cleavage results directly from the extensive piRNA–target RNA complementarity required for this conformational change. A 16 nt piRNA allows MIWI to more easily adopt a pre-catalytic state, perhaps because the 3′ end of the short guide cannot reach the PAZ domain. Notably, single-stranded siRNA guides as short as 14 nt allow mammalian AGO3, initially believed to have lost its endonuclease activity, to efficiently cleave RNA targets[60]. In golden hamsters, piRNAs bound to PIWIL1 are initially around 29 nt long, but a shorter population of approximately 23 nt piRNAs appears at metaphase II and predominates in 2-cell embryos[61]. piRNAs bound to PIWIL3, a female-specific PIWI protein that is absent from mice, are around 19 nt long in hamster and around 20 nt long in human oocytes[61,62]. We speculate that these short piRNAs enable PIWIL1 and PIWIL3 to function as multiple-turnover endonucleases.

Our model proposes that GTSF1 recognizes the pre-catalytic piRISC state, facilitating a second conformational change in PIWI proteins that may occur spontaneously in Argonaute clade proteins. GTSF1 binding probably stabilizes the catalytically active conformation, facilitating target cleavage, but has no detectable effect on subsequent release of the cleaved products. Slow product release may be a general property of PIWI proteins: purified *E. fluviatilis* PIWI similarly catalyses only a single round of target cleavage[35].

Potentiation of the endonuclease activity of PIWI proteins by GTSF1 probably represents its ancestral function. It remains unknown why GTSF1 is required for the function of PIWI proteins such as mouse MIWI2[6,9] and fly Piwi[7,10], which silence transcription rather than cleave target RNAs. We propose that GTSF1 stabilizes the functional conformation of non-catalytic PIWI proteins, enabling them to bind the downstream proteins required for transcriptional silencing.

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

## Methods

### Plasmids and cell lines

Supplementary Data Table 1 provides the sequences of all oligonucleotides used. To create pScalps_Puro_eGFP, an IRES-driven eGFP was inserted downstream of the multiple cloning site and upstream of the puromycin coding sequence of the lentivirus transfer vector pScalps_Puro. MIWI cDNA was obtained from Mammalian Gene Collection (https://genecollections.nci.nih.gov/MGC/). MILI cDNA was amplified by RT–PCR from mouse testis total RNA. Gibson assembly and restriction cloning were used to clone the MILI and MIWI coding sequences into pScalps_Puro_eGFP, fusing them in-frame with N-terminal 3×Flag and SNAP tags. Lentivirus transfer vectors were packaged by co-transfection with psPAX2 and pMD2.G (4:3:1) using TransIT-2020 (Mirus Bio) in HEK293T cells. Supernatant containing lentivirus was used to transduce HEK293T cells in the presence of 16 µg ml$^{-1}$ polybrene (Sigma) to obtain stable PIWI-expressing cell lines. Three sequential transductions were performed to maximize recombinant protein production. The transduced cells were selected in the presence of 2 µg ml$^{-1}$ Puromycin for two weeks, then the cells expressing the 5–10% highest eGFP fluorescence were selected by FACS (UMASS Medical School Flow Cytometry Core). Selected cells stably expressing the recombinant PIWI proteins were expanded, collected, and cell pellets flash-frozen and stored at −80 °C.

Mouse GTSF1 cDNA was synthesized at Twist Biosciences and cloned into pCold-GST (Takara Bio) bacterial expression vector by restriction cloning. GTSF1 mutants, GTSF1L, and GTSF2- expressing pCold-GST vectors were synthesized at Twist Biosciences. pIZ-Flag-6×His-Siwi was described previously[63]. Flag-tagged MILI and MIWI-expressing vectors were the kind gift of Shinpei Kawaoka (Kyoto University, Kyoto, Japan). MmGtsf1 cDNA was amplified by PCR with reverse transcription (RT–PCR) from mouse spermatogonial stem cell total RNA[64]. BmGtsf1 and BmGtsf1-like cDNAs were amplified by RT–PCR from BmN4 cell total RNA. The amplified cDNA fragment and a DNA fragment coding V5SBP were cloned into pcDNA5/FRT/TO vector (Thermo Fisher Scientific) by In-fusion cloning (Takara). The *E. muelleri* Gtsf1 coding sequence was obtained from[65] and cloned into bacterial expression vector pSV272 (IJM).

### Mice

Generation of 3×Flag–HA–tagged GTSF1 mice (*C57BL6/J-Gtsf1*$^{em1(Flag)Pdz}$) was performed at Cyagen. The coding sequence for the tags was inserted into the endogenous locus by CRISPR–Cas9. In brief, fertilized mouse embryos were injected with the sgRNA targeting the sequence GTCTT CCATGCTGATGGCAAAGG (PAM), a 3×Flag–HA-tag cassette HDR donor, and Cas9 mRNA. Supplementary Data Table 1 provides the sequences of the HDR donor and oligonucleotide primers used for genotyping. Founder $F_0$ mice were genotyped and bred to generate $F_1$ mice carrying the germline-transmitted knock-in allele. Mice were maintained and used according to the guidelines of the Institutional Animal Care and Use Committee of the University of Massachusetts Chan Medical School (A201900331). C57BL/6J mice (RRID: IMSR_JAX:000664) were used as wild-type control, where indicated. Animals were housed in an AALAC-accredited barrier facility with controlled temperature (22 ± 2 °C), relative humidity (40% ± 15%), and a 12 h:12 h dark:light cycle.

### Recombinant protein purification

**PIWI proteins.** PIWI-expressing stable cells were collected by centrifugation and stored at −80 °C until they were lysed in 10 ml lysis buffer (30 mM HEPES-KOH, pH 7.5, 100 mM potassium acetate, 3.5 mM magnesium acetate, 1 mM DTT, 0.1% v/v Triton X-100, 20% v/v glycerol, and 1× protease inhibitor cocktail (1 mM 4-(2-aminoethyl)benzenesulfonyl fluoride hydrochloride (Sigma, A8456), 0.3 µM aprotinin, 40 µM betanin hydrochloride, 10 µM. E-64 (Sigma, E3132), 10 µM leupeptin hemisulfate)) per g frozen cells. Cell lysis was monitored by staining with trypan blue. Crude cytoplasmic lysate was clarified at 20,000$g$ (S20) or 100,000$g$ (S100) for 30 min, aliquoted, flash-frozen, and stored at −80 °C. To capture PIWI proteins, clarified lysate was incubated with 20 µl anti-Flag M2 paramagnetic beads (Sigma) per ml of lysate for 2 h to overnight rotating at 4 °C. Beads were washed five times with high salt wash buffer (30 mM HEPES-KOH, pH 7.5, 2 M potassium acetate, 3.5 mM magnesium acetate, 1 mM DTT), twice with low salt buffer (high salt wash buffer except containing 100 mM potassium acetate and 0.01% v/v Triton X-100). To assemble piRISC, beads were resuspended in low salt buffer containing 100 nM synthetic guide piRNA and incubated with rotation at 37 °C or room temperature for 30 min. MIWI piRISC or unloaded apo MIWI was eluted from the beads with 200 ng µl$^{-1}$ 3×Flag peptide in lysis buffer without 0.1% v/v Triton X-100 or protease inhibitors for 2 h at 4 °C. Eluate containing PIWI piRISC was aliquoted and stored at −80 °C. EfPiwi piRISC purification has been described[35]. AGO2 siRISC was purified using oligo-affinity purification as described[66].

**GTSF1, GTSF1 mutants and GTSF1 homologues.** pCold-GST GTSF-expression vectors were transformed into Rosetta-Gami 2 competent cells (Novagen). Cells were grown in the presence of 1 µM ZnSO$_4$ at 37 °C until OD$_{600}$ ~0.6–0.8, then chilled on ice for 30 min to initiate cold shock. Protein expression was induced with 0.5 mM IPTG for 18 h at 15 °C. Cells were collected by centrifugation, washed twice with PBS, and cell pellets were flash frozen and stored at −80 °C. Cell pellets were resuspended in lysis/GST column buffer containing 20 mM Tris-HCl pH 7.5, 500 mM NaCl, 1 mM DTT, 5% v/v glycerol, and complete EDTA-free protease inhibitor cocktail (Roche). Cells were lysed by a single pass at 18,000 psi through a high-pressure microfluidizer (Microfluidics M110P), and the resulting lysate clarified at 30,000$g$ for 1 h at 4 °C. Clarified lysate was filtered through a 0.2 µm low-protein binding syringe filter (Millex Durapore; EMD Millipore) and applied to glutathione Sepharose 4b resin (Cytiva) equilibrated with GST column buffer. After draining the flow-through, the resin was washed with 50 column volumes GST column buffer. To elute the bound protein and cleave the GST tag in a single step, 50 U HRV3C Protease (Novagen) was added to the column, the column was sealed and incubated for 3 h at 4 °C, following which, the column was drained to collect the cleaved protein. The protein was diluted to 50 mM NaCl and further purified using a HiTrap Q (Cytiva) anion-exchange column equilibrated with 20 mM Tris-HCl, pH 7.5, 50 mM NaCl, 1 mM DTT, and 5% v/v glycerol. The bound protein was eluted using a 100–500 mM NaCl gradient in the same buffer. Peak fractions were analysed for purity by SDS–PAGE and the purest were pooled and dialysed into storage buffer containing 30 mM HEPES-KOH, pH 7.5, 100 mM potassium acetate, 3.5 mM magnesium acetate, 1 mM DTT, 20% v/v glycerol. To avoid precipitation of GTSF2, which has a pI of 7.3, 20 mM Tris-HCl, pH 8.8, was substituted for HEPES-KOH during purification and dialysis.

For Extended Data Fig. 6a, HEK293T cells were transfected with mouse and insect GTSF1 coding sequences cloned into pcDNA5 expression vectors using Lipofectamine 3000 (Thermo Fisher Scientific) according to the manufacturer's instructions. Cells were collected 36 h later and homogenized in lysis buffer (20 mM Tris-HCl (pH 7.4), 150 mM NaCl, 1.5 mM MgCl$_2$, 0.15% v/v Triton X-100, 100 µg ml$^{-1}$ RNase A (Qiagen), 0.5 mM DTT, 1× Complete EDTA-free protease inhibitor (Roche)). After centrifugation at 17,000$g$ for 20 min at 4 °C, additional NaCl (0.65 M final concentration) and 1% v/v Triton X-100 (final concentration) were added to the supernatant and the lysate was incubated with Streptavidin–Sepharose High Performance (Cytiva) beads at 4 °C for 1 h. The beads were washed with wash buffer (20 mM Tris-HCl (pH 7.4), 1 M NaCl, 1.5 mM MgCl$_2$, 1% v/v Triton X-100, 0.5 mM DTT) five times and rinsed with lysis buffer without RNase A. SBP-tagged recombinant proteins were eluted with elution buffer (30 mM HEPES-KOH, pH 7.4, 100 mM potassium acetate, 2 mM magnesium acetate, 2.5 mM biotin, 0.5 mM DTT).

For Fig. 3h, EmGtsf1 expression vector was transformed into BL21(DE3) cells (New England Biolabs). Transformed cells were grown

in LB supplemented with 1 µM ZnSO$_4$ at 37 °C until OD600 ~0.6–0.8. The incubation temperature was lowered to 16 °C and protein expression was induced by addition of 1 mM IPTG for 4 h. Cells were collected by centrifugation and cell pellets were flash frozen in liquid nitrogen and stored at −80 °C for future use. Thawed cell pellets were resuspended in lysis buffer (50 mM Tris, pH 8, 300 mM NaCl, 0.5 mM TCEP) and passed through a high-pressure (18,000 psi) microfluidizer (Microfluidics M110P) to induce cell lysis. The resulting lysate was clarified by centrifugation at 30,000$g$ for 20 min at 4 °C. Clarified lysate was applied to Ni-NTA resin (Qiagen) and incubated for 1 h. The resin was extensively washed with nickel wash buffer (300 mM NaCl, 20 mM imidazole, 0.5 mM TCEP, 50 mM Tris, pH 8). Protein was eluted in four column volumes of nickel elution buffer (300 mM NaCl, 300 mM imidazole, 0.5 mM TCEP, 50 mM Tris, pH 8). TEV protease was added to the eluted protein to induce cleavage and removal of the N-terminal His$_6$ and MBP tags. The resulting mixture was dialysed against HiTrap Dialysis Buffer (300 mM NaCl, 20 mM imidazole, 0.5 mM TCEP, 50 mM Tris, pH 8) at 4 °C overnight. The dialysed protein was then passed through a 5-ml HiTrap Chelating column (Cytiva) and the unbound material collected. Unbound material was concentrated and further purified by size-exclusion chromatography using a Superdex 200 Increase 10/300 column (Cytiva) equilibrated in 50 mM Tris, pH 8, 300 mM NaCl, and 0.5 mM TCEP. Peak fractions were analysed for purity by SDS–PAGE, and the purest were pooled, concentrated to 150 µM, aliquoted, and stored at −80 °C.

## Immunodepletion and western blotting

Approximately 300,000 FACS-purified secondary spermatocytes (Sp2) from wild-type (C57BL/6) or 3×Flag–HA–GTSF1-expressing (*Gtsf-1$^{Flag/Flag}$*) mice were lysed in lysis buffer (30 mM HEPES-KOH, pH 7.5, 100 mM potassium acetate, 3.5 mM magnesium acetate, 1 mM DTT, 0.1% v/v Triton X-100, 20% v/v glycerol, and 1× protease inhibitor cocktail (1 mM 4-(2-aminoethyl)benzenesulfonyl fluoride hydrochloride (Sigma, A8456), 0.3 µM aprotinin, 40 µM betanin hydrochloride, 10 µM E-64 (Sigma, E3132), 10 µM leupeptin hemisulfate)). Crude cytoplasmic lysate was clarified at 20,000$g$ (S20) for 30 min at 4 °C. The S20 protein concentration was measured using Bradford assay. The S20 was incubated with anti-Flag paramagnetic beads overnight at 4 °C. The supernatant was removed, and the beads washed thrice with lysis buffer but containing 1 M potassium acetate (high salt wash) and then twice with lysis buffer. Bound proteins were eluted by incubation with 200 ng µl$^{-1}$ 3×Flag peptide for 2 h at 4 °C. The extent of depletion was assayed by western blotting. In brief, the samples were resolved on SDS–PAGE and transferred to nitrocellulose membrane (Amersham Protran 0.45 NC, Cytiva). The membrane was probed using mouse anti-Flag antibody (Sigma, F1804; 1:1,000 in blocking buffer, Rockland Immunochemicals, MB-070) and rabbit anti-tubulin (Cell Signaling, 2144; 1:1,000) for 2 h at room temperature. After washing 5× with TBS (+0.1% v/v Tween-20), the membrane was incubated with secondary antibodies—donkey anti-rabbit IRDye 680RD (LI-COR, 926-68073; 1:15,000) and goat anti-mouse IRDye 800CW (LI-COR, 926-32210; 1:15,000)—for 30 min at room temperature. The membrane was washed 5× with TBS (+0.1% v/v Tween-20) and imaged using Odyssey Infrared Imaging System (LI-COR).

## Northern blotting

Northern blotting was performed as described[67]. In brief, piRNA guide standards and PIWI RISCs were first resolved on a denaturing 15% polyacrylamide gel, followed by transfer to Hybond-NX (Cytiva) neutral nylon membrane by semi-dry transfer at 20 V for 1 h. Next, crosslinking was performed in the presence of 0.16 M EDC in 0.13 M 1-methylimidazole, pH 8.0, at 60 °C for 1 h. The crosslinked membrane was pre-hybridized in Church's buffer (1% w/v BSA, 1 mM EDTA, 0.5 M phosphate buffer, and 7% w/v SDS) at 45 °C for 1 h. Radiolabeled, 5′ $^{32}$P-DNA probe (25 pmol) in Church's buffer was added to the membrane

and allowed to hybridize overnight at 45 °C, followed by five washes with 1× SSC containing 0.1% w/v SDS. The membrane was air dried and exposed to a storage phosphor screen.

## Chromatographic fractionation of the MIWI-potentiating activity

Dissected animal tissues were homogenized in lysis buffer in a Dounce homogenizer using 10 strokes of the loose-fitting pestle A, followed by 20 strokes of tight-fitting pestle B. Lysate was clarified at 20,000$g$, followed by 0.2 µm filtration to yield an S20 for further chromatographic purification. Lysates used without further purification were directly prepared in 30 mM HEPES-KOH, pH 7.5, 100 mM potassium acetate, 3.5 mM magnesium acetate, 1 mM DTT, and 20% v/v glycerol ('dialysis buffer') with 1× protease inhibitor homemade cocktail; column fractions were dialysed into this buffer before assaying. Protein concentration was measured using the BCA assay. Chromatography buffers were filtered prior to use.

For chromatography, lysate was prepared as described except using 30 mM HEPES-KOH, pH 7.5, 50 mM NaCl, 1 mM DTT, 5% v/v glycerol, and protease inhibitors. The lysate was applied to HiTrap SP column (Cytiva) equilibrated with the lysis buffer. The column was washed, and the bound proteins eluted stepwise using increasing NaCl concentrations. The NaCl content of the SP column fractions containing the peak MIWI-potentiating activity was adjusted to 2 M and applied to HiTrap Phenyl (Cytiva) equilibrated with column buffer containing 2 M NaCl. Bound proteins were eluted stepwise using decreasing NaCl concentrations. The peak MIWI-potentiating fractions elute from the HiTrap Phenyl column were pooled, concentrated (10 kDa MWCO Amicon Ultra filter), and applied to Superdex 200 Increase 10-300 GL size-exclusion chromatography column (bed volume ~24 ml) equilibrated with the dialysis buffer but containing 5% v/v glycerol. The void volume ($V_0$) of the gel filtration column was determined with blue dextran, and all fractions (0.5 ml each), starting from just before $V_0$ to the end of the column ($V_t$) were assayed for MIWI-potentiating activity. The molecular mass of the potentiating activity was determined relative to molecular mass markers (beta amylase, 200 kDa; alcohol dehydrogenase, 150 kDa; albumin, 66 kDa; carbonic anhydrase, 29 kDa; and cytochrome C, 12.4 kDa).

## Zn$^{2+}$ and Ni$^{2+}$ immobilized metal affinity chromatography

HiTrap Chelating HP (Cytiva) columns were charged with 0.1 M NiSO$_4$ or ZnSO$_4$, washed with water, and then equilibrated in column buffer (20 mM potassium phosphate buffer, pH 7.5, 500 mM NaCl, 0.5 mM DTT, 5% v/v glycerol). S20 testis lysate was applied to the column, the flow-through was collected, and the column was washed with the column buffer until absorbance at 280 nm stabilized. Bound proteins were eluted in two steps: first, with 20 mM potassium phosphate, pH 7.5, 2 M ammonium chloride, 0.5 mM DTT, 5% v/v glycerol (elution buffer 1), and, second, with 20 mM potassium phosphate, pH 7.5, 500 mM NaCl, 200 mM imidazole, pH 8.0, 0.5 mM DTT, 5% v/v glycerol (elution-buffer 2). The peak of each step was dialysed into the dialysis buffer and assayed for the ability to potentiate MIWI catalysis.

## Target-cleavage assays

Target RNA substrates for in vitro cleavage assays were prepared as described[36,68,69]. In brief, piRNA target site-containing templates were amplified by PCR, in vitro transcribed with T7 RNA polymerase, purified by urea PAGE, and radiolabelled using α-$^{32}$P GTP (3,000 Ci mmol$^{-1}$; Perkin Elmer), $S$-adenosyl methionine, and vaccinia virus RNA guanylyl transferase as described[69]. Unincorporated α-$^{32}$P GTP was removed using a G-25 spin column (Cytiva), and target RNA was gel purified. In Extended Data Fig. 8a, target cleavage was monitored using synthetic RNA oligonucleotides radiolabelled by ligating [5′-$^{32}$P] cytidine 3′,5′-bisphosphate to the 3′ end of the target with T4 RNA ligase I (Ambion). The [5′-$^{32}$P] cytidine 3′,5′-bisphosphate was prepared by incubating 1 mM cytidine 3′-monophosphate (Sigma) with 312.5 pmol

[γ-32P] ATP (6,000 Ci mmol$^{-1}$; Perkin Elmer) with 25 U T4 polynucleotide kinase (NEB) at 37 °C for 1 h, followed by 70 °C for 30 min to inactivate the kinase.

Radiolabelled target (3–100 nM final concentration) was added to purified PIWI piRISC (2–8 nM), plus ~1 μg tissue or sorted germ cell lysate per 10 μl reaction volume or 0.5 μM (final concentration) of purified GTSF protein. At the indicated times, an aliquot of a master reaction was quenched in 4 volumes 50 mM Tris-HCl, pH 7.5, 100 mM NaCl, 25 mM EDTA, 1% w/v SDS, then proteinase K (1 mg ml$^{-1}$ final concentration) was added and incubated at 45 °C for 15 min. An equal volume of urea loading buffer (8 M urea, 25 mM EDTA) was added to the reaction time points, heated at 95 °C for 2 min, and resolved by 7–10% denaturing PAGE. Gels were dried, exposed to a storage phosphor screen, and imaged on a Typhoon FLA 7000 (GE Healthcare).

The raw image file was used to quantify the substrate and product bands, corrected for background. Data were fit to the reaction scheme

$$E + S \underset{k_{-1}}{\overset{k_1}{\rightleftharpoons}} ES \overset{k_2}{\to} EP \overset{k_3}{\to} E + P$$

using the burst-and-steady-state equation:

$$[P] = f(t) = [E][(k_2/(k_2 + k_3))^2 \times (1 - e^{-(k_2 + k_3)t}) + (k_2 k_3/(k_2 + k_3))t] \quad (1)$$

The time-dependence of product formation corresponded to a pre-steady-state exponential burst ($k_{burst} = k_2 + k_3$) followed by a linear steady-state phase, described by $k_{cat}$, where $k_{cat} = k_{ss} = k_2 k_3/(k_2 + k_3)$.

The affinity ($K_d$) of wild-type or GTSF1(W98A/W107A/W112A) for MIWI piRISC–target ternary complex and the maximum observable rate ($k_{pot}$) were estimated by measuring the pre-steady-state rate of target cleavage ($k_{burst}$) of MIWI at increasing concentrations of GTSF1 (1–5,000 nM) and fitting the data[70] to equation (2):

$$k_{burst} = k_{pot}[GTSF1]/(K_d + [GTSF1]) \quad (2)$$

All data were graphed using Igor Pro 8.

### Flag–Siwi, Flag–MIWI and Flag–MILI target-cleavage assays

Vectors expressing Flag-tagged BmSiwi, MIWI or MILI were transfected into BmN4 cells with X-tremeGENE HP DNA Transfection Reagent (Sigma). Preparation of loading lysates and single-stranded RNA loading were as described[71]. After loading with synthetic guide RNAs in the cell lysate, piRISC was immunoprecipitated with anti-Flag antibody (Sigma) conjugated to Dynabeads protein G superparamagnetic beads (Thermo Fisher Scientific). The beads were washed 5 times with lysis buffer containing 30 mM HEPES-KOH, pH 7.4, 100 mM potassium acetate, 2 mM magnesium acetate, 0.5 mM DTT, and 0.1% v/v Empigen. Target-cleavage assays were performed at 25 °C for 3.5 h in 8 μl reaction containing 2.4 μl '40×' reaction mix[69], 100 nM recombinant GTSF1 protein and 0.5 nM 32P cap-radiolabelled target RNA as described[72].

### BmAgo3 target-cleavage assays

Naive BmN4 cells were resuspended in lysis buffer (30 mM HEPES-KOH, pH 7.4, 100 mM potassium acetate, 2 mM magnesium acetate, 0.05% v/v Triton X-100, 0.5 mM DTT, 1× Complete EDTA-free protease inhibitor (Roche)) and lysed using a Dounce homogenizer. After centrifugation at 17,000g for 20 min at 4 °C, the supernatant was incubated at 4 °C for 1 h with anti-BmAgo3 antibody as described[73] conjugated to Dynabeads protein G (Thermo Fisher Scientific). The superparamagnetic beads were washed 5 times with wash buffer (30 mM HEPES-KOH, pH 7.4, 100 mM potassium acetate, 2 mM magnesium acetate, 0.5% v/v Empigen, 0.5% v/v Triton X-100, 0.5 mM DTT) and rinsed with lysis buffer. Target-cleavage assays were performed at 25 °C for 2.5 h in 8 μl reactions containing 2.4 μl 40× reaction mix, 55 nM recombinant protein, and 0.5 nM 32P cap-radiolabelled target 1 (complementary to an

endogenous *pi6* piRNA) or target 2 (complementary to an endogenous *pi9* piRNA), prepared by in vitro transcription as described[74].

### EfPiwi target cleavage assays

EfPiwi piRISC was purified as described[35]. Purified EfPiwi piRISC (100 nM final concentration) was incubated at 37 °C with a 5′-32P-radiolabelled target RNA complementary to g2–g21 (10 nM final concentration) in reaction buffer composed of 20 mM Tris, pH 8.0, 100 mM NaCl, 2 mM MgCl$_2$, 2 mM MnCl$_2$, and 0.5 mM TCEP. Where indicated, purified *E. muelleri* GTSF1 (500 nM final concentration) or 0.1 mg ml$^{-1}$ baker's yeast tRNA were also included. Target cleavage was stopped at indicated times by mixing aliquots of each reaction with an equal volume of denaturing gel loading buffer (98% w/v formamide, 0.025% w/v xylene cyanol, 0.025% w/v bromophenol blue, 10 mM EDTA, pH 8.0). Intact and cleaved target RNAs were resolved by denaturing PAGE (15%) and visualized by phosphorimaging. Quantification of signal was performed using ImageQuant TL (GE Healthcare).

### Mouse germ cell purification

Germ cells from mouse testes were sorted and purified as described[19,20]. In brief, freshly dissected mouse testes were decapsulated with 0.4 mg ml$^{-1}$ collagenase type IV (Worthington) in 1× Gey's balanced salt solution (GBSS) at 33 °C for 15 min. The separated seminiferous tubules were treated with 0.5 mg ml$^{-1}$ trypsin and 1 μg ml$^{-1}$ DNase I in 1× GBSS at 33 °C for 15 min. Trypsin was then inactivated by adding 7.5% v/v fetal bovine serum (FBS). The cell suspension was filtered through a 70-μm cell strainer, and cells pelleted at 300g at 4 °C for 10 min. Cell staining was performed at 33 °C for 15 min with 5 μg ml$^{-1}$ Hoechst 33342 prepared in 1× GBSS, 5% v/v FBS, and 1 μg ml$^{-1}$ DNase I, then the cells were treated with 0.2 μg ml$^{-1}$ propidium iodide, followed by final pass through a 40-μm cell strainer before sorting. Sorted cells were pelleted at 100g for 5 min, the buffer was removed, and the cell pellets were flash frozen and stored at −80 °C. Cell lysates were prepared as described for HEK293T cells. Protein concentration was estimated using the BCA assay, and an equal amount of total protein from each cell type was used to assay for the ability to potentiate MIWI catalysis.

### Analysis of RNA-seq data

Publicly available datasets[17,75–77] were analysed. rRNA reads were removed using Bowtie 2.2.5 with default parameters[78]. After rRNA removal, the remaining reads were mapped to corresponding genomes (mouse, mm10; rat, rn6; macaque, rheMac8; human, hg19) using STAR 2.3 with default parameters that allowed ≤ 2 mismatches and 100 mapping locations[79]. Mapped results were generated in SAM format, duplicates removed and translated to BAM format using SAMtools 1.8[80]. HTSeq 0.9.1 with default parameters was used to count uniquely mapping reads[81] (steady-state transcript abundance was reported in reads per kilobase per million uniquely mapped reads (RPKM)).

### Methylarginine analysis

Recombinant MIWI was immunopurified and resolved by electrophoresis on a 4–20% gradient SDS-polyacrylamide gel. The gel was fixed, stained with Coomassie G-250 (Simply Blue, Invitrogen), and the recombinant MIWI band excised and analysed at the UMASS Mass Spectrometry Core. Gel slices were chopped into ~1 mm$^2$ pieces, 1 ml water added, followed by 20 μl 45 mM DTT in 250 mM ammonium bicarbonate. Samples were incubated at 50 °C for 30 min, cooled to room temperature, then 20 μl 100 mM iodoacetamide was added and incubated for 30 min. The solution was removed, and the gel pieces were three times washed with 1 ml water, 1 ml 50 mM ammonium bicarbonate:acetonitrile (1:1), quenched with 200 μl acetonitrile, and dried in a SpeedVac. Gel pieces were then rehydrated in 50 μl 50 mM ammonium bicarbonate containing 4 ng μl$^{-1}$ trypsin (Promega, Madison, WI) and 0.01% proteaseMAX (Promega) and incubated at 37 °C for 18 h. Supernatants were collected and extracted with 200 μl an

80:20 solution of acetonitrile: 1% v/v formic acid in water. Supernatants were then combined, and the peptides lyophilized in a SpeedVac and resuspended in 25 μl 5% acetonitrile, 0.1% v/v formic acid and subject to mass spectrometry analysis. Data were acquired using a NanoAcquity UPLC (Waters Corporation) coupled to an Orbitrap Fusion Lumos Tribrid (Thermo Fisher Scientific) mass spectrometer. Peptides were trapped and separated using an in-house 100 μm internal diameter fused-silica pre-column (Kasil frit) packed with 2 cm ProntoSil (Bischoff Chromatography) C18 AQ (200 Å, 5 μm) media and configured to an in-house packed 75 μm internal diameter fused-silica analytical column (gravity-pulled tip) packed with 25 cm ProntoSil (Bischoff; 100 Å, 3 μm) media. Mobile phase A was 0.1% v/v formic acid in water; mobile phase B was 0.1% v/v formic acid in acetonitrile. Following a 3.8 μl sample injection, peptides were trapped at flow rate of 4 μl/min with 5% B for 4 min, followed by gradient elution at a flow rate of 300 nl min$^{-1}$ from 5–35% B in 90 min (total run time, 120 min). Electrospray voltage was delivered by liquid junction electrode (1.5 kV) located between the columns and the transfer capillary to the mass spectrometer was maintained at 275 °C. Mass spectra were acquired over $m/z$ 300–1,750 Da with a resolution of 120,000 ($m/z$ 200), maximum injection time of 50 ms, and an AGC target of 400,000. Tandem mass spectra were acquired using data-dependent acquisition (3 s cycle) with an isolation width of 1.6 Da, HCD collision energy of 30%, resolution of 15,000 ($m/z$ 200), maximum injection time of 22 ms, and an AGC target of 50,000.

Raw data were processed using Proteome Discoverer 2.1.1.21 (Thermo Fisher Scientific), and the database search performed by Mascot 2.6.2 (Matrix Science) using the Swiss-Prot human database (downloaded 4 September 2019; https://www.uniprot.org/). Search parameters were: semi-tryptic digestion with up to two missed cleavages; precursor mass tolerance 10 ppm; fragment mass tolerance 0.05 Da; peptide N-terminal acetylation, cysteine carbamidomethylation, methionine oxidation, N-terminal glutamine to pyroglutamate conversion, arginine methylation and arginine demethylation were specified as variable modifications. Peptide and protein validation and annotation was done in Scaffold 4.8.9 (Proteome Software, Portland, OR) employing Peptide Prophet[82] and Protein Prophet[83]. Peptides were filtered at 1% FDR, while protein identification threshold was set to greater than 99% probability and with a minimum of two identified peptides per protein. Only arginine modification sites detected in all three replicates from separate immunoprecipitation experiments are reported in the figure.

## Statistics and reproducibility
Figures 1b, 3, 4b,c and 5a and Extended Data Fig. 4a show mean ± s.d. for three independent trials. Figure 2a,b, Extended Data Figs. 1a–c,h and 6a show three independent trials with similar results. Figure 2e,f and Extended Data Figs. 1f,g, 2b, 3a,c and 5 show two independent trials with similar results. The experiments in Figs. 1c and 4a and Extended Data Figs. 1d,e, 2a, 3b,d–h, 4b, 6b and 8a,b were performed once. Protein sequences were aligned using Clustal Omega (1.2.4); unrooted tree was constructed using randomized axelerated maximum likelihood (RAxML 1.0.0) with default parameters[84] and visualized in Interactive Tree of Life[85].

## Reporting summary
Further information on research design is available in the Nature Research Reporting Summary linked to this article.

## Data availability
All data are available from the authors upon request. Mass spectrometry data are available from MassIVE using accession number MSV000089490.

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

**Acknowledgements** We thank K.-E. Aryee for providing pScalps_Puro, psPAX2, pMD2.G plasmids and for guidance on lentiviral transduction; P.-H. Wu for help with mouse testis germ cell FACS and for providing sorted cells; K. Cecchini for mouse tissues; E. Norowski for rat testes; K. Orwig for rhesus macaque testes; P. Albosta and C. Tipping for guidance with *T. ni* and *D. melanogaster* dissection; L. Joshua-Tor for sharing data prior to publication; T. Gardner and K. Jouravleva for help formatting and proofreading the manuscript; the UMass Chan Flow Cytometry and Mass Spectrometry cores; and members of the Zamore laboratory for discussions and comments on the manuscript. This work was supported in part by NIGMS grants R37 GM062862 and R35 GM136275 (P.D.Z.) and R35 GM127090 (I.J.M.) and JSPS KAKENHI grants 18H05271 (Y.T.) and 19K06484 (N.I.).

**Author contributions** A.A., P.D.Z., I.J.M. and Y.T. conceived and designed the experiments. A.A., S.B., N.I. (insect proteins), T.A.A. (sponge proteins), C.A. (bacterial culture) and I.G. (endogenous piRNAs) performed the experiments. D.M.O. and I.G. provided and analysed the sequencing data. P.D.Z. supervised the research. A.A. and P.D.Z. prepared the manuscript. All authors discussed the results and approved the manuscript.

**Competing interests** The authors declare no competing interests.

**Additional information**
**Correspondence and requests for materials** should be addressed to Phillip D. Zamore.

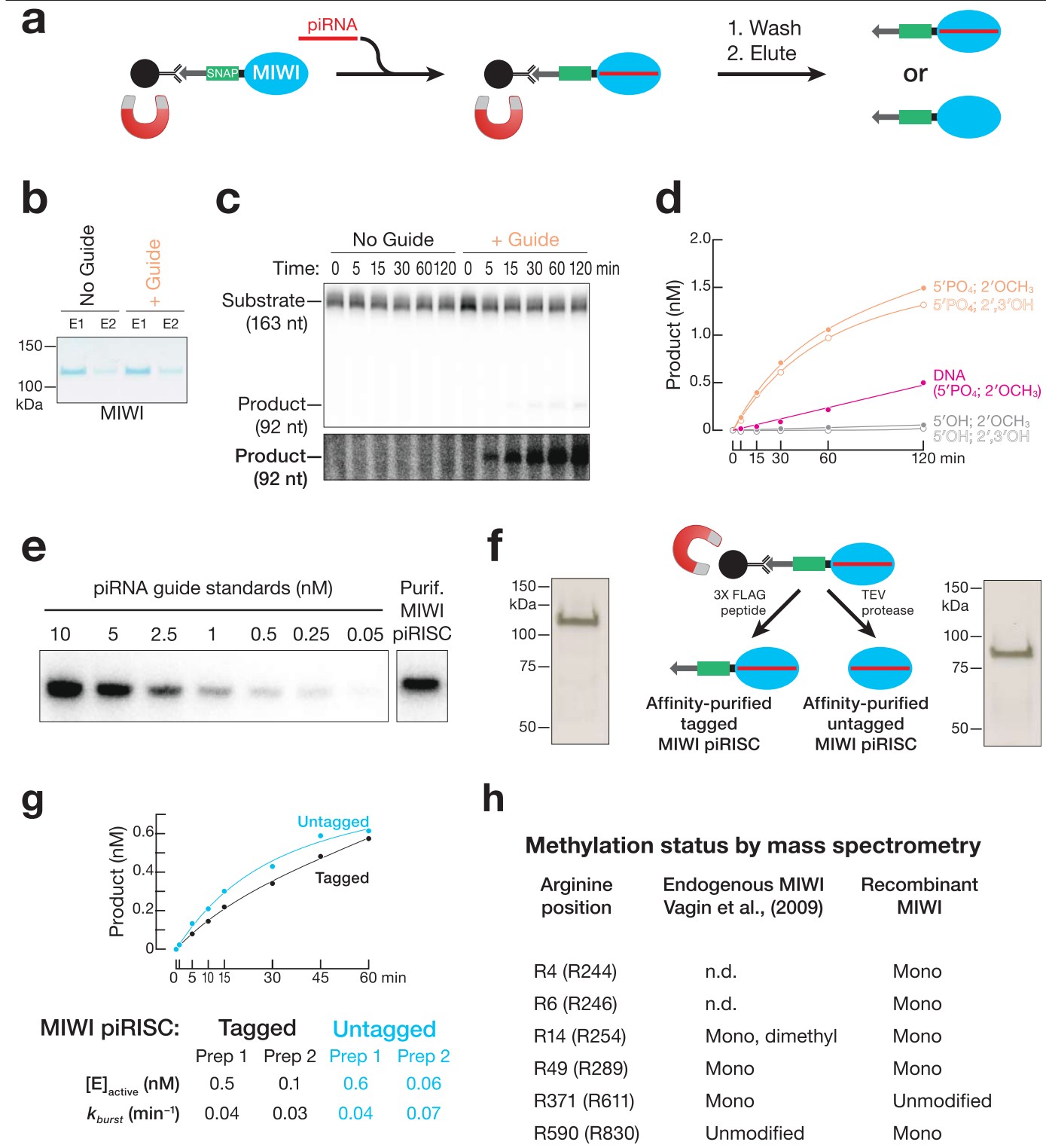

**a**

**b**

**c**

**d**

**e**

**f**

**g**

**h**

### Methylation status by mass spectrometry

| Arginine position | Endogenous MIWI Vagin et al., (2009) | Recombinant MIWI |
|---|---|---|
| R4 (R244) | n.d. | Mono |
| R6 (R246) | n.d. | Mono |
| R14 (R254) | Mono, dimethyl | Mono |
| R49 (R289) | Mono | Mono |
| R371 (R611) | Mono | Unmodified |
| R590 (R830) | Unmodified | Mono |

MIWI piRISC:

| | Tagged | | Untagged | |
|---|---|---|---|---|
| | Prep 1 | Prep 2 | Prep 1 | Prep 2 |
| $[E]_{active}$ (nM) | 0.5 | 0.1 | 0.6 | 0.06 |
| $k_{burst}$ (min$^{-1}$) | 0.04 | 0.03 | 0.04 | 0.07 |

**Extended Data Fig. 1 | Recombinant MIWI programmed with guide piRNA cleaves complementary target RNA. a**, Purification of apo-MIWI and of MIWI loaded with a synthetic piRNA guide (piRISC). **b**, Coomassie-stained SDS-PAGE of purified apo-MIWI and MIWI piRISC. E1, E2: first and second eluates from sequential incubation of the paramagnetic beads with 3XFLAG peptide. E1 was used for the cleavage assay. **c**, Representative denaturing polyacrylamide gel of target cleavage assay using apo-MIWI or MIWI piRISC. A digital overexposure is shown below the gel. **d**, MIWI piRISC loaded with synthetic guides differing at their 5′ or 3′ termini was assayed for the ability to cleave a complementary target RNA. **e**, Northern blot to estimate the yield of purified MIWI piRISC. **f**, Silver-stained gel showing purified MIWI piRISC with and without an

amino-terminal epitope tag. **g**, Target cleavage assay comparing the activity of piRISC with and without an amino-terminal epitope tag. Data were fit to Equation 1 (see Methods); error of fit (SD) is reported. **h**, Mass spectrometry was used to determine arginine methylation status for recombinant, affinity-purified MIWI ($n$ = 3); arginine methylation status of endogenous MIWI from mouse testis is provided for comparison[76]. Arginine position reports the amino acid number in endogenous mouse MIWI and in parentheses, the corresponding residue in the epitope-tagged recombinant protein. Only modified arginine residues detected in all three independent preparations are reported. N.D., not detected. For gel source data, see Supplementary Fig. 1.

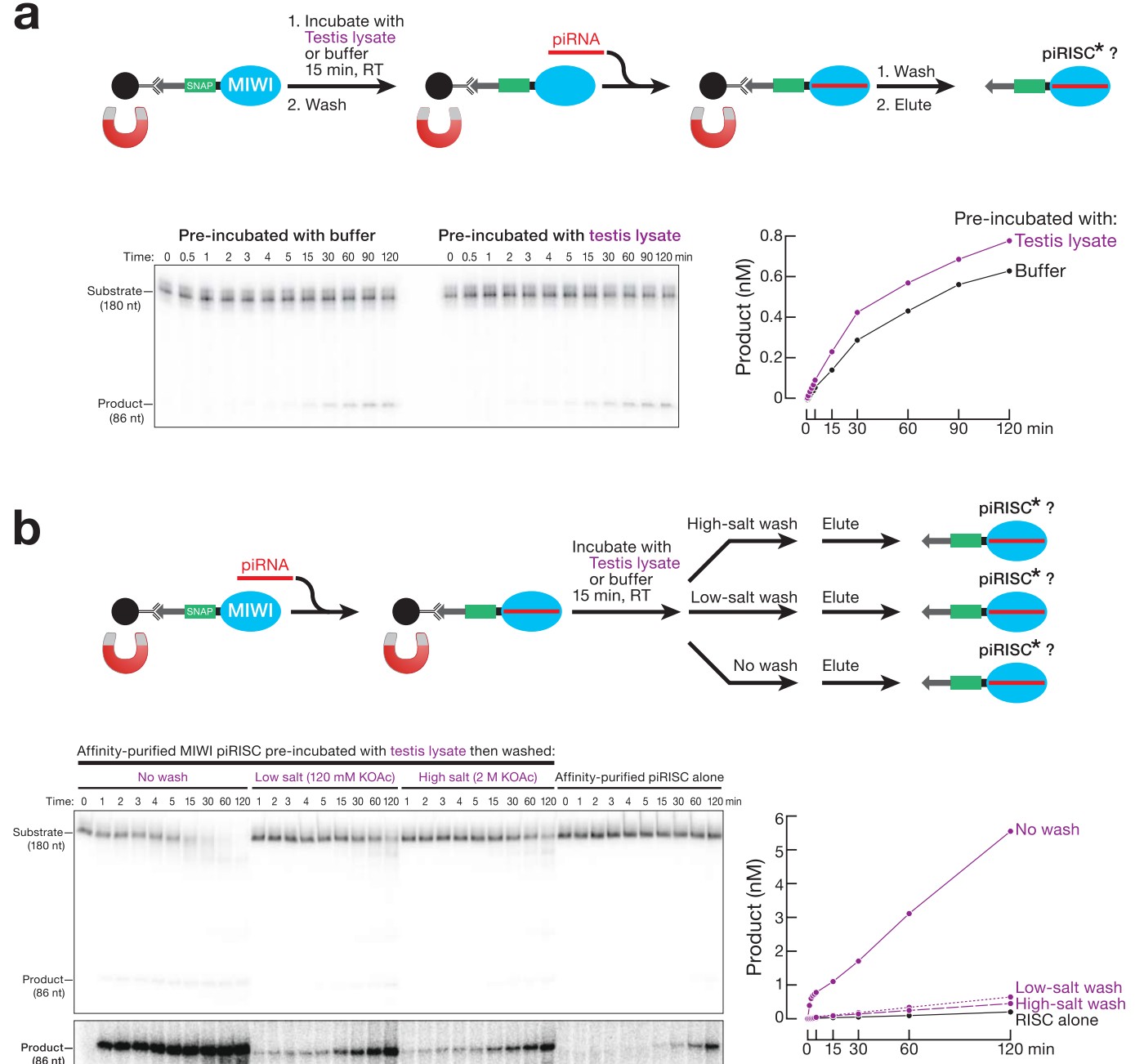

**Extended Data Fig. 2 | The target-cleavage potentiating component in testis lysate component transiently interacts with MIWI. a**, Immobilized, unloaded apo-MIWI was preincubated with testis lysate, purified as depicted, and then assayed for target cleavage activity. **b**, Immobilized MIWI piRISC was preincubated with testis lysate. purified as depicted, and then assayed for target cleavage activity. A digital overexposure is shown below the gel. For gel source data, see Supplementary Fig. 1.

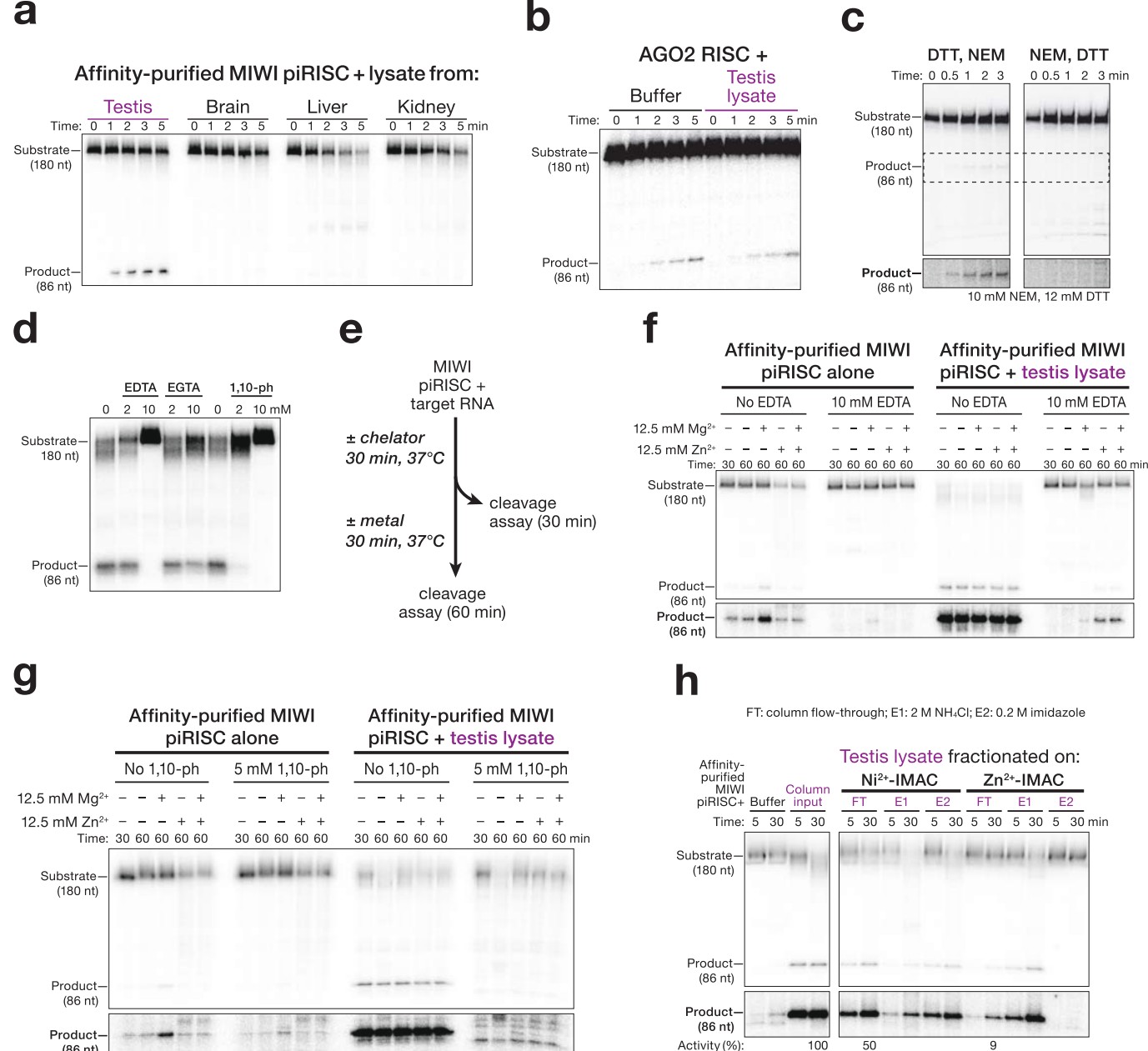

**Extended Data Fig. 3 | Biochemical properties of the MIWI potentiating component in testis lysate. a**, The MIWI-potentiating factor is specific to testis lysate. **b**, Testis lysate does not enhance target cleavage by purified mouse AGO2. **c**, The MIWI-potentiating factor is sensitive to alkylation by *N*-ethylmaleimide. **d**, Effect of divalent cation chelators on MIWI-potentiating activity. For 0 nM chelator, water was added for EDTA and EGTA and ethanol for 1,10-phenanthroline. **e**, Strategy to test metal rescue after incubation of testis lysate with divalent cation chelators. **f, g**, Metal rescue experiments were performed as in (**e**) using EDTA (**f**) or 1,10-phenanthroline (**g**). **h**, Fractionation of testis lysate using Zn²⁺ and Ni²⁺ immobilized metal affinity chromatography (IMAC). In **c** and **f**–**h**, a digital overexposure is shown below each gel. Numbers below the gel indicate relative product produced in each lane. For gel source data, see Supplementary Fig. 1.

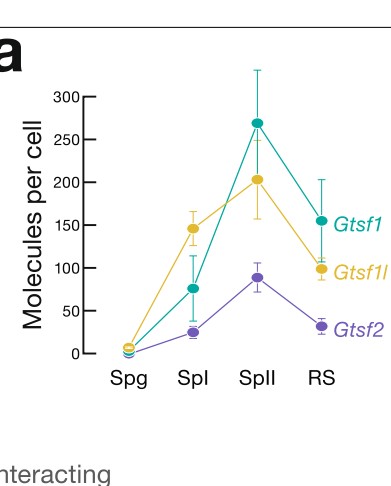

**a**

**b**

Wild-type GTSF1 — RNA-binding mutant GTSF1 — PIWI-interacting mutant GTSF1 — GTSF1L — GTSF2 — Mock / MmGTSF1 / BmGtsf1 / BmGtsf1l (Oriole stain) — EmGtsf1 (Coomassie)

RNA-binding — CHHC ZnF1 — CHHC ZnF2 — PIWI-interacting — GTSF1

| Category | Residue | Threshold, Residue group |
|---|---|---|
| Basic | K,R | >60%, KR; >80%, K,R,Q |
| Acidic | E | >60%, KR; >50%, QE; >85%, E,Q,D |
| Acidic | D | >60%, KR; >85%, K,R,Q; >50%, ED |
| Polar | N | >50%, N; >85%, N,Y |
| Polar | Q | >60%, KR; >50%, QE; >85%, Q,E,K,R |
| Polar | S,T | >60%, WLVIMAFCHP; >50%, TS; >85%, S,T |
| Unconserved | any/gap | none of the above criteria met |

| Category | Residue | Threshold, Residue group |
|---|---|---|
| Hydrophobic | A,I,L,M,F,W,V | >60%, WLVIMAFCHP |
| Hydrophobic | C | >60%, WLVIMAFCHP |
| Cysteine | C | >85%, C |
| Glycine | G | >0%, G |
| Proline | P | >0%, P |
| Aromatic | H,Y | >60%, WLVIMAFCHP; >85%, W,Y,A,C,P,Q,F,H,I,L,M,V |

**c**

| | | GTSF1 | GTSF1 PIWI-interaction mutant | GTSF1 RNA-binding mutant | GTSF1L | GTSF2 | Buffer |
|---|---|---|---|---|---|---|---|
| **MIWI** | $k_{burst}$ (min$^{-1}$) | 1.1 ± 0.1 | 0.67 ± 0.08 | < LQ | 1.1 ± 0.1 | 1.1 ± 0.1 | < LQ |
| | $k_{rel}$ | 1 | 0.6 | — | 1 | 1 | — |
| **MILI** | $k_{burst}$ (min$^{-1}$) | 0.17 ± 0.03 | 0.04 ± 0.01 | < LQ | 0.02 ± 0.02 | < LQ | < LQ |
| | $k_{rel}$ | 1 | 0.2 | — | 0.06 | — | — |

**Extended Data Fig. 4 | *Gtsf1*, *Gtsf1l*, and *Gtsf2* mRNA abundance and recombinant protein expression. a**, For each germ cell type, mRNA abundance (from publicly available data; see Methods) is reported as mean ± SD (*n* = 3) in molecules/cell. Spg: spermatogonia; SpI: primary spermatocytes; SpII: secondary spermatocytes; RS: round spermatids. **b**, Right: Coomassie-stained SDS-PAGE of purified, recombinant wild-type GTSF1, GTSF1 mutants, and wild-type GTSF1Like and GTSF2. Left: Oriole protein stain detection of purified, recombinant, C-terminal V5SBP-tagged GTSF1 orthologs. Mm: *Mus musculus* (mouse); Bm: *Bombyx mori* (silkmoth). Coomassie protein stain detection of purified, recombinant *Ephydatia muelleri* Gtsf1. Bottom: Clustal Omega amino acid sequence alignment of the three mouse GTSF1 paralogs. **c**, Pre-steady-state rates ($k_{burst}$) of target cleavage by MIWI and MILI determined by fitting the data in Fig. 3d–f (*n* = 3) to the burst-and-steady-state equation. LQ: limit of quantification. For gel source data, see Supplementary Fig. 1.

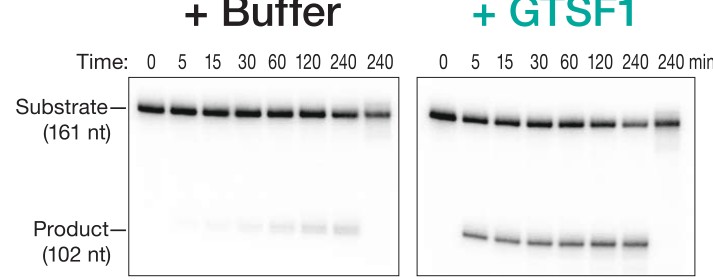
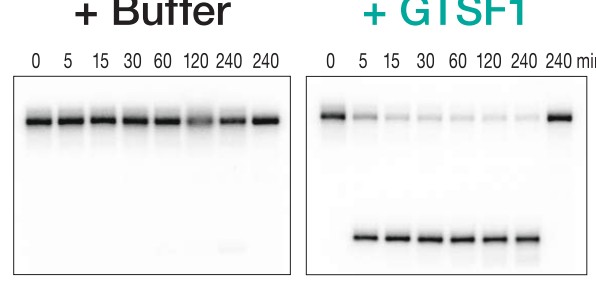

## *pi9* piRNA, *L1MC* target *pi6* piRNA, *Scpep1* target

**Extended Data Fig. 5 | Efficient target cleavage by MIWI guided by endogenous piRNAs requires GTSF1.** Representative target cleavage assay using MIWI piRISC programmed with an abundant mouse pachytene piRNA antisense to the L1MC transposon (left) or targeting the *Scpep1* mRNA (right). piRNA sequences are in orange; below each piRNA sequence, the target in the RNA sequence is in black. Scissors indicates the cleavage site of the target RNA.

The last lane in each gel shows the extent of cleavage using the reciprocal, non-cognate target, incubated with piRISC for 240 min as a negative control. Where data points from the two trials overlap, jitter has been introduced for clarity. Curves show the fit of the burst-and-steady equation to the mean of the two trials. For gel source data, see Supplementary Fig. 1.

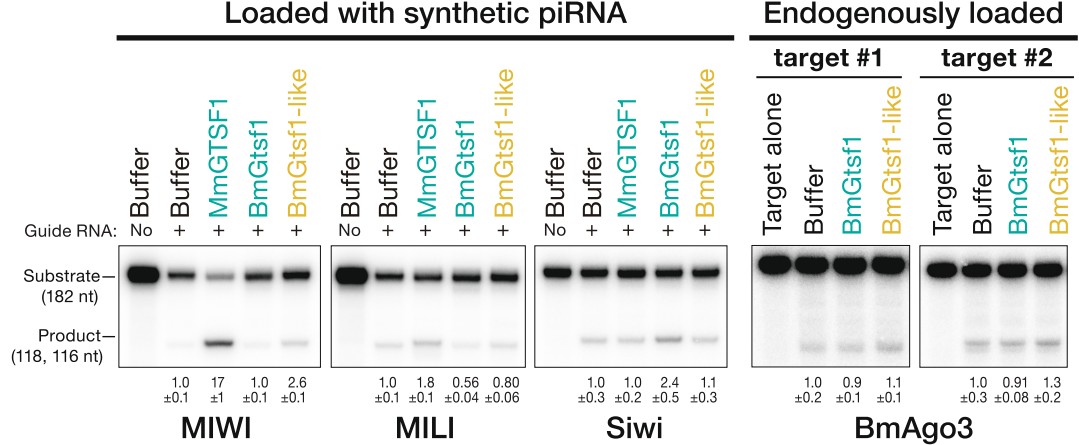

## a

**Loaded with synthetic piRNA**

**Endogenously loaded**

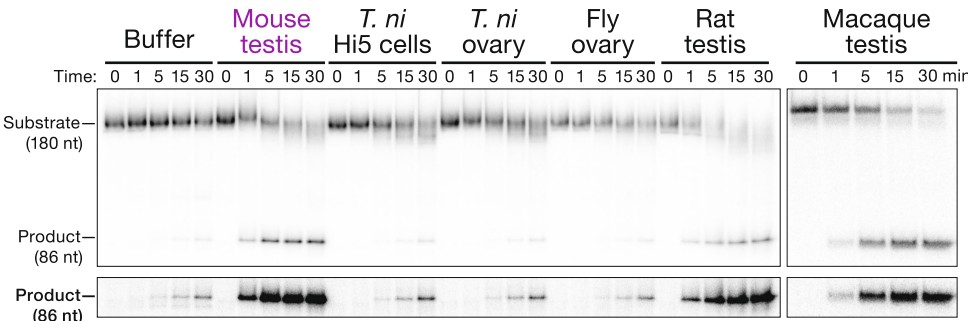

## b

**Affinity-purified MIWI piRISC plus lysate from:**

## c

| | % Identity (GTSF1) | |
|---|---|---|
| *Drosophila melanogaster* (Dm) | 100.00 | 19.86 |
| *Bombyx mori* (Bm) | 20.00 | 22.38 |
| *Trichoplusia ni* (Tn) | 19.40 | 21.17 |
| *Mus musculus* (Mm) | 19.86 | 100.00 |
| *Rattus norvegicus* | 19.86 | 100.00 |
| *Macaca mulatta* (Mmul) | 20.55 | 91.62 |
| *Homo sapiens* (Hs) | 20.55 | 91.62 |
| *Ephydatia muelleri* (Em) | 14.97 | 21.47 |

**Extended Data Fig. 6 | GTSF proteins are PIWI selective. a**, MIWI piRISC target cleavage assay in the presence of lysates from different animals. *T.ni: Trichoplusia ni*. Macaque: rhesus macaque. A digital overexposure is shown below the gel. **b**, Representative target cleavage assay using the indicated PIWI protein in the presence of different GTSF1 orthologs (100 nM). Numbers below the gel lanes report fraction target cleaved (mean ± SD; *n* = 3). **c**, Left: Percent identity of different GTSF1 orthologs Right: Unrooted tree of GTSF1 orthologs. For gel source data, see Supplementary Fig. 1.

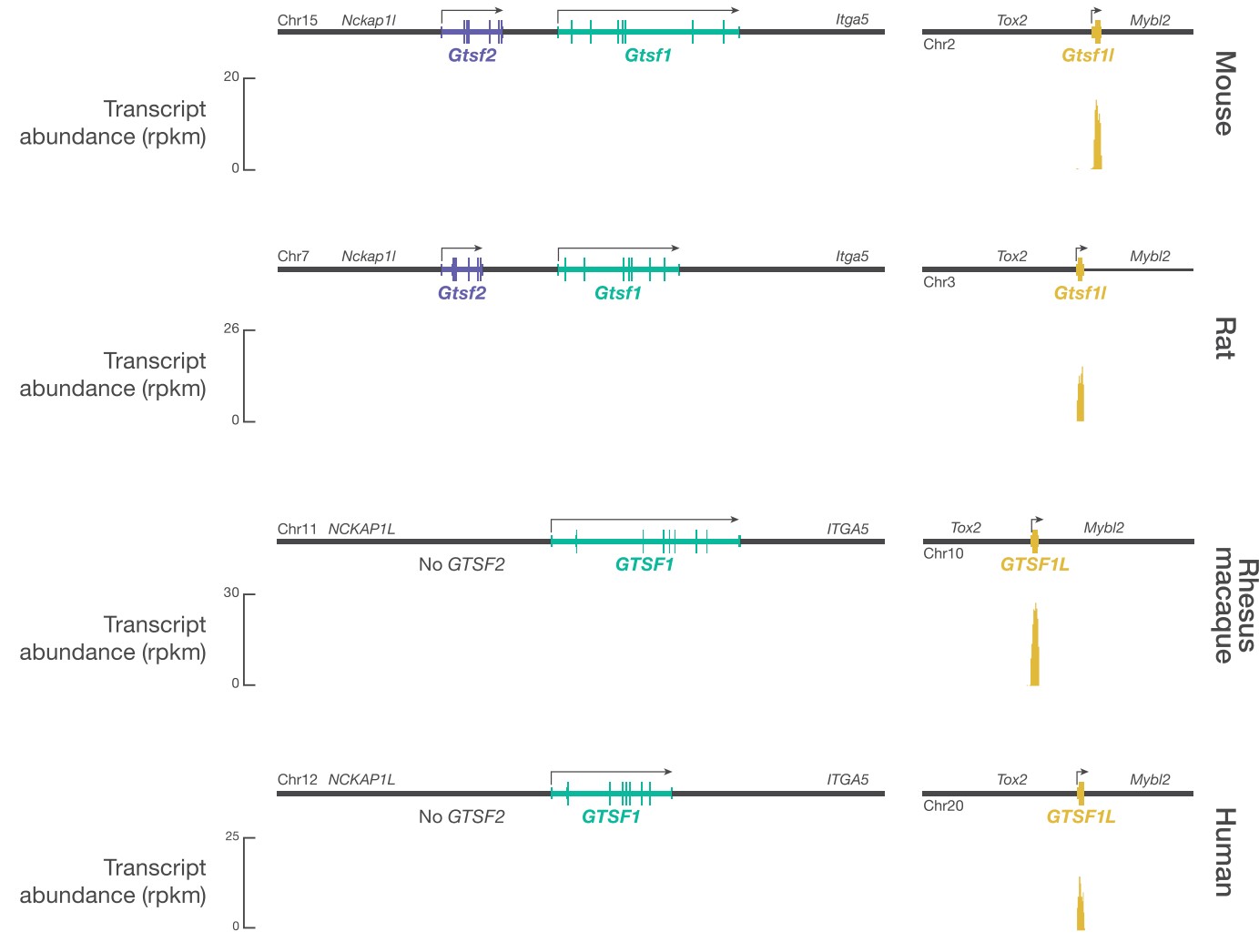

**Extended Data Fig. 7 | Expression of GTSF1 paralogs in mouse, rat, rhesus macaque, and human whole testes.** *Gtsf* genes are syntenic in mice, rats, macaques, and humans. Primates, including humans, do not express GTSF2.

RNA-seq from total testis RNA of the indicated animals sourced from previously published datasets (see Methods).

**a**

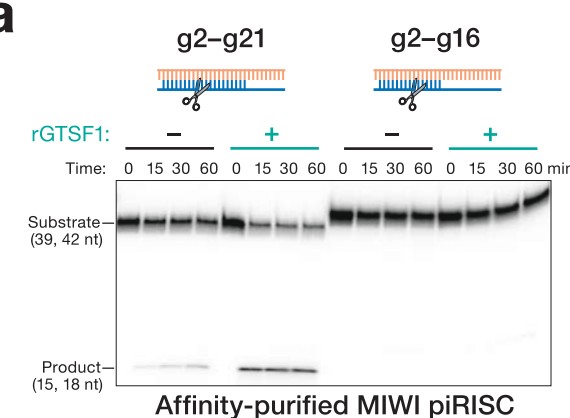

Affinity-purified MIWI piRISC

**b**

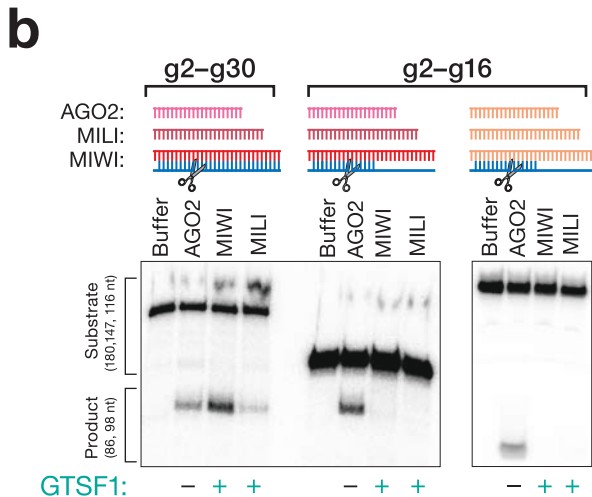

**Extended Data Fig. 8 | Complementarity and GTSF1 requirements for target cleavage by AGO2, MIWI, and MILI. a**, MIWI target cleavage ± GTSF1 using synthetic piRNA guide 1. **b**, Target cleavage by AGO2 siRISC compared with MIWI or MILI piRISC loaded with synthetic piRNA guide 1 or 2, for targets with complete (g2–g30) or partial (g2–g16) complementarity to the piRNA. For gel source data, see Supplementary Fig. 1.

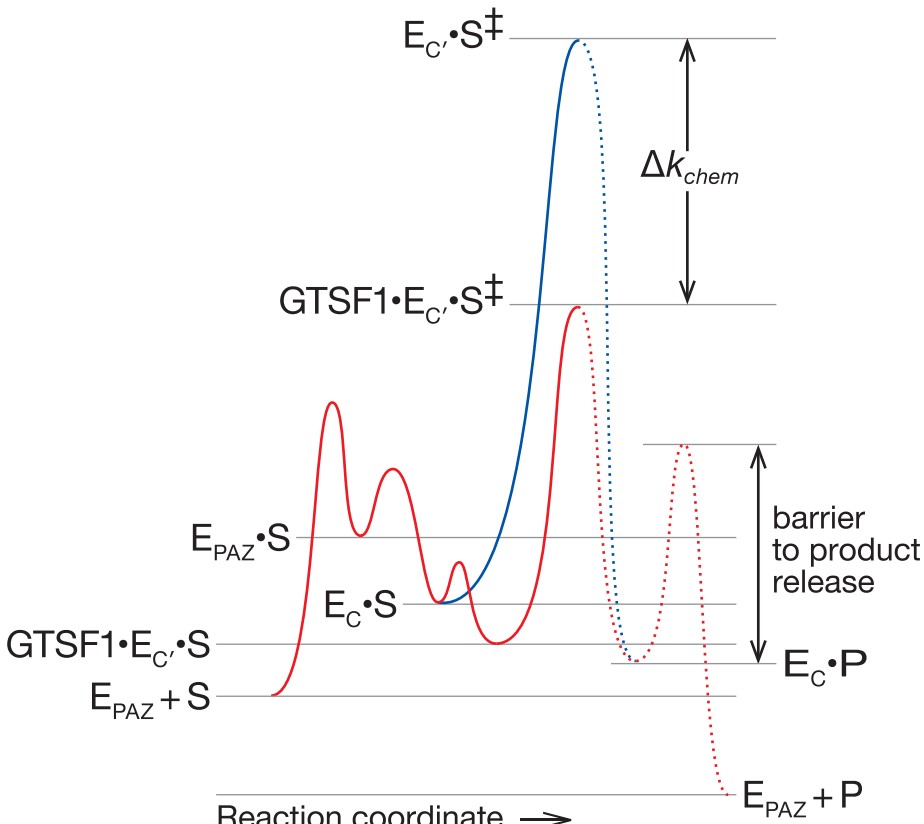

**Extended Data Fig. 9 | Model of the free-energy landscape of GTSF1-induced potentiation of target cleavage by PIWI.** Two conformational changes in piRISC (E) are proposed to be required for efficient target cleavage catalyzed by PIWI proteins. $E_{PAZ}$: piRISC with the 3′ end of the piRNA bound to the PAZ domain; $E_C$: piRISC with the piRNA fully paired to the target RNA, i.e., the pre-catalytic conformation; $E_{C′}$: $E_C$ in the catalytically competent conformation; $S^{‡}$: transition-state. For piRNAs of biologically relevant length, extensive complementarity is proposed to promote the first conformational change; GTSF1 is proposed to promote the second. Blue: energy barrier to catalysis without GTSF1, i.e., the spontaneous conversion of $E_C$ to $E_{C′}$.

**Extended Data Table 1 | Pairwise comparison of percent identity (blue) or similarity (orange) for domain sequence of GTSF1-related proteins**

| | | | Mus musculus (Mm) | | | Rattus norvegicus (Rn) | | | Macaca mulatta (Mmul) | | Trichoplusia ni (Tn) | | Bombxy mori (Bm) | | Drosophila melanogaster (Dm) |
|---|---|---|---|---|---|---|---|---|---|---|---|---|---|---|---|
| | | | GTSF1 | GTSF1L | GTSF2 | GTSF1 | GTSF1L | GTSF2 | GTSF1 | GTSF1L | TnGtsf1 | TnGtsf1l | BmGtsf1 | BmGtsf1l | Asterix |
| **Zinc finger 1** | Mm | GTSF1 | 100 | 50 | 54.2 | 100 | 50 | 54.2 | 100 | 54.2 | 45.8 | 54.2 | 45.8 | 33.3 | 37.5 |
| | | GTSF1L | 62.5 | 100 | 73.9 | 50 | 100 | 73.9 | 50 | 95.7 | 41.7 | 52.2 | 37.5 | 56.5 | 39.1 |
| | | GTSF2 | 66.7 | 87 | 100 | 54.2 | 73.9 | 100 | 54.2 | 78.3 | 33.3 | 43.5 | 33.3 | 56.5 | 34.8 |
| | Rn | GTSF1 | 100 | 62.5 | 66.7 | 100 | 50 | 54.2 | 100 | 54.2 | 45.8 | 54.2 | 45.8 | 33.3 | 37.5 |
| | | GTSF1L | 62.5 | 100 | 87 | 62.5 | 100 | 73.9 | 50 | 95.7 | 41.7 | 52.2 | 37.5 | 56.5 | 39.1 |
| | | GTSF2 | 66.7 | 87 | 100 | 66.7 | 87 | 100 | 54.2 | 78.3 | 33.3 | 43.5 | 33.3 | 56.5 | 34.8 |
| | Mmul | GTSF1 | 100 | 62.5 | 66.7 | 100 | 62.5 | 66.7 | 100 | 54.2 | 45.8 | 54.2 | 45.8 | 33.3 | 37.5 |
| | | GTSF1L | 66.7 | 95.7 | 91.3 | 66.7 | 95.7 | 91.3 | 66.7 | 100 | 37.5 | 47.8 | 37.5 | 52.2 | 34.8 |
| | Tn | Gtsf1 | 54.2 | 54.2 | 54.2 | 54.2 | 54.2 | 54.2 | 54.2 | 50 | 100 | 41.7 | 83.3 | 45.8 | 37.5 |
| | | Gtsf1-like | 66.7 | 65.2 | 52.2 | 66.7 | 65.2 | 52.2 | 66.7 | 60.9 | 62.5 | 100 | 41.7 | 43.5 | 39.1 |
| | Bm | Gtsf1 | 58.3 | 45.8 | 54.2 | 58.3 | 45.8 | 54.2 | 58.3 | 50 | 87.5 | 54.2 | 100 | 37.5 | 37.5 |
| | | Gtsf1-like | 54.2 | 65.2 | 65.2 | 54.2 | 65.2 | 65.2 | 54.2 | 60.9 | 66.7 | 73.9 | 58.3 | 100 | 52.2 |
| | Dm | Asterix | 58.3 | 56.5 | 56.5 | 58.3 | 56.5 | 56.5 | 58.3 | 52.2 | 66.7 | 69.6 | 62.5 | 73.9 | 100 |
| **Zinc finger 2** | Mm | GTSF1 | 100 | 39.1 | 39.1 | 100 | 39.1 | 39.1 | 100 | 39.1 | 43.5 | 43.5 | 39.1 | 30.4 | 39.1 |
| | | GTSF1L | 65.2 | 100 | 78.3 | 39.1 | 100 | 78.3 | 39.1 | 82.6 | 39.1 | 30.4 | 34.8 | 30.4 | 26.1 |
| | | GTSF2 | 56.5 | 91.3 | 100 | 39.1 | 78.3 | 100 | 39.1 | 82.6 | 34.8 | 39.1 | 34.8 | 39.1 | 26.1 |
| | Rn | GTSF1 | 100 | 65.2 | 56.5 | 100 | 39.1 | 39.1 | 100 | 39.1 | 43.5 | 43.5 | 39.1 | 30.4 | 39.1 |
| | | GTSF1L | 65.2 | 100 | 91.3 | 65.2 | 100 | 78.3 | 39.1 | 82.6 | 39.1 | 30.4 | 34.8 | 30.4 | 26.1 |
| | | GTSF2 | 56.5 | 91.3 | 100 | 56.5 | 91.3 | 100 | 39.1 | 82.6 | 34.8 | 39.1 | 34.8 | 39.1 | 26.1 |
| | Mmul | GTSF1 | 100 | 65.2 | 56.5 | 100 | 65.2 | 56.5 | 100 | 39.1 | 43.5 | 43.5 | 39.1 | 30.4 | 39.1 |
| | | GTSF1L | 52.2 | 87 | 87 | 52.2 | 87 | 87 | 52.2 | 100 | 34.8 | 39.1 | 39.1 | 30.4 | 26.1 |
| | Tn | Gtsf1 | 56.5 | 52.2 | 52.2 | 56.5 | 52.2 | 52.2 | 56.5 | 52.2 | 100 | 43.5 | 82.6 | 34.8 | 26.1 |
| | | Gtsf1-like | 52.2 | 47.8 | 56.5 | 52.2 | 47.8 | 56.5 | 52.2 | 52.2 | 52.2 | 100 | 43.5 | 60.9 | 30.4 |
| | Bm | Gtsf1 | 52.2 | 47.8 | 52.2 | 52.2 | 47.8 | 52.2 | 52.2 | 56.5 | 91.3 | 52.2 | 100 | 39.1 | 26.1 |
| | | Gtsf1-like | 60.9 | 47.8 | 47.8 | 60.9 | 47.8 | 47.8 | 60.9 | 39.1 | 65.2 | 82.6 | 60.9 | 100 | 26.1 |
| | Dm | Asterix | 56.5 | 60.9 | 56.5 | 56.5 | 60.9 | 56.5 | 56.5 | 60.9 | 60.9 | 65.2 | 56.5 | 47.8 | 100 |
| **Carboxy terminal domain** | Mm | GTSF1 | 100 | 7.3 | 8.3 | 100 | 7.3 | 10.4 | 88.5 | 7.3 | 7.3 | 6.2 | 8.3 | 6.2 | 8 |
| | | GTSF1L | 20.8 | 100 | 32.6 | 7.3 | 93.2 | 31.5 | 9.4 | 60 | 4.3 | 4.4 | 5.4 | 4.4 | 5.5 |
| | | GTSF2 | 28.1 | 54.3 | 100 | 8.3 | 32.6 | 89 | 9.4 | 34.8 | 2.2 | 4.4 | 4.3 | 5.5 | 10.1 |
| | Rn | GTSF1 | 100 | 20.8 | 28.1 | 100 | 7.3 | 10.4 | 88.5 | 7.3 | 7.3 | 6.2 | 8.3 | 6.2 | 8 |
| | | GTSF1L | 20.8 | 97.7 | 54.3 | 20.8 | 100 | 31.5 | 9.4 | 58.9 | 3.2 | 4.4 | 4.3 | 4.4 | 5.5 |
| | | GTSF2 | 28.1 | 53.3 | 93.4 | 28.1 | 53.3 | 100 | 11.5 | 33.7 | 2.2 | 4.4 | 4.3 | 5.5 | 9.2 |
| | Mmul | GTSF1 | 92.7 | 19.8 | 28.1 | 92.7 | 19.8 | 28.1 | 100 | 8.3 | 8.3 | 8.3 | 9.4 | 7.3 | 7.1 |
| | | GTSF1L | 24 | 66.7 | 53.3 | 24 | 68.9 | 51.1 | 24 | 100 | 6.5 | 1.1 | 9.7 | 2.2 | 4.6 |
| | Tn | Gtsf1 | 20.8 | 20.4 | 17.4 | 20.8 | 20.4 | 16.3 | 16.7 | 19.4 | 100 | 6.2 | 60 | 5 | 8.3 |
| | | Gtsf1-like | 11.5 | 8.9 | 8.8 | 11.5 | 7.8 | 8.8 | 12.5 | 6.7 | 7.5 | 100 | 5 | 44.4 | 4.6 |
| | Bm | Gtsf1 | 27.1 | 17.2 | 21.7 | 27.1 | 19.4 | 20.7 | 22.9 | 22.6 | 71.2 | 11.2 | 100 | 7.5 | 8.3 |
| | | Gtsf1-like | 11.5 | 12.2 | 14.3 | 11.5 | 11.1 | 13.2 | 12.5 | 7.8 | 6.2 | 60 | 10 | 100 | 1.8 |
| | Dm | Asterix | 17.9 | 15.6 | 18.3 | 17.9 | 14.7 | 17.4 | 18.8 | 14.7 | 18.3 | 10.1 | 17.4 | 11 | 100 |

# Reporting Summary

## Statistics

For all statistical analyses, confirm that the following items are present in the figure legend, table legend, main text, or Methods section.

| n/a | Confirmed | |
|---|---|---|
| ☐ | ☒ | The exact sample size ($n$) for each experimental group/condition, given as a discrete number and unit of measurement |
| ☒ | ☐ | A statement on whether measurements were taken from distinct samples or whether the same sample was measured repeatedly |
| ☐ | ☒ | The statistical test(s) used AND whether they are one- or two-sided *Only common tests should be described solely by name; describe more complex techniques in the Methods section.* |
| ☒ | ☐ | A description of all covariates tested |
| ☒ | ☐ | A description of any assumptions or corrections, such as tests of normality and adjustment for multiple comparisons |
| ☐ | ☒ | A full description of the statistical parameters including central tendency (e.g. means) or other basic estimates (e.g. regression coefficient) AND variation (e.g. standard deviation) or associated estimates of uncertainty (e.g. confidence intervals) |
| ☐ | ☒ | For null hypothesis testing, the test statistic (e.g. $F$, $t$, $r$) with confidence intervals, effect sizes, degrees of freedom and $P$ value noted *Give P values as exact values whenever suitable.* |
| ☒ | ☐ | For Bayesian analysis, information on the choice of priors and Markov chain Monte Carlo settings |
| ☒ | ☐ | For hierarchical and complex designs, identification of the appropriate level for tests and full reporting of outcomes |
| ☒ | ☐ | Estimates of effect sizes (e.g. Cohen's $d$, Pearson's $r$), indicating how they were calculated |

*Our web collection on statistics for biologists contains articles on many of the points above.*

## Software and code

Policy information about availability of computer code

| Data collection | Typhoon FLA 7000, NanoAcquity UPLC, Orbitrap Fusion Lumos tribrid, Li-CoR Odyssey Clx, Image Studio |
|---|---|
| Data analysis | Igor Pro 8, Microsoft Excel 16.59, Bowtie 2.2.5, STAR 2.3, SAMtools 1.8, HTSeq 0.9.1, Proteome Discoverer 2.1.1.21, Mascot 2.6.2, Scaffold 4.8.9, Image Studio Lite (5.2.5), MacVector 18.0.1, Clustal Omega 1.2.4, RAxML (1.0.0) |

For manuscripts utilizing custom algorithms or software that are central to the research but not yet described in published literature, software must be made available to editors and reviewers. We strongly encourage code deposition in a community repository (e.g. GitHub). See the Nature Portfolio guidelines for submitting code & software for further information.

## Data

Policy information about availability of data

All manuscripts must include a data availability statement. This statement should provide the following information, where applicable:
- Accession codes, unique identifiers, or web links for publicly available datasets
- A description of any restrictions on data availability
- For clinical datasets or third party data, please ensure that the statement adheres to our policy

All data are available from the authors upon request.
Swiss-Prot human database (download 04/09/2019; https://www.uniprot.org/)

March 2021

# Field-specific reporting

Please select the one below that is the best fit for your research. If you are not sure, read the appropriate sections before making your selection.

☒ Life sciences  ☐ Behavioural & social sciences  ☐ Ecological, evolutionary & environmental sciences

For a reference copy of the document with all sections, see nature.com/documents/nr-reporting-summary-flat.pdf

# Life sciences study design

All studies must disclose on these points even when the disclosure is negative.

| | |
|---|---|
| Sample size | No statistical method was used to determine sample size. Three replicates is standard for biochemical assays. |
| Data exclusions | No data were excluded from the analysis. |
| Replication | The number of times each experiment was performed is specified in the "Statistics and Reproducibility" section of the methods. All attempts at replication were successful. |
| Randomization | Randomization is not relevant to this study. Biochemical experiments are rarely randomized. |
| Blinding | Blinding was not performed during data acquisition or analysis. Biochemical experiments are rarely blinded |

# Reporting for specific materials, systems and methods

We require information from authors about some types of materials, experimental systems and methods used in many studies. Here, indicate whether each material, system or method listed is relevant to your study. If you are not sure if a list item applies to your research, read the appropriate section before selecting a response.

### Materials & experimental systems

| n/a | Involved in the study |
|---|---|
| ☐ | ☒ Antibodies |
| ☐ | ☒ Eukaryotic cell lines |
| ☒ | ☐ Palaeontology and archaeology |
| ☐ | ☒ Animals and other organisms |
| ☒ | ☐ Human research participants |
| ☒ | ☐ Clinical data |
| ☒ | ☐ Dual use research of concern |

### Methods

| n/a | Involved in the study |
|---|---|
| ☒ | ☐ ChIP-seq |
| ☐ | ☒ Flow cytometry |
| ☒ | ☐ MRI-based neuroimaging |

## Antibodies

| | |
|---|---|
| Antibodies used | Anti-FLAG antibody (M2, Sigma F1804); anti-BmAgo3 (Created by Izumi in: Izumi et al., 2020, Nature 578, 311-316 and used by Izumi in this manuscript); alpha-Tubulin antibody #2144 (https://www.cellsignal.com/) |
| Validation | Anti-FLAG antibody (https://www.sigmaaldrich.com/content/dam/sigma-aldrich/docs/Sigma/Bulletin/f1804bul.pdf); anti-BmAgo3 was produced and validated in a previous study (Izumi et al., 2020, Nature 578, 311-316); anti-alpha-Tubulin antibody (https://www.cellsignal.com/) |

## Eukaryotic cell lines

Policy information about cell lines

| | |
|---|---|
| Cell line source(s) | HEK 293T (lab stock; commercially available from ATCC CRL-3216) and BmN4 cell line (provided by Dr. Kusakabe, Kyushu University and available for purchase from Riken Cell Bank- https://cellbank.brc.riken.jp/cell_bank/CellInfo/?cellNo=RCB2126&lang=En) |
| Authentication | The cells were not authenticated; cells were used only to produce recombinant proteins |
| Mycoplasma contamination | Not tested. |
| Commonly misidentified lines (See ICLAC register) | No commonly misidentified lines were used in the study. |

# Animals and other organisms

Policy information about <u>studies involving animals</u>; <u>ARRIVE guidelines</u> recommended for reporting animal research

| | |
|---|---|
| Laboratory animals | C57BL/6 mice: JAX#000664; adult male<br>FLAG-Gtsf1KI mice: generated in the C57BL/6J background in this study; adult male.<br>Animals were housed in an AALAC-accredited barrier facility with controlled temperature (22°± 2°), relative humidity (40% ± 15%), and a 12-hour dark/light cycle.<br><br>All animals used in experiments were two to six months old. |
| Wild animals | No wild animals were used in this study. |
| Field-collected samples | No field-collected samples were used in this study. |
| Ethics oversight | (1) PI on IACUC protocol: Phillip D. Zamore<br>(2) Name of IACUC: UMass Chan Medical School Institutional Animal Care and Use Committee<br>(3) IACUC Docket: A2222-17, "Investigation of mechanisms of small RNA function in vivo"<br>(4) Mice were maintained and used according to the guidelines of the Institutional Animal Care and Use Committee of the University of Massachusetts Chan Medical School (A201900331). |

Note that full information on the approval of the study protocol must also be provided in the manuscript.

# Flow Cytometry

## Plots

Confirm that:

☐ The axis labels state the marker and fluorochrome used (e.g. CD4-FITC).

☐ The axis scales are clearly visible. Include numbers along axes only for bottom left plot of group (a 'group' is an analysis of identical markers).

☐ All plots are contour plots with outliers or pseudocolor plots.

☐ A numerical value for number of cells or percentage (with statistics) is provided.

## Methodology

| | |
|---|---|
| Sample preparation | *Describe the sample preparation, detailing the biological source of the cells and any tissue processing steps used.* |
| Instrument | *Identify the instrument used for data collection, specifying make and model number.* |
| Software | *Describe the software used to collect and analyze the flow cytometry data. For custom code that has been deposited into a community repository, provide accession details.* |
| Cell population abundance | *Describe the abundance of the relevant cell populations within post-sort fractions, providing details on the purity of the samples and how it was determined.* |
| Gating strategy | *Describe the gating strategy used for all relevant experiments, specifying the preliminary FSC/SSC gates of the starting cell population, indicating where boundaries between "positive" and "negative" staining cell populations are defined.* |

☐ Tick this box to confirm that a figure exemplifying the gating strategy is provided in the Supplementary Information.

