## [Peer Review File · Nature]

Manuscript Title: GTSF1 accelerates target RNA cleavage by PIWI-clade Argonaute proteins

Reviewer Comments & Author Rebuttals

Reviewer Reports on the Initial Version:

Referees' comments:

Referee #1 (Remarks to the Author):

In this study, Arif et al. report that binding of GTSF1 to piRISCs increases the target cleavage by more than 100-fold, answering the long-standing question of the low endonuclease activity of PIWI proteins in vitro. The current study is comprehensive, and the data are convincing. The discoveries of the specificity of the GTSF carboxy-terminal domain and the evolutionary divergence are interesting. The paper will be of great interest to the PIWI community and meets the standards for publication in Nature. This referee's major comment is that the authors mentioned they identified GTSF1 as an auxiliary factor of piRISCs based on their classical column chromatography and activity assays. But they didn't seem to describe how they identified GTSF1.

Major concerns

1. (Extended Data Fig. 5) Was the amount of the affinity-purified proteins adjusted? Otherwise, the results would be misread. If they did, the authors should describe the methods in detail.
2. (Page 11) The main point of this paper is that GTSF1 increases the PIWI endoribonuclease activity. Therefore, in the sentence "Moreover, a requirement for ...," the authors would be better to describe the shortest guide length required for efficient target cleavage rather than for detectable cleavage.
3. (Extended Data Fig. 3c) In the NEM-treated samples, the substrate disappeared even in 5 min. This result seems to suggest NEM affected the substrate as well as the potentiating activity.
4. (The last line on Page 6) It is not clear how the authors were able to identify GTSF1 as a MIWI-potentiating factor from only the results of the SEC and the activity test (Fig. 3b). Did they determine the protein by mass spectrometry or other methods?
5. Compared to MIWI piRISC, a 100-times excess amount of GTSF1 was used in this study. Does their molar ratio reflect that in physiological conditions? If the authors don't know, the in vitro target cleavage needs to be done with 1- and 10-times excess amount of GTSF1, too.
6. The authors of ref. 95 didn't seem to mutate R26, R29, K36, and K39 or test the mutant for RNA binding. In addition, the current study examined the mutant for target cleavage but not for RNA binding. RNA-binding assays are required to conclude that the four basic residues are essential for GTSF1 to bind RNAs.

Other comments

7. (Page 5) Based on the context and Extended Data Figs. 2a and b, the sentence of “Pre-incubation of MIWI with testis lysate prior to, or after loading ...” needs to be changed to something like “Pre-incubation of MIWI with testis lysate or buffer prior to loading the protein with piRNA...”.
8. (Fig. 2b and Extended Data Fig. 1c) This referee guesses that the image on the bottom is shown to increase image contrast. If so, that should be specified.
9. (Fig. 3f) The markers don't seem to show up properly.
10. (Fig. 3) It is hard to distinguish the color codes of GTSF1 and GTSF1L.

Referee #2 (Remarks to the Author):

Manuscript 'The tiny, conserved zinc-finger protein GTSF1 helps PIWI proteins achieve their full catalytic potential'

By Arif and colleagues

Here, the authors assay for piRISC-guided cleavage of a target RNA in vitro. They observe that the addition of testis extract to immunopurified MIWI-piRISC increases target cleavage activity (Fig. 1). They proceed to identify the activator by classical biochemical fractionation of the testis extract, and pinpoint some of the activity to a low molecular weight fraction (Fig. 2). The authors subsequently hypothesize that the activity might be dependent on zinc (EDF 3). Due to some indication for an activating function of zinc and the rough agreement of the low molecular weight of the activator, the authors decide to focus on a candidate that has previously been implicated in piRNA silencing in mice and flies (Yoshimura and Miyazaki, 2018; Donertas and Brennecke, 2013; Ohtani and Saito, 2013; Muerdter and Hannon, 2013). Gtsf1 is a small zinc-finger protein that interacts with PIWI proteins and preferentially binds tRNAs (Ipsaro et al., 2021). The authors show that the addition of Gtsf1 increases MIWI- and MILI- target-cleavage activity in their in vitro assay (Fig 3). The effect is dependent on the ability of Gtsf1 to bind RNA. However, the results about a potential interaction between PIWI and Gtsf1 in this context are inconclusive: Fig. 3 suggests that the binding mutant reduces activation; however, the stimulatory activity from testis extract does not associate with MIWI (EDF 2). Finally, the authors show that piRISC-cleavage activity increases with increasing complementarity between the piRNA and the target RNA (Fig. 4).

The authors suggest there is a requirement for GTSF1 as a potentiating factor in piRNA-guided target cleavage in vitro. Nevertheless, the molecular mechanism and the biological relevance of this finding remains unknown. The presented data seem preliminary and fail to explain previous genetic and structure-function data. The manuscript is well written, but the data are weak, and the conclusions remain speculative. This work might be suitable for a more specialized journal after revision.

Specific comments:

Experimental system: Addressing enzyme kinetics in vitro requires a stable and homogenous experimental system. Such a system optimally consists of highly purified soluble particles. In contrast, the presented system consists of a simple immuno-purification of ectopically expressed FLAG-MIWI followed by loading with a synthetic oligo in vitro. Loading of Argonaute proteins with a small RNA is difficult and inefficient in vitro. The FLAG-eluate is expected to contain unloaded, misloaded and some correctly loaded MIWI. This heterogenous mixture poses a problem for the interpretation of the presented data, especially because any modulating activity may depend on direct interaction with the MIWI protein. To address this technical problem, the authors previously established a purification procedure that uses antisense complementarity to enrich for the correctly loaded complex (Flores-Jasso and Zamore 2013). This purification scheme should be added to improve the homogeneity of the assayed complexes. Alternatively, the authors could establish a more elegant and robust in vitro system based on soluble recombinant proteins (Anzelon and MacRae bioRxiv 2020.12.07.413112).

Further technical improvements:

The datapoints in Fig. 3f are highly variable and the curve seems overfitted. Datapoints and standard deviations should be depicted as in 3d. Replicates should be shown in Fig 4b,c.

The authors speculate on effects of PAZ-piRNA interactions (Fig 5c), but there are no experiments addressing this point. These speculations should be tested using PAZ mutants.

The model in EDF 8 is a theoretical depiction and not based on actual data. It contains variables that can't be assessed in the presented assays. The authors should clarify the purely speculative nature of this depiction.

Requirement for zinc? These assays are not clear. (1) EDTA is expected to inhibit piRISC activity because Argonaute proteins are Mg-dependent nucleases. This argument cannot be used as line of evidence for Zn. (2) The NEM treatment seems to either degrade or precipitate the reaction so that neither the substrate nor the products are visible (EDF 3c). (3) Phenanthroline seems to result in significant RNA degradation, which makes the interpretation of EDF 3g difficult. (4) The fact that both Ni and Zn columns retain the activity (EDF 3h), could have various explanations. Do they also deplete the activity from the flow though?

The concentration of zinc used in the in vitro assay exceeds physiological concentrations by >10000 fold and is thus irrelevant.

Interestingly, high concentrations of Mg (12.5 mM ~ 6x physiological concentration) seem to stimulate the reaction most efficiently (EDF 3f). The authors should test if the initially observed activator is indeed only magnesium, which might exist at higher concentrations in germ granules (speculation). The authors should also evaluate the effect of other divalent cations (Anzelon and MacRae bioRxiv 2020.12.07.413112).

The molecular mechanism of GTSF1:

The RNA binding activity of GTSF1: The authors show that the RNA binding activity of GTSF1 is required for its function (Fig. 3). Which RNAs do GTSF1 proteins bind in this system? Is the function related to the GTSF-bound tRNAs that co-purify with the protein (Ipsaro and Joshua-Tor, 2021)? Does Gtsf1 bind either the piRNA or the substrate in the in vitro assay?

RNA binding has only clearly been shown for the first zinc finger. The authors should use RNA binding mutants for each zinc-finger separately to test which one is required and which binds RNA.

Protein-protein interactions: GTSF1 interacts with PIWI complexes via aromatic residues in its central region (Donertas and Brennecke 2013; Yoshimura and Miyazaki, 2018). Mutant GTSF1 seems impaired in stimulating cleavage activity in vitro (Fig. 3F). However, the activator from testis extracts does not associate with MIWI (EDF 2). This discrepancy in the presented data needs clarification. Perhaps GTSF1 is not the same factor that is responsible for the stimulating activity of the mouse testis extract. To resolve this question, the authors should test extract from GTSF1 knockout testis or testis extract depleted of GTSF1 and compare its stimulative function with wild type extract. It would also be interesting to know where within PIWI complexes GTSF1 binds, whether the binding is direct, and if not, which cofactor mediates the interaction. The authors should test the interaction of different GTSF proteins with different PIWI proteins and check if interaction strength correlates with activation of cleavage activity (EDF 5)

Biological relevance:

PiRISC is supposed to catalyze multiple rounds of cleavage in vivo. Target cleavage takes place in specialized cytoplasmic granules and turn-over is likely stimulated by the RNA helicase VASA/DDX4 (Xiol and Pillai, 2004). Identifying regulatory mechanisms of piRISC activity requires more effort to reconstitute a relevant environment in vitro and/or complementary assays in vivo.

Drosophila GTSF1 localizes to the nucleus and associates with the piRISC that induces transcriptional silencing without target cleavage (Donertas and Brennecke, 2013). There is no report of cytoplasmic GTSF or any involvement in ping-pong. This existing literature does not fit with the proposed model.

Mouse Gtsf1 has been reported to localize to granules in both, nuclear and cytoplasmic compartments (Yoshimura 2018). While MIWI-1 piRNAs are reduced in the mutant, MILI-piRNAs, which are also ping-pong generated, are not. If GTSF1 stimulation was important for MILI-piRISC activity, one would expect a reduction of all ping-pong piRNAs. The discrepancy must be addressed.

GTSF1 has been shown to interact with tRNAs. It is suggested to prevent transposition mutagenesis (Ipsaro and Joshua-Tor, 2021). How do the authors integrate the bound tRNA in their model?

Overall, the proposed (in vitro) model does not fit with prior observations in vivo. These discrepancies are not sufficiently addressed.

Referee #3 (Remarks to the Author):

PIWI proteins are slicers that use the piRNA as a guide to cleave target RNAs. However, unlike AGO

proteins, in vitro purified PIWI RISC has inefficient catalytic activity. The manuscript by Arif et al. identifies GTSF1 as a protein factor that significantly enhances the catalytic activity of mammalian PIWI proteins in vitro. GTSF1 paralogs GTSF1L and GTSF2 and ortholog silkworm BmGTSF1 also have the ability to potentiate in vitro piRNA-directed PIWI cleavage activities. This work clearly advances the piRNA-guided PIWI in vitro cleavage system, and sheds light on a potential physiological role for GTSF1 as a PIWI-slicing auxiliary protein in vivo.

Major comments

1. GTSF1 is known to have a nuclear function not associated with PIWI slicer activity in both flies and mice (ref. 7, 9). *Drosophila* Gtsf1 mainly colocalizes with nuclear Piwi, but not with cytoplasmic Aub or Ago3. In mice, GTSF1 colocalizes predominately with MIWI2, to a lesser extent with cytoplasmic MILI. Both *Drosophila* Piwi and mouse MIWI2 direct transcriptional silencing but lack slicer activity. How does GTSF1 facilitate PIWI nuclear function as a binding partner? The authors propose GTSF1 as an auxiliary protein to enhance PIWI catalytic activities, how can this function be proven in a physiological setting in germ cells?
2. The biochemical basis for GTSF1 to facilitate PIWI slicing remains unclear. How GTSF1 interacts with PIWI proteins and binds RNA was poorly demonstrated in the manuscript. The authors mainly cited data from others' published work. On page 11, "GTSF1 does not detectably co-immunoprecipitate with MIWI piRISC from mouse testis (ref. 5, 77)". This seemingly contradicts with a role for GTSF1 as a MIWI slicing auxiliary factor. It would be important to comprehensively map the interaction of GTSF1 with MIWI, MILI, and MIWI2 in the testis and by ectopic expression in cell lines using co-IP and co-staining. To experimentally show how GTSF1 uses different domains/motifs to bind RNA and interact with PIWI proteins will greatly improve this manuscript.
3. One crucial result leading to the unbiased identification of MIWI potentiating factor was not shown. In Fig. 2b, have the authors performed mass spectrometry on the ~17 peak? Is GTSF1 in the fraction? What else is in the fraction? This is important to know.

Other comments

1. Sequences of synthetic piRNAs and corresponding targets used in this study were not provided. A table summarizing these sequences and their use in specific figures is needed. How are the piRNA sequences selected? Do they have natural targets in germ cells? How many different piRNAs have been tested in cleavage assays and has GTSF promoted slicing in all cases?
2. It will be helpful to show the domain architecture of GTSF1L and GTSF2. How do these proteins interact with MIWI or MILI? Do they have a similar PIWI interacting motif like GTSF1?
3. Have the authors tested other recombinant piRNA biogenesis factors (e.g. MVH) in the target cleavage assay to show GTSF1 is specific in potentiating slicing?
4. Page 8, What is the structural basis for 3 Ws in GTSF1 to interact with PIWI? Have the authors validated mutation in these residues ablate binding to MIWI?

5. Page 8, Can GTSF1 bind target RNA in the cleavage assay? Need to show the GTSF1 RRKK mutant abolishes RNA binding but does not affect its binding to PIWI.

6. Page 8, “mutating these residues-W98A, W107A, and W112A in mice”, please clarify if this is mutating the mouse protein or generating mutant mice?

Author Rebuttals to Initial Comments:

In this study, Arif et al. report that binding of GTSF1 to piRISCs increases the target cleavage by more than 100-fold, answering the long-standing question of the low endonuclease activity of PIWI proteins in vitro. The current study is comprehensive, and the data are convincing. The discoveries of the specificity of the GTSF carboxy-terminal domain and the evolutionary divergence are interesting. The paper will be of great interest to the PIWI community and meets the standards for publication in Nature.

Thank you!

1. *(Extended Data Fig. 5) Was the amount of the affinity-purified proteins adjusted? Otherwise, the results would be misread. If they did, the authors should describe the methods in detail.*

We used 100 nM recombinant purified GTSF1. This information was hidden in the Methods (“FLAG-Siwi, MIWI, and MILI target cleavage assays”). To make it easier to find, we have added the concentration to the figure legend.

2. *(Page 11) The main point of this paper is that GTSF1 increases the PIWI endoribonuclease activity. Therefore, in the sentence “Moreover, a requirement for ...,” the authors would be better to describe the shortest guide length required for efficient target cleavage rather than for detectable cleavage.*

We would prefer to write “detectable” (observed signal above the limit of detection), because “efficient” is an arbitrary metric. This section of the manuscript describes the effect of guide:target mismatches on cleavage using the biologically relevant guide length (30-nt). The consequence of different guide lengths is examined in the next section. Guide length was examined in order to test the two-state hypothesis for Argonaute function. Such short guides are not naturally bound to MIWI in vivo.

3. *(Extended Data Fig. 3c) In the NEM-treated samples, the substrate disappeared even in 5 min. This result seems to suggest NEM affected the substrate as well as the potentiating activity.*

Neither the substrate nor piRISC was exposed to unreacted NEM; all NEM was inactivated by adding excess DTT before adding target or piRISC. The mock and treated samples differ only in the order of addition of NEM and DTT to the testis lysate. As the Reviewer notes, the NEM treated lysate degrades the substrate at long time points, probably because a ribonuclease inhibitor protein such as RNH1 or the testis-specific RNH2 (Miyamoto & Hasuike *J Assist Reprod Genet* 2002) is inactivated; as much as 7% of the amino acids in ribonuclease inhibitors can be cysteine, and ribonuclease inhibitors are known to be inactivated by NEM (Dickson et al., *Prog Nucleic Acid Res Mol Biol* 2005). To avoid target

degradation by nucleases in the NEM-treated testis lysate, we have repeated the experiment using a shorter time course (Reviewers' Figure R1). We have substituted these data for the original longer time course in Extended Data Figure 3c.

4. *(The last line on Page 6) It is not clear how the authors were able to identify GTSF1 as a MIWI-potentiating factor from only the results of the SEC and the activity test (Fig. 3b). Did they determine the protein by mass spectrometry or other methods?*

No, we did not use mass spectrometry. We matched the properties of the activity to those of known genetic mutants. What we knew was:

- the protein was ~17 kDa
- required one or more structural zinc ions
- required reduced cysteines for its activity
- was highly expressed in meiotic spermatocytes compared to mitotic spermatogonia.

Among known genetic mutants, only GTSF1 met all of these criteria, so we immediately tested the recombinant protein. One week later, the lab was shut-down for three months because of the pandemic.

We would like to note that it is extremely difficult to identify very small proteins such as GTSF1 by mass spectrometry. Published experiments have used mass spectrometry to identify Piwi proteins as GTSF1 interactors, but the reverse has only been successful by searching for GTSF1 peptides, rather than looking for proteins with the most enriched peptides. Based on the Reviewers' comments, we performed mass spectrometry on our original gel filtration fractions. As expected, we obtained peptides corresponding to GTSF1, but we would not have been able to identify GTSF1 with sufficient statistical rigor to conclude by mass spectrometry that it was the MIWI potentiating factor. In retrospect, we're lucky we guessed right.

5. *Compared to MIWI piRISC, a 100-times excess amount of GTSF1 was used in this study. Does their molar ratio reflect that in physiological conditions? If the authors don't know, the in vitro target cleavage needs to be done with 1- and 10-times excess amount of GTSF1, too.*

The experiments requested by the Reviewer were presented in Supplementary Data Figure 2 and Figure 3f of our original manuscript (3e in the revised manuscript), which showed the rate of target cleavage by 5 nM MIWI piRISC incubated with GTSF concentrations spanning three orders of magnitude. This corresponds to GTSF1:piRISC molar ratios from 0.2 to 200 for wild-type GTSF1 and from 0.2 to 1000 for the GTSF1^{W98A,W107A,W112A} mutant. To make this more clear, we have added to Figure 3f an additional x-axis indicating the molar ratio of GTSF1 to piRISC (see also Reviewers' Figure R2). We have also changed the

way we display the data and the fitted curve in response to the concerns of Reviewer 2 (below).

6. *The authors of ref. 95 didn't seem to mutate R26, R29, K36, and K39 or test the mutant for RNA binding. In addition, the current study examined the mutant for target cleavage but not for RNA binding. RNA-binding assays are required to conclude that the four basic residues are essential for GTSF1 to bind RNAs.*

Ipsaro et al. (Ref. 95) tested these mutants for RNA-binding in their Supplemental Figure 4. They reported that “Mutation of certain basic residues in ZnF1 abrogate RNA-binding activity including R26, R29, K36, and K39.”

7. *(Page 5) Based on the context and Extended Data Figs. 2a and b, the sentence of “Pre-incubation of MIWI with testis lysate prior to, or after loading ...” needs to be changed to something like “Pre-incubation of MIWI with testis lysate or buffer prior to loading the protein with piRNA...”.*

We have rewritten this to read, “Pre-incubation of MIWI with testis lysate either before or after loading with a piRNA...” (Page 6, line 140).

8. *(Fig. 2b and Extended Data Fig. 1c) This referee guesses that the image on the bottom is shown to increase image contrast. If so, that should be specified.*

Yes, the image at the bottom shows a digital overexposure. We have added this information to the figure legends.

9. *(Fig. 3f) The markers don't seem to show up properly.*

Each data point was marked with the word “rep1”, “rep2,” or “rep3” instead of a circular marker. We apologize that this was confusing. In response to the comments of Reviewer 2 (below), **we have now drawn the figure in a more traditional style**, fitting to the mean of the three replicate data sets.

10. *(Fig. 3) It is hard to distinguish the color codes of GTSF1 and GTSF1L.*

We have updated our color choice; thank you for pointing this out.

Referee #2

Here, the authors assay for piRISC-guided cleavage of a target RNA in vitro. They observe that the addition of testis extract to immunopurified MIWI-piRISC increases target cleavage activity (Fig. 1). They proceed to identify the activator by classical biochemical fractionation of the testis extract, and pinpoint some of the activity to a low molecular weight fraction (Fig. 2). The authors subsequently hypothesize that the activity might be dependent on zinc (EDF 3). Due to some indication for an activating function of zinc and the rough agreement of the low molecular weight of the activator, the authors decide to focus on a candidate that has previously been implicated in piRNA silencing in mice and flies (Yoshimura and Miyazaki, 2018; Donertas and Brennecke,

2013; Ohtani and Saito, 2013; Muerdter and Hannon, 2013). *Gtsf1* is a small zinc-finger protein that interacts with PIWI proteins and preferentially binds tRNAs (Ipsaro et al., 2021).

The authors show that the addition of *Gtsf1* increases MIWI- and MILI- target-cleavage activity in their *in vitro* assay (Fig 3). The effect is dependent on the ability of *Gtsf1* to bind RNA. However, the results about a potential interaction between PIWI and *Gtsf1* in this context are inconclusive: Fig. 3 suggests that the binding mutant reduces activation;

The Reviewer mischaracterizes our result; Figure 3d shows that the RNA-binding mutant has **no detectable activity** to potentiate MIWI-catalyzed target cleavage at saturating concentrations of GTSF1. The rate of cleavage is indistinguishable from piRISC alone. We cannot detect potentiation activity in the RNA-binding mutant using any available assay.

however, the stimulatory activity from testis extract does not associate with MIWI (EDF 2).

Again, this is not what our data showed. Neither pre-incubating MIWI nor incubating piRISC with testis lysate increased piRISC activity. These experiments only show that under the specific conditions of our protocol (extensive washing while MIWI is tethered to magnetic beads), the potentiating activity does not co-purify with MIWI. They do not speak to whether the activity associates with MIWI under more permissive conditions or in the presence of target RNA.

The authors suggest there is a requirement for GTSF1 as a potentiating factor in piRNA-guided target cleavage in vitro. Nevertheless, the molecular mechanism and the biological relevance of this finding remains unknown. The presented data seem preliminary and fail to explain previous genetic and structure-function data.

We respectfully disagree. Our detailed kinetic data, use of point mutants, and multiple GTSF1 orthologs explain nearly all previous genetic and structure-function data. **Reviewers' Table R1 summarizes the published genetic and molecular observations for GTSF1 in flies, silk moths, and mice and note how our data help explain them.** Remarkably, GTSF1 orthologs in mice and silk moth can distinguish between PIWI-protein paralogs. We note that GTSF1 is the only example of a protein that alters the catalytic activity of an Argonaute protein in any eukaryote or prokaryote.

Experimental system: Addressing enzyme kinetics in vitro requires a stable and homogenous experimental system. Such a system optimally consists of highly purified soluble particles.

Over the past 22 years, our lab developed the *in vitro* systems used to study the kinetics of fly (Tuschl et al., *Genes Dev* 1999; Zamore et al., *Cell* 2000), plant

(Tang et al., *Genes Dev* 2003), and mammalian (Schwarz et al. *Mol Cell* 2002; Hutvagner et al., *Science* 2002) Argonaute proteins. Before 2012 (Wee et al., *Cell* 2012), all kinetic experiments studying siRNAs and miRNAs were performed in fly embryo lysate, wheat germ extract, or HeLa cell S100. **No system allowing the kinetic study of any mammalian or insect PIWI protein was available before our current manuscript.** The work we present is the first time MIWI or MILI has been purified to homogeneity and loaded with a synthetic piRNA guide of defined sequence. This advance allowed us to perform detailed pre-steady-state and steady-state kinetic experiments. **Moreover, our system does in fact comprise highly purified soluble particles, as we explain below.**

In contrast, the presented system consists of a simple immuno-purification of ectopically expressed FLAG-MIWI followed by loading with a synthetic oligo in vitro.

No purified fly or mammalian PIWI protein has been loaded with a small RNA of defined sequence prior to our work. Getting this to work was not a question of a simple immunopurification, but rather the culmination of four years of work. Expressing MIWI at sufficiently high levels to produce unloaded protein required multiple rounds of lentiviral transfection, each time purifying those cells with the highest level of expression. Loading required development of novel methods and optimization. If this had been simple, we or others would have reported it years ago.

Loading of Argonaute proteins with a small RNA is difficult and inefficient in vitro.

We respectfully disagree. We have shown that loading fly, plant, mouse, human, and bacterial Argonaute proteins with a small RNA (or DNA) is highly efficient and straightforward (Zamore et al., *Cell* 2000; Nykanen et al., *Cell* 2001; Schwarz et al., *Mol Cell* 2002; Schwarz et al., *Cell* 2003; Tang et al., *Genes Dev* 2003; Tomari et al., *Science* 2004; Haley et al., *NSMB* 2004; Tomari et al., *Cell* 2004; Schwarz et al., *Curr Biol* 2004; Matranga et al., *Cell* 2004; Schwarz et al., *PLoS Genet* 2006; Forstemann et al., *Cell* 2007; Tomari et al., *Cell* 2007; Wee et al., *Cell* 2012; Flores-Jasso et al., *RNA* 2013; Salomon et al., *Cell* 2015; Smith et al., *Nat Comm* 2019; Becker et al., *Mol Cell* 2019; Jolly et al., *Cell* 2020).

The FLAG-eluate is expected to contain unloaded, misloaded and some correctly loaded MIWI.

In the experiments in our manuscript, the FLAG-MIWI was purified to apparent homogeneity. **The protein is highly purified and soluble, since the lysate was first centrifuged at 100,000 × g for 30 min, as indicated in Figure 1a (“S100”).** Extended Data Figure 1a and Supplementary Data Figure 1a showed that our purified MIWI corresponds to a single band on a Coomassie-stained gel. These data are also shown in Reviewers’ Figure R3.

It is not necessary to have a preparation of MIWI in which all the protein is loaded with the same guide, because **the concentration of active MIWI loaded with the synthetic guide was precisely determined using pre-steady-state kinetics, a method we have used in multiple papers beginning in 2012.** We note that this same approach was used for *Thermus thermophilus* Argonaute (TtAgo; Jolly et al., *Cell* 2021), which also cannot be purified by sequence-affinity chromatography.

This heterogenous mixture poses a problem for the interpretation of the presented data, especially because any modulating activity may depend on direct interaction with the MIWI protein.

Our purified MIWI is not a heterogeneous mixture. It has been purified to apparent homogeneity and comprises only soluble protein. As we have done for nearly a decade (Wee et al., *Cell* 2012), we report only the concentration of *active* protein containing the synthetic guide. We used pre-steady-state “burst” measurements to quantify the active protein for every preparation of purified piRISC. Thus, for all experiments in which target cleavage is measured, the background of MIWI loaded with guides derived from RNA in HEK-293 cells does not influence the measurements: without loading of a synthetic guide, the purified MIWI does not cleave the target RNA (Extended Data Figure 1a). Moreover, all of our experiments using recombinant wild-type or mutant GTSF1 or the GTSF1 paralogs used saturating concentrations of the protein that were substantially greater than the *total* MIWI or MILI concentration.

Because we use the hyperbolic fit to rates rather than to protein concentrations, [MIWI] or [piRISC] do not affect our estimates of the K_D of the GTSF1:MIWI interaction.

To address this technical problem, the authors previously established a purification procedure that uses antisense complementarity to enrich for the correctly loaded complex (Flores-Jasso and Zamore 2013). This purification scheme should be added to improve the homogeneity of the assayed complexes.

Needless to say, we spent a great deal of effort unsuccessfully trying to apply our previously described sequence-affinity purification method to MIWI. Not all Argonaute proteins can be purified by the method of Flores-Jasso and Zamore. For example, TtAgo cannot be purified by this method (Salomon et al., *Cell* 2015; Smith et al., *Nature Communications* 2019; Jolly et al., *Cell* 2020). Similarly, PIWI proteins do not elute when the capture oligo has sufficient complementarity to bind MIWI piRISC.

Alternatively, the authors could establish a more elegant and robust *in vitro* system based on soluble recombinant proteins (Anzelon and MacRae *bioRxiv* 2020.12.07.413112).

Our *in vitro* system is based on soluble recombinant protein. We established a stable mammalian cell line by three sequential cycles of transduction with lentivirus and FACS sorting, establishing a line over-expressing tagged, unloaded MIWI. Tagged MIWI was purified from the supernatant of a $100,000 \times g$ spin to ensure that only soluble protein was used for subsequent immunoaffinity purification. We note that in both Anzelon et al. (*Nature* 2021) and our study, the PIWI protein was purified in a single step using an engineered N-terminal tag and corresponds to a single band on a gel (see Reviewers' Figure R3).

While, we certainly agree that the work of Anzelon et al. (*Nature* 2021) is elegant, one cannot study the mammalian piRNA pathway using sponge PIWI. Moreover, in their original *bioRxiv* preprint, Anzelon et al. note that despite considerable efforts, they were not able to produce empty, loadable Piwi protein from any vertebrate, including mouse.

We previously used recombinant mouse AGO2—produced by over-expression in cultured cells and purified via an epitope tag—successfully in both ensemble and single-molecule studies (Wee et al., *Cell* 2012; Salomon et al., *Cell* 2015). In our published studies of fly Ago2, mouse AGO2, and bacterial TtAgo, we determined the percent active protein for every preparation using either burst kinetics or stoichiometric binding titrations. None of those proteins were 100% active. This was not an impediment to our detailed thermodynamic and kinetic experiments because we measured the percent active protein (as well as the percent active RNA in binding studies). These same analytical methods were used for every preparation of MIWI and MILI in this manuscript.

Nonetheless, the question as to whether GTSF1 can potentiate EfPiwi—the sponge protein studied by Anzelon et al.—is important, since it speaks to the evolutionary origins of GTSF1 as an activator of the Piwi endonuclease. Although the *Ephydatia fluviatilis* genome remains to be sequenced, a high quality genome assembly of its sister species, *Ephydatia muelleri*, is available. In our revised manuscript, we have now purified EmGtsf1 and performed target cleavage reactions using EfPiwi piRISC, which was loaded with a synthetic piRNA and purified by the methods described by Anzelon et al. (*Nature* 2021). Adding EmGtsf1 to EfPiwi increased the rate (k_{obs}) of target-cleavage by EfPiwi ~100-fold in physiological Mg^{2+} and >6-fold in supraphysiological Mn^{2+} (2 mM; revised Figure 3g and Extended Data Fig. 4c). Mn^{2+} , whose ionic radius is larger than that of Mg^{2+} , often enhances catalysis by Argonaute proteins. The larger effect of EmGtsf1 on EfPiwi in Mg^{2+} than Mn^{2+} is consistent with Gtsf1 increasing the time Piwi proteins spend in their catalytically active conformation.

Porifera (sponges) were the first phylum to branch off the evolutionary tree from the last common ancestor of all animals; all other animals are members of the clade Eumetazoa (Diploblasts). Our finding that EmGtsf1 stimulates target cleavage by EfPiwi is strong evidence that (1) the ancestral function of GTSF1 proteins was to stimulate the catalytic activity of Piwi proteins and (2) the ancestral Piwi protein was a sluggish enzyme. Together, our results demonstrate activation of a PIWI by GTSF1 can be reconstituted from entirely recombinant components for proteins from mouse and from sponges, animals whose last common ancestor lived ~900 Mya.

The datapoints in Fig. 3f are highly variable and the curve seems overfitted. Datapoints and standard deviations should be depicted as in 3d.

The data points in Figure 3f are *rates* for different concentrations of GTSF1 for three independent replicates. All of the 48 individual experiments (240 individual gel lanes) used to determine the rates were presented in Supplementary Data Figure 2. As described in the legend to Figure 3f, the curve was not fitted (and therefore was not overfitted), but rather corresponded to the mean K_D and k_{pot} of three independent experiments, each fitted independently (χ^2 for goodness of fit: 0.01, 0.02, 0.46 for the wild-type replicates; 0.02, 0.04, 0.02 for the GTSF1^{W98A,W107A,W112A} mutant). The data for all three replicates were shown in the figure, but the curve was not obtained by fitting to them. We sought to provide the most data-rich presentation possible, rather than provide only the mean and S.D. We apologize that this proved to be confusing rather than informative.

We have prepared a new figure (revised Figure 3f and Reviewers' Figure R2) using the more standard method of data presentation, showing rates as mean \pm S.D., fit to the equation,

$$k_{burst} = k_{pot} [\text{GTSF1}] / (K_D + [\text{GTSF1}]). \quad (\text{Equation R1})$$

The data are now displayed on a \log_{10} x-axis for improved visualization. We have also added a second x-axis showing the molar ratio of GTSF1:piRISC. The goodness-of-fit χ^2 ranged from 0.021–0.023 ($p \sim 4 \times 10^{-6}$).

Finally, to test that we have fit the appropriate model to our data (i.e., to ensure that the data are not overfitted), we performed a lack-of-fit F -test. The F -test takes as the null hypothesis that the relationship assumed by the model is reasonable, i.e., there is no lack of fit. For our model (Equation R1), the test statistic is $F^* = 0.032$. We compared our F^* to an F -distribution with 5 numerator degrees of freedom and 14 denominator degrees of freedom. At a significance level of $\alpha = 0.05$, the test statistic F^* is less than the critical value $F(0.95, df1 = 5, df2 = 14) = 2.96$. Therefore we cannot reject the null hypothesis (Equation R1), indicating that the model appropriately fits the data.

Replicates should be shown in Fig 4b,c.

We have performed the additional replicates and added them to the figures. We would like to note that the experiments required to add replicates to Figure 4b alone required eight independent piRISC preps, 384 time points, and 12 gels.

The authors speculate on effects of PAZ-piRNA interactions (Fig 5c), but there are no experiments addressing this point.

We respectfully disagree. The published literature and our experiments do address this point. Computational (Jung et al., *J Am Chem Soc* 2013), biochemical (Hur et al., *J Biol Chem* 2013), and structural studies (Schirle et al. *Science* 2014; Schirle et al. *eLife* 2015) of Argonaute proteins and structural studies of lepidopteran (Matsumoto et al., *Cell* 2016) and sponge (Anzelon et al., *Nature* 2021) PIWI show that only three regions of the protein contact the guide: (1) the 5' phosphate and the g1 base reside in the MID domain; (2) the MID and PIWI domains display all or part of the seed sequence in a stacked helical conformation; (3) and the 3' end and the last two piRNA nucleotides reside in the PAZ domain. In our experiments, the identity of the scissile phosphate—the site of target cleavage—is unaltered, demonstrating that shortening the piRNA from 30 nt to 16 nt preserved the interaction of the protein with the piRNA 5' phosphate and seed sequence. The only interpretation of our results consistent with all known properties of Argonaute proteins is that the 3' end of the piRNA spends less time in the PAZ domain as the piRNA is shortened.

These speculations should be tested using PAZ mutants.

Unfortunately, the experiment cannot be performed, since PAZ mutations in PIWI proteins prevent them from stably loading with piRNAs (Matsumoto et al., *Cell* 2016).

The model in EDF 8 is a theoretical depiction and not based on actual data. It contains variables that can't be assessed in the presented assays. The authors should clarify the purely speculative nature of this depiction.

We respectfully disagree. The model incorporates all available data from the field, including those in Anzelon et al. (2021) and our manuscript. Of course, by definition models are speculative; they are not intended solely to summarize what is known. A good model should provide testable predictions for future work. We would also like to note that the “two-state model” for Argonaute function was first proposed by us 16 years ago (Tomari and Zamore, *Genes Dev* 2005). All structural and kinetic data published to date support the model.

Requirement for zinc? These assays are not clear. (1) EDTA is expected to inhibit piRISC activity because Argonaute proteins are Mg-dependent nucleases. This argument cannot be used as line of evidence for Zn.

We respectfully disagree. The Mg^{2+} ions associated with Argonaute proteins are freely exchangeable with those in solvent (Schwarz et al., *Curr Biol* 2004; Yuan et al., *Mol Cell* 2005). Argonaute proteins, including MIWI, can be stripped of Mg^{2+} by EDTA, but are readily rescued when an excess of divalent cation (compared to EDTA) is added back, because the metal is not structural. Extended Data Figure 3 shows that excess Mg^{2+} rescues EDTA for the basal MIWI piRISC cleavage activity but not for testis-lysate-stimulated activity. These experiments demonstrate that (1) a divalent cation—distinct from the Mg^{2+} bound to MIWI—is required for testis-lysate to potentiate target cleavage and (2) excess divalent cation cannot restore potentiation activity, i.e., the metal bound to the protein is not in free equilibrium with divalent cation in solution. Zn-finger proteins are typically irreversibly denatured by chelating the zinc.

(2) The NEM treatment seems to either degrade or precipitate the reaction so that neither the substrate nor the products are visible (EDF 3c).

Neither the substrate nor piRISC was exposed to unreacted NEM; all NEM was inactivated by adding excess DTT before adding target or piRISC. The mock and treated samples differ only in the order of addition of NEM and DTT to the testis lysate. The NEM treated lysate degrades the substrate at long time points, probably because a ribonuclease inhibitor protein such as RNH1 or the testis-specific RNH2 (Miyamoto & Hasuike *J Assist Reprod Genet* 2002) is inactivated. As much as 7% of the amino acids in ribonuclease inhibitors can be cysteine, and ribonuclease inhibitors are known to be inactivated by NEM (Dickson et al., *Prog Nucleic Acid Res Mol Biol* 2005). To avoid target degradation by nucleases in the NEM-treated testis lysate, we have repeated the experiment using a shorter time course (Reviewers' Figure R1). We have substituted these data for the original longer time course in Extended Data Figure 3c.

(3) Phenanthroline seems to result in significant RNA degradation, which makes the interpretation of EDF 3g difficult.

We agree that treatment with phenanthroline increases nuclease activity in the lysate. Nonetheless, our interpretation, that Zn is required for the activity led us to discover that GTSF1 is the potentiating factor. Our original and new data show that GTSF1 is both necessary and sufficient for this, and the data explain all published observations about *Gtsf1* mouse mutants.

(4) *The fact that both Ni and Zn columns retain the activity (EDF 3h), could have various explanations. Do they also deplete the activity from the flow though?*

Yes: 50% of the piRISC-stimulating activity flows through the Ni-IMAC, whereas just 9% flows through the Zn-IMAC, i.e., Zn-IMAC retains 91% of the activity. We have added this information to Extended Data Figure 3h.

The concentration of zinc used in the in vitro assay exceeds physiological concentrations by >10 000 fold and is thus irrelevant.

The intracellular concentration of zinc is ~0.3 mM in a eukaryotic cell (Maret, *Metallomics* 2015). We used 12.5 mM Zn²⁺ in the presence of 10 mM EDTA. Thus, the free zinc concentration was 2.5 mM. That would be ~8-fold higher (and not 10⁴) than physiological conditions in the presence of EDTA and ~40-fold higher than physiological without EDTA.

Interestingly, high concentrations of Mg (12.5 mM ~ 6x physiological concentration) seem to stimulate the reaction most efficiently (EDF 3f). The authors should test if the initially observed activator is indeed only magnesium, which might exist at higher concentrations in germ granules (speculation).

The addition of supraphysiological magnesium generally stimulates all Argonaute proteins because the bound divalent ion is freely exchangeable with those in solvent. This is expected generally for metal-dependent nucleases, which typically have $K_{D,Mg} \sim 0.5\text{--}2$ mM. ($K_{D,Mg}$ is the Mg²⁺ concentration at which the rate of reaction, $v = \frac{1}{2}v_{max}$.) v will continue to increase asymptotically well beyond 10 mM Mg²⁺. By contrast, under physiological magnesium concentrations, testis lysate or rGTSF1 stimulate MIWI activity by two orders of magnitude.

It is highly unlikely that magnesium concentrations are different between germ granules and cytosol, given that only the macroviscosity is thought to be higher in macromolecular condensates; ions freely diffuse in and out of such materials.

The authors should also evaluate the effect of other divalent cations (Anzelon and MacRae bioRxiv 2020.12.07.413112).

In our manuscript, we had tested Mg²⁺ and Zn²⁺. We have now tested adding a physiological concentration (~0.1 mM) of Mn²⁺: the addition of 0.1 mM Mn²⁺ stimulated target cleavage ~4-fold, far less than the two-log stimulation by GTSF1. As described above, EmGTSF1 stimulates target cleavage by EfPiwi, even in the presence of Mn²⁺.

The RNA binding activity of GTSF1: The authors show that the RNA binding activity of GTSF1 is required for its function (Fig. 3). Which RNAs do GTSF1 proteins bind in this system?

We are currently developing assays to determine whether GTSF1 interacts with the piRNA guide or the target or both, but answering this question may require a cryo-EM structure of the quaternary complex of MIWI, GTSF1, piRNA, and target. We hope we can answer this important question in a future study.

Is the function related to the GTSF-bound tRNAs that co-purify with the protein (Ipsaro and Joshua-Tor, 2021)?

A UV scan of our purified rGTSF1 showed that the protein is free from contaminating nucleic acids. Like us, Ipsaro et al. found that GTSF1-bound RNA is removed by ion exchange chromatography (Supplementary Figure 1A, Ipsaro et al., 2021), which is one of the steps in our rGTSF1 purification protocol. We were unable to detect any change in piRNA-directed, MIWI-catalyzed target cleavage in the presence of GTSF1 when tRNA was added.

Does Gtsf1 bind either the piRNA or the substrate in the in vitro assay?

We do not currently know, but are working hard to determine whether GTSF1 binds the piRNA, substrate, or, most likely, both. We hope to report the results of our studies in a future paper.

RNA binding has only clearly been shown for the first zinc finger. The authors should use RNA binding mutants for each zinc-fingers separately to test which one is required and which binds RNA.

Ipsaro and Joshua-Tor (*Cell Reports* 2021; their Figure 1B) demonstrated that the second zinc finger does not participate in RNA binding.

Protein-protein interactions: GTSF1 interacts with PIWI complexes via aromatic residues in its central region (Donertas and Brennecke 2013; Yoshimura and Miyazaki, 2018). Mutant GTSF1 seems impaired in stimulating cleavage activity in vitro (Fig. 3F). However, the activator from testis extracts does not associate with MIWI (EDF 2). This discrepancy in the presented data needs clarification.

Our experiments showed that under the specific conditions of our protocol (extensive washing while MIWI is tethered to magnetic beads), the potentiating activity does not co-purify with MIWI. They do not speak to whether the activity associates with MIWI under more permissive conditions or in the presence of target RNA. We do not see any discrepancy.

Perhaps GTSF1 is not the same factor that is responsible for the stimulating activity of the mouse testis extract. To resolve this question, the authors should test extract from

GTSF1 knockout testis or testis extract depleted of GTSF1 and compare its stimulative function with wild type extract.

This is an excellent suggestion, and we agree with the reviewer that testing testis extract depleted of GTSF1 is a critical experiment that was missing from our manuscript. To address the Reviewer's question, we used CRISPR/Cas9 to insert a 3XFLAG tag into the endogenous locus. These epitope-tagged GTSF1 knock-in mice are male fertile, demonstrating that the tagged protein functions normally in vivo. Because the stimulatory activity is greatest in secondary spermatocytes, we prepared lysate from FACS-purified germ cells isolated from the homozygous knock-in mice and, as a control, from C57BL/6 mice. We then used anti-FLAG antibody to immunodeplete lysate from purified secondary spermatocytes isolated from the homozygous knock-in mice (Reviewers' Figure R4a). In parallel, we performed the same experiment using testis lysate from C57BL/6 mice, in which GTSF1 is not tagged.

For both the knock-in strain and the untagged control, lysate prepared from secondary spermatocytes stimulated piRNA-directed target cleavage by MIWI. **In contrast, the stimulatory activity was dramatically reduced in the immunodepleted extract prepared from secondary spermatocytes from the tag-expressing, but not the C57BL/6 control, mice** (Reviewers' Figure R4b). Moreover, the stimulatory activity was eluted with 3XFLAG peptide from the anti-FLAG beads incubated with germ cell lysate prepared from the tag-expressing, but not the control, mice. We conclude that the stimulatory activity present in the secondary spermatocytes corresponds to GTSF1. We have added these new data to the manuscript as Figure 2d–f, and discuss them on page 9 of the text.

It would also be interesting to know where within PIWI complexes GTSF1 binds, whether the binding is direct, and if not, which cofactor mediates the interaction. The authors should test the interaction of different GTSF proteins with different PIWI proteins and check if interaction strength correlates with activation of cleavage activity (EDF 5)

These experiments were in our original manuscript. Figure 3f showed that rGTSF1 binds directly to MIWI (GTSF1 and MIWI are the only two proteins in the reaction). In our original manuscript, we measured the K_D for wild-type GTSF1 and the GTSF1^{W98A,W107A,W112A} mutant (Figure 3f): the triple W mutation reduced the affinity of GTSF1 for MIWI >60-fold and can account for the difference in stimulatory activity of the wild-type and mutant proteins. The K_D for wild-type GTSF1 binding MIWI piRISC ~7 nM; the concentration of piRISC in meiotic spermatocytes is ~5 μ M (Gainetdinov et al., *Mol Cell* 2018). No additional protein should be required for GTSF1 to bind piRISC in vivo.

PiRISC is supposed to catalyze multiple rounds of cleavage in vivo.

To the best of our knowledge, no published work has tested whether piRISC is multiple-turnover in vivo, and no PIWI protein has ever been shown to catalyze multiple rounds of cleavage in vitro. Work from the Pillai and Siomi labs suggests that the DEAD-box proteins Vasa and DDX43 facilitate transfer of sliced precursors between ping-pong partners, not that Vasa promotes multiple turnover catalysis. We note that the proposed mechanism for ping-pong amplification of piRNAs does not require PIWI proteins to catalyze multiple rounds of cleavage. The finding that, in mice, MIWI and MILI are nearly as abundant as ribosomes (Gainetdinov et al., *Mol Cell* 2018) may be a hint that cleavage is not multiple turnover in vivo.

Target cleavage takes place in specialized cytoplasmic granules and turn-over is likely stimulated by the RNA helicase VASA/DDX4 (Xiol and Pillai, 2014). Identifying regulatory mechanisms of piRISC activity requires more effort to reconstitute a relevant environment in vitro and/or complementary assays in vivo.

No published study has shown that any function of the piRNA pathway occurs in nuage or other granules. Many piRNA pathway proteins localize to nuage, but whether they function there is unknown. No study has attempted to quantify the distribution of piRNA pathway proteins among cellular compartments. This is a common problem in cell biology: it is far easier to detect highly localized macromolecules than those that are more diffusely distributed. In a classic study of this phenomenon, Bergsten and Gavis (*Development* 1999) showed that the highly localized pool of *nanos* mRNA in polar granules—once thought to represent most if not all of the *nanos* mRNA in a fly embryo—actually corresponds to just 4% of the total.

Drosophila GTSF1 localizes to the nucleus and associates with the piRISC that induces transcriptional silencing without target cleavage (Donertas and Brennecke, 2013). There is no report of cytoplasmic GTSF or any involvement in ping-pong. This existing literature does not fit with the proposed model.

We strongly disagree.

First, Yoshimura et al. (*EMBO Reports* 2018) showed that ping-pong is lost in *Gtsf1*^{-/-} homozygous mutant mouse fetal testes. (We discuss this in detail below in response to the Reviewer's next comment.)

Second, Chen et al. (*PLoS Genetics* 2020) showed that BmGtsf1 is required for both piRNA biogenesis and BmSiwi effector functions in silk moth. Silk moths, like other lepidoptera, have no nuclear PIWI protein, and piRNA production depends strongly on the ping-pong pathway. We reanalyzed the raw piRNA sequencing data from Chen et al. When normalized to miRNAs, the *BmGtsf1* mutant shows a ~5-fold reduction in piRNA abundance and a >7-fold decrease in ping-pong Z_{10} -score in testis (Reviewers' Figure R5a). These data

are fully consistent with our finding that BmGtsf1 is specific for BmSiwi (revised Figure 3f), which predicts that BmAgo3 homotypic ping-pong should persist in the $\Delta BmGtsf1$ mutant.

Finally, our hypothesis—that Gtsf1 stabilizes a specific conformation of PIWI proteins is entirely consistent with the *Drosophila* literature. Our data suggest that without *Asterix/DmGtsf1*, Piwi is unable to attain the protein conformation required to recruit the downstream factors required for transcriptional silencing. Of course, our data do not directly test this, nor, we believe, should we be required to test this in this manuscript. Fly Piwi is a recent, highly derived, adaptation of Brachycera, a suborder of Diptera (Lewis et al., *Genome Biol Evol* 2016). BmPiwi, MILI, and MIWI are all homologs of fly Aub, and are thus more representative of the ancestral pathway that evolved into the modern piRNA pathways in mammals and most arthropods. Flies are a useful model, but many features of their piRNA pathways are unique and not found in other animals, including other non-Drosophilid Diptera.

Mouse Gtsf1 has been reported to localize to granules in both, nuclear and cytoplasmic compartments (Yoshimura 2018). While MIWI-1 piRNAs are reduced in the mutant, MILI-piRNAs, which are also ping-pong generated, are not. If GTSF1 stimulation was important for MILI-piRISC activity, one would expect a reduction of all ping-pong piRNAs. The discrepancy must be addressed.

We respectfully disagree. Yoshimura et al (*EMBO Reports*, 2018) clearly show that loss of GTSF1 in mouse testis disrupts the ping-pong pathway. **They reported that the g10A secondary piRNAs generated by MILI-MILI ping-pong are lost in *Gtsf1*^{-/-} mutants.** We reanalyzed the high-throughput sequencing data from Yoshimura et al. Consistent with what Yoshimura et al. originally described, MILI:MILI ping-pong—as measured by the abundance of ping-pong pairs and by the Z_{10} score (a measure of significance relative to background)—is completely lost in *Gtsf1*^{-/-} mutant testes. In *Gtsf1*^{+/-} heterozygous pre-natal mouse testes, 12.7% of piRNAs are in ping-pong pairs ($Z_{10} = 9.1$; i.e., $p < 0.00001$), whereas in *Gtsf1*^{-/-} homozygotes there is no significant ping-pong ($Z_{10} = -0.2$; i.e., $p > 0.8$) (Reviewers' Figure R5b).

GTSF1 has been shown to interact with tRNAs. It is suggested to prevent transposition mutagenesis (Ipsaro and Joshua-Tor, 2021). How do the authors integrate the bound tRNA in their model?

Ipsaro et al. (*Cell Reports* 2021) reported that when GTSF1 was expressed in Sf9 insect cells or p19 teratoma cells, it co-purifies with tRNA. Although tRNA interaction provided a convenient way for Ipsaro and Joshua-Tor to map the RNA-binding domains of GTSF1, there is currently no evidence that the GTSF1-tRNA interaction plays a role in transposon silencing in animals. In contrast, we

detected no RNA associated with our purified rGTSF1, which was produced in bacteria and purified using affinity and ion exchange chromatography.

Overall, the proposed (in vitro) model does not fit with prior observations in vivo. These discrepancies are not sufficiently addressed.

We respectfully disagree, as we have noted above and in Reviewers' Table R1. Our model explains all prior observations in mammals and lepidoptera, but does not fully address GTSF1 function in *Drosophila*. As we noted above, the *Drosophila* piRNA pathway diverges considerably from the ancestral pathway in arthropods (Lewis et al., *Nature Ecol Evol* 2018), and Piwi itself is found only in Brachycera. Piwi is not even present in the Nematocera suborder of Diptera, which includes mosquitos (Lewis et al., *Genome Biol Evol.* 2016). Mouse MILI and MIWI are the equivalent of fly Aub, not Piwi. Unfortunately, Aub homologs are generally named "Piwi," causing considerable confusion in the field.

Referee #3

This work clearly advances the piRNA-guided PIWI in vitro cleavage system, and sheds light on a potential physiological role for GTSF1 as a PIWI-slicing auxiliary protein in vivo.

Thank you!

1. *GTSF1 is known to have a nuclear function not associated with PIWI slicer activity in both flies and mice (ref. 7, 9). Drosophila Gtsf1 mainly colocalizes with nuclear Piwi, but not with cytoplasmic Aub or Ago3. In mice, GTSF1 colocalizes predominately with MIWI2, to a lesser extent with cytoplasmic MILI. Both Drosophila Piwi and mouse MIWI2 direct transcriptional silencing but lack slicer activity. How does GTSF1 facilitate PIWI nuclear function as a binding partner?*

We agree that nuclear PIWI function in mouse fetal germ cells and in fly nurse and follicle cells does not involve target slicing (please also see our response to the last comment of Reviewer 2, above). **In mouse fetal germ cells, GTSF1 is required for MILI-MILI ping-pong, a process that requires slicing; ping-pong is completely lost in *Gtsf1*^{-/-} fetal testis (Reviewers' Figure R5b).** Without ping-pong, MIWI2 cannot acquire piRNA guides and thus cannot enter the nucleus (Yoshimura et al., *EMBO Reports* 2018).

In flies, the requirement for Asterix/DmGtsf1 likely reflects a role for GTSF1 in stabilizing a specific conformation of the PIWI protein required for downstream steps in transcriptional silencing, such as binding other proteins. Testing this idea will not be simple, and we do not currently have a way to produce fly Piwi. Moreover, no one has recapitulated transcriptional silencing in a biochemical system. Our data speak to how GTSF1 functions in mammals and in *Bombyx mori*, animals whose piRNA pathways are more closely related to the

ancestral piRNA pathway (Lewis et al., *Genome Biol Evol* 2016; Anzelon et al., *Nature* 2021).

The authors propose GTSF1 as an auxiliary protein to enhance PIWI catalytic activities, how can this function be proven in a physiological setting in germ cells?

To provide an additional test that GTSF1 is the MIWI-stimulatory activity in vivo, we generated epitope-tagged GTSF1-expressing mice using CRISPR/Cas9 to insert a FLAG tag into the endogenous locus. Homozygous knock-in mice are male fertile, demonstrating that the tagged protein functions normally in vivo. Because the stimulatory activity was greatest in secondary spermatocytes, we prepared lysate from FACS-purified germ cells isolated from the homozygous knock-in mice and, as a control, from C57BL/6 mice. We then used anti-FLAG antibody to immunodeplete the lysate (Reviewers' Figure R4a). As a control, we performed the same immunodepletion using lysate from C57BL/6 mice, in which GTSF1 is not tagged.

For both the knock-in strain and the untagged control, lysate prepared from secondary spermatocytes stimulated piRNA-directed target cleavage by MIWI. **In contrast, the stimulatory activity was dramatically reduced in the Gtsf1-depleted lysate prepared from the tag-expressing mice (Reviewers' Figure R4b).** Immunodepletion of the untagged, C57BL/6 control had no effect on lysate activity. Moreover, the stimulatory activity was eluted with FLAG peptide from the anti-FLAG beads incubated with germ cell lysate from the tag-expressing mice, but not the control mice. We conclude that the stimulatory activity present in the secondary spermatocytes corresponds to GTSF1. We have added these new data to the manuscript as Figure 2d–f, and discuss them on page 9 of the revised text.

2. The biochemical basis for GTSF1 to facilitate PIWI slicing remains unclear. How GTSF1 interacts with PIWI proteins and binds RNA was poorly demonstrated in the manuscript. The authors mainly cited data from others' published work. On page 11, "GTSF1 does not detectably co-immunoprecipitate with MIWI piRISC from mouse testis (ref. 5, 77)". This seemingly contradicts with a role for GTSF1 as a MIWI slicing auxiliary factor.

It is not surprising that a tiny protein like GTSF1 was not previously co-immunoprecipitated with MIWI when using mass spectrometry to detect MIWI interactors. Although GTSF1 does not co-immunoprecipitate with MIWI, **MIWI and MILI (both large proteins) were shown by mass spectrometry to co-immunoprecipitate with GTSF1 (Yoshimura et al., 2018).** This is consistent with Dönertas et al. (*Genes Dev* 2013), who showed that in flies, only a very small fraction of Piwi is bound to GTSF1. Our work is consistent with these published findings.

It would be important to comprehensively map the interaction of GTSF1 with MIWI, MILI, and MIWI2 in the testis and by ectopic expression in cell lines using co-IP and co-staining. To experimentally show how GTSF1 uses different domains/motifs to bind RNA and interact with PIWI proteins will greatly improve this manuscript.

Respectfully, the reviewer is asking us to repeat published experiments. Others have mapped the binding interface of Asterix/GTSF1 for PIWI proteins and shown that only Zn-finger 1 is required for RNA binding. GTSF1 is a tiny protein. Its domains have been mapped, and distinct point mutations were shown to impair RNA-binding or interaction with PIWI proteins. At this point, a more detailed structural study will require a three-dimensional structure of the quaternary complex of PIWI protein, GTSF1, piRNA, and target. **Surely those experiments are beyond the scope of our manuscript, which reports (1) the discovery of the first protein that potentiates the cleavage activity of any Argonaute protein and (2) the function of GTSF1 in the piRNA pathway.**

3. One crucial result leading to the unbiased identification of MIWI potentiating factor was not shown. In Fig. 2b, have the authors performed mass spectrometry on the ~17 peak? Is GTSF1 in the fraction? What else is in the fraction? This is important to know.

No, we did not use mass spectrometry. We matched the properties of the activity to those of known genetic mutants. What we knew was:

- the protein was ~17 kDa
- required one or more structural zinc ions
- required reduced cysteines for its activity
- was highly expressed in meiotic spermatocytes compared to mitotic spermatogonia.

Among known genetic mutations, only GTSF1 met all of these criteria, so we immediately tested the recombinant protein. GTSF1 is ~19 kDa; the peak of activity on size-exclusion chromatography is ~17 kDa. It is hard to understand how anything else relevant to the activity could be in the fraction, because (1) it too would have to fortuitously be ~17 kDa; (2) recombinant GTSF1 potentiates the activity of MIWI by two orders of magnitude; and (3) immunodepletion of epitope-tagged GTSF1 from secondary spermatocyte lysate removes the potentiating activity.

We would like to note that it is extremely difficult to identify very small proteins such as GTSF1 by mass spectrometry. Published experiments have used mass spectrometry to identify Piwi proteins as GTSF1 interactors, but the reverse has only been successful by searching for GTSF1 peptides, rather than looking for proteins with most enriched peptides. Based on the Reviewers' comments, we performed mass spectrometry on our original gel filtration fractions. As expected, we obtained peptides corresponding to GTSF1, but we

would not have been able to identify GTSF1 with sufficient statistical rigor to conclude by mass spectrometry that it was the MIWI potentiating factor.

1. *Sequences of synthetic piRNAs and corresponding targets used in this study were not provided. A table summarizing these sequences and their use in specific figures is needed.*

We apologize for the inconvenience. The sequences were in Supplementary Data Table 2, which we neglected to upload when we submitted the manuscript.

How are the piRNA sequences selected? Do they have natural targets in germ cells? How many different piRNAs have been tested in cleavage assays and has GTSF promoted slicing in all cases?

For MIWI and MILI, we used two different guide sequences, none of which corresponded to endogenous piRNA sequences, highlighting the programmability of our in vitro system. For BmSiwi, we used one synthetic piRNA sequence and two endogenous piRNAs. We generally use these guide sequences for different Argonaute proteins, because they are free from stable secondary structures in either the or the target. GTSF1 promoted slicing in all cases tested.

2. *It will be helpful to show the domain architecture of GTSF1L and GTSF2. How do these proteins interact with MIWI or MILI? Do they have a similar PIWI interacting motif like GTSF1?*

We have added this information to Extended Data Figure 4 (see also Reviewers' Figure R6). None of the tryptophan residues previously shown to interact with PIWI proteins (Yoshimura et al., *EMBO Reports* 2018) are conserved in GTSF1L and GTSF2. A tryptophan residue farther downstream is conserved in all three paralogs, and there are other conserved residues that are likely involved in interacting with PIWI proteins. The amino acid sequences of MIWI and MILI are only about 30% identical, so it will be fascinating to study this in more detail, but such experiments are outside the scope of this manuscript.

3. *Have the authors tested other recombinant piRNA biogenesis factors (e.g. MVH) in the target cleavage assay to show GTSF1 is specific in potentiating slicing?*

Yes. Purified recombinant MVH in the presence of ATP and an ATP regenerating system had no effect on either the pre-steady-state or steady-state rate of target cleavage by MIWI alone or in the presence of GTSF1.

4. *Page 8, What is the structural basis for 3 Ws in GTSF1 to interact with PIWI?*

Given there is no three-dimensional structure of GTSF1 bound to a PIWI protein, the structural role of these residues is beyond the scope of our study.

Have the authors validated mutation in these residues ablate binding to MIWI?

Yes. In our original manuscript, we measured the K_D for wild-type GTSF1 and the W98A,W107A,W112A GTSF1 mutant, finding that the triple W mutation reduces the affinity of GTSF1 for MIWI by >60-fold (Figure 3f).

5. Page 8, Can GTSF1 bind target RNA in the cleavage assay? Need to show the GTSF1 RRKK mutant abolishes RNA binding but does not affect its binding to PIWI.

We do not know whether GTSF1 binds the piRNA or target or both. We are trying to develop an assay to answer this question. Currently, our assay for binding to PIWI in requires that GTSF1 retain some cleavage activity (e.g., Figure 3f). Others have shown that only the C-terminal domain is required for the interaction with PIWI proteins.

6. Page 8, “mutating these residues-W98A, W107A, and W112A in mice”, please clarify if this is mutating the mouse protein or generating mutant mice?

We apologize for the confusion. The intent of the sentence was to convey that the specific mouse tryptophan residues were W98, W107, and W112, not that the experiments were performed in vivo. We have revised the sentence to read, “Mutating these residues in the recombinant protein—W98A, W107A, and W112A for mouse GTSF1...”

Reviewers' Table R1

Reference	Animal (in vitro or in vivo)	Observation	Our Data
Ipsaro et al. (2021)	Recombinant mouse GTSF1 in cultured Sf9 or P19 cells; Asterix in Drosophila cultured OSS somatic cells.	When produced in cultured somatic cell lines, GTSF1 binds tRNAs.	No requirement for tRNA observed for GTSF1 to activate MIWI or MILI catalysis
		Point mutations in ZnF 1 impair tRNA binding. ZnF 2 does not bind tRNA.	These same mutations prevent GTSF1 from potentiating piRNA-directed target cleavage by MIWI or MILI piRISC.
Donertas et al. (2013)	Drosophila (in vivo) and in OSC cells** (ex vivo model for somatic follicle cells)	Asterix/DmGTSF1 required for transposon silencing in somatic follicle cells, female germline, and in OSC cells.	piRNA function in germline requires piRNA-directed PIWI-protein catalyzed target slicing.
		ZnF domains required for transposon silencing	Gtsf1 potentiation of PIWI protein catalysis requires ZnF domains
		Piwi and Asterix/DmGTSF1 co-IP from nuclear OSC lysate	Gtsf1 interacts directly with MIWI through its C-terminal domain
		Asterix/GTSF1 C-terminal domain is required for transposon silencing	
		C-terminal domain and C-terminal peptides interact with Piwi; identified W mutants	

*OSC is an immortalized, cultured cell line that provides an ex vivo model for somatic follicle cells. OSC cells and follicle cells express only Piwi. piRNAs are produced by the PIWI-protein-independent, phased piRNA biogenesis pathway; the PIWI-protein-dependent ping-pong pathway does not operate in OSC cells and transposon silencing is transcriptional.

Reviewers' Table R1

		Anti-sense germline-specific piRNA abundance is reduced in Asterix/DmGtsf1 mutant ovaries	Production of both Aub- and Piwi-bound antisense piRNAs in the female germline requires piRNA-directed PIWI-protein catalyzed target slicing.
Ohtani et al. (2013)	Drosophila OSC cells	Depletion of Asterix/DmGTSF1 by RNAi leads to loss of transposon silencing in OSC cells. Female flies mutant for Asterix/DmGTSF1 are sterile and have tiny ovaries.	
		Depletion of Asterix/DmGTSF1 does not alter the abundance of two individual piRNA species. [N.B.: the authors examined the abundance of these two piRNAs by qualitative Northern blotting. No piRNA sequencing was performed.]	
		Asterix/DmGTSF1 is a nuclear protein that interacts with Piwi. The interaction is resistant to RNase, but the data do not distinguish between a direct interaction and one that requires additional protein factors.	
		The zinc fingers are required for transposon silencing	

Reviewers' Table R1

Yoshimura et al. (2018)	Mouse testis (in vivo)	MILI-MILI ping-pong signature lost	GTSF1 potentiates the MILI endonuclease activity
		Target cleavage by MILI is impaired in the absence of GTSF1	
Takemoto et al. (2016)	Mouse testis (in vivo)	Gtsf1; Gtsf2 double knockout has no phenotype	GTSF1 can stimulate both MILI and MIWI endonuclease activity, whereas GTSFL1 and GTSF2 only efficiently stimulate MIWI.
Chen et al. (2020)	Bombyx mori (in vivo) and BmN cells in culture	BmGtsf1 mutant females show partial sex reversal, piRNA biogenesis defects, and transposon derepression.	BmGtsf1 stimulates target cleavage by BmSiwi. (piRNA biogenesis in B. mori is highly dependent on ping-pong.)
		BmGtsf1 is required for piRNA biogenesis, especially for the sex-determining piRNAs, Fem and Masc .	piRNA biogenesis and target regulation requires piRNA-directed, PIWI protein-catalyzed RNA cleavage, which is enhanced by BmGtsf1
		BmGtsf1 is required for piRNA effector function	
		BmGtsf1 interacts with BmSiwi	BmGtsf1 is specific for BmSiwi

$$k_{pot} = 1.2 \pm 0.04 \text{ min}^{-1}$$

$$K_D = 6 \pm 1 \text{ nM}$$

$$\chi^2 = 0.023$$

$$k_{pot} = 0.82 \pm 0.07 \text{ min}^{-1}$$

$$K_D = 600 \pm 100 \text{ nM}$$

$$\chi^2 = 0.021$$

Anzelon et al.

Our Manuscript

a

b

a

Bombyx mori testis

b

Mus musculus fetal testis

Reviewer Reports on the First Revision:

Referees' comments:

Referee #1 (Remarks to the Author):

Summary

The authors adequately answered the requests from this referee. There are a few concerns, though.

Major concerns:

(Page 11) It is risky to discuss the impact of BmGtsf1-like on MIWI and MILI based on such small differences in their target cleavage in Extended Data Figure 5b. This referee also has the same concern in Fig. 3f. If it is challenging to optimize the assay condition further, the authors may want to explain the reason, etc.

(Fig. 4c) Is the rate of steady-state cleavage (k_{ss}) of MIWI loaded with the 16 nt guide, i.e., 0.03 nM/min, correct? The steady-state slope in the 21 nt graph is steeper than that in the 16 nt.

Other comments:

(Page 9) W98A, W107A, and W112A seem to be located in the middle region rather than in the carboxy-terminus, according to Fig. 2c.

(Page 9) The K_d values of the GTSF1 wild-type and mutant, 8 nM and 500 nM, are different from those in Fig. 3e.

(Page 10) Does ~900 Mya mean 900 million years ago?

(Fig. 4b) It is hard to distinguish the colors in Fig. 4b. They need to be changed to noticeably different colors.

Referee #2 (Remarks to the Author):

My initial concerns remain. The presented work is limited to an in vitro observation that does not integrate well with the current literature. The molecular mechanism remains unknown, and thus the physiological relevance cannot be tested at this stage. Further studies are required to elucidate the molecular mechanisms and formulate a testable hypothesis to probe the biological relevance in vivo. This project is premature for consideration in Nature.

Referee #2

Here, the authors assay for piRISC-guided cleavage of a target RNA in vitro. They observe that the addition of testis extract to immunopurified MIWI-piRISC increases target cleavage activity (Fig. 1). They proceed to identify the activator by classical biochemical fractionation of the testis extract, and pinpoint some of the activity to a low molecular weight fraction (Fig. 2). The authors subsequently hypothesize that the activity might be dependent on zinc (EDF 3). Due to some indication for an activating function of zinc and the rough agreement of the low molecular weight of the activator, the authors decide to focus on a candidate that has previously been implicated in piRNA silencing in mice and flies (Yoshimura and Miyazaki, 2018; Donertas and Brennecke, 2013; Ohtani and Saito, 2013; Muerdter and Hannon, 2013). Gtsf1 is a small zinc-finger protein that interacts with PIWI proteins and preferentially binds tRNAs (Ipsaro et al., 2021).

The authors show that the addition of Gtsf1 increases MIWI- and MILI- target- cleavage activity in their in vitro assay (Fig 3). The effect is dependent on the ability of Gtsf1 to bind RNA. However, the results about a potential interaction between PIWI and Gtsf1 in this context are inconclusive: Fig. 3 suggests that the binding mutant reduces activation;

The Reviewer mischaracterizes our result; at saturating concentrations of GTSF1. The rate of cleavage is indistinguishable from piRISC alone. We cannot detect potentiation activity in the RNA-binding mutant using any available assay.

Yes, the RNA binding mutant is indistinguishable from a control assay in vitro. Thus it is crucial to identify what the RNA binding domain interacts with in this assay, and then test the formulated hypothesis in vivo. Does it interact with the substrate or the small RNA? Does it really increase cleavage activity, or does it present the substrate, or stabilize the detected product? The mechanism remains unclear.

however, the stimulatory activity from testis extract does not associate with MIWI (EDF 2).

Again, this is not what our data showed. Neither pre-incubating MIWI nor incubating piRISC with testis lysate increased piRISC activity. These experiments only show that under the specific conditions of our protocol (extensive washing while MIWI is tethered to magnetic beads), the potentiating activity does not co-purify with MIWI. They do not speak to whether the activity associates with MIWI under more permissive conditions or in the presence of target RNA.

It remains unclear how GTSF1 stimulates piRISC activity. It does not seem to directly interact with MIWI (potential association under more permissive conditions is really just handwaving).

The authors suggest there is a requirement for GTSF1 as a potentiating factor in piRNA-guided target cleavage in vitro. Nevertheless, the molecular mechanism and the

biological relevance of this finding remains unknown. The presented data seem preliminary and fail to explain previous genetic and structure-function data.

We respectfully disagree. Our detailed kinetic data, use of point mutants, and multiple GTSF1 orthologs explain nearly all previous genetic and structure- function data. **Reviewers' Table R1 summarizes the published genetic and molecular observations for GTSF1 in flies, silk moths, and mice and note how our data help explain them.** Remarkably, GTSF1 orthologs in mice and silk moth can distinguish between PIWI-protein paralogs. We note that GTSF1 is the only example of a protein that alters the catalytic activity of an Argonaute protein in any eukaryote or prokaryote.

The effect on the catalytic activity is only shown in *in vitro* assays. Without further studies *in vivo* or at least in cellulo (single molecule imaging techniques might be able to address this in lepidopteran cell culture models like BmN4). The alteration of piRISC activity *in vivo* (‘*in eukaryotes or procaryotes*’) remains speculative.

Experimental system: Addressing enzyme kinetics in vitro requires a stable and homogenous experimental system. Such a system optimally consists of highly purified soluble particles.

Over the past 22 years, our lab developed the *in vitro* systems used to study the kinetics of fly (Tuschl et al., *Genes Dev* 1999; Zamore et al., *Cell* 2000), plant mutant has **no detectable activity** to potentiate MIWI-catalyzed target cleavage (Tang et al., *Genes Dev* 2003), and mammalian (Schwarz et al. *Mol Cell* 2002; Hutvagner et al., *Science* 2002) Argonaute proteins. Before 2012 (Wee et al., *Cell* 2012), all kinetic experiments studying siRNAs and miRNAs were performed in fly embryo lysate, wheat germ extract, or HeLa cell S100. **No system allowing the kinetic study of any mammalian or insect PIWI protein was available before our current manuscript.** The work we present is the first time MIWI or MILI has been purified to homogeneity and loaded with a synthetic piRNA guide of defined sequence. This advance allowed us to perform detailed pre-steady-state and steady-state kinetic experiments. **Moreover, our system does in fact comprise highly purified soluble particles, as we explain below.**

I respectfully disagree. Ian MacRae's group recently established a robust model for piRNA targeting and piRISC function, and addressed mechanisms with elegant structure function studies (Nature 2021, PMID: 34471284).

In contrast, the presented system consists of a simple 2mune-purification of ectopically expressed FLAG-MIWI followed by loading with a synthetic oligo in vitro.

No purified fly or mammalian PIWI protein has been loaded with a small RNA of defined sequence prior to our work. Getting this to work was not a question of a simple immunopurification, but rather the culmination of four years of work. Expressing MIWI at sufficiently high levels to produce unloaded protein required multiple rounds of lentiviral transfection, each time purifying those cells with the highest level of

expression. Loading required development of novel methods and optimization. If this had been simple, we or others would have reported it years ago.

I do not doubt that this was a lot of work. However, Ian MacRae's system is much more robust and better characterized, and thus more suited to address kinetic questions, complement them with structure and elucidate a molecular mechanism (Nature 2021, PMID: 34471284).

Loading of Argonaute proteins with a small RNA is difficult and inefficient in vitro.

We respectfully disagree. We have shown that loading fly, plant, mouse, human, and bacterial Argonaute proteins with a small RNA (or DNA) is highly efficient and straightforward (Zamore et al, *Cell* 2000; Nykanen et al., *Cell* 2001; Schwarz et al., *Mol Cell* 2002; Schwarz et al., *Cell* 2003; Tang et al., *Genes Dev* 2003; Tomari et al., *Science* 2004; Haley et al., *NSMB* 2004; Tomari et al., *Cell* 2004; Schwarz et al., *Curr Biol* 2004; Matranga et al., *Cell* 2004; Schwarz et al., *PLoS Genet* 2006; Forstemann et al., *Cell* 2007; Tomari et al., *Cell* 2007; Wee et al., *Cell* 2012; Flores-Jasso et al., *RNA* 2013; Salomon et al., *Cell* 2015; Smith et al., *Nat Comm* 2019; Becker et al., *Mol Cell* 2019; Jolly et al., *Cell* 2020).

This seems in slight contrast with the above-mentioned quest. Regardless of how difficult we judge the assay (It might be easy for someone as experienced as the author), this discussion distracts from the lack of data for the molecular mechanisms or the biological relevance of the observed phenomenon.

The FLAG-eluate is expected to contain unloaded, misloaded and some correctly loaded MIWI.

In the experiments in our manuscript, the FLAG-MIWI was purified to apparent homogeneity. **The protein is highly purified and soluble, since the lysate was first centrifuged at 100,000 × g for 30 min, as indicated in Figure 1a (“S100”).** Extended Data Figure 1a and Supplementary Data Figure 1a showed that our purified MIWI corresponds to a single band on a Coomassie-stained gel. These data are also shown in Reviewers' Figure R3.

See above

It is not necessary to have a preparation of MIWI in which all the protein is loaded with the same guide, because **the concentration of active MIWI loaded with the synthetic guide was precisely determined using pre-steady-state kinetics, a method we have used in multiple papers beginning in 2012.** We note that this same approach was used for *Thermus thermophilus* Argonaute (TtAgo; Jolly et al., *Cell* 2021), which also cannot be purified by sequence-affinity chromatography.

See above

This heterogenous mixture poses a problem for the interpretation of the presented data, especially because any modulating activity may depend on direct interaction with the MIWI protein.

Our purified MIWI is not a heterogeneous mixture. It has been purified to apparent homogeneity and comprises only soluble protein. As we have done for nearly a decade (Wee et al., *Cell* 2012), we report only the concentration of *active* protein containing the synthetic guide. We used pre- steady-state —burstll measurements to quantify the active protein for every preparation of purified piRISC. Thus, for all experiments in which target cleavage is measured, the background of MIWI loaded with guides derived from RNA in HEK-293 cells does not influence the measurements: without loading of a synthetic guide, the purified MIWI does not cleave the target RNA (Extended Data Figure 1a). Moreover, all of our experiments using recombinant wild-type or mutant GTSF1 or the GTSF1 paralogs used saturating concentrations of the protein that were substantially greater than the *total* MIWI or MILI concentration.

Because we use the hyperbolic fit to rates rather than to protein concentrations, [MIWI] or [piRISC] do not affect our estimates of the K_D of the GTSF1:MIWI interaction.

See above

To address this technical problem, the authors previously established a purification procedure that uses antisense complementarity to enrich for the correctly loaded complex (Flores-Jasso and Zamore 2013). This purification scheme should be added to improve the homogeneity of the assayed complexes.

Needless to say, we spent a great deal of effort unsuccessfully trying to apply our previously described sequence-affinity purification method to MIWI. Not all Argonaute proteins can be purified by the method of Flores-Jasso and Zamore. For example, TtAgo cannot be purified by this method (Salomon et al., *Cell* 2015; Smith et al., *Nature Communications* 2019; Jolly et al., *Cell* 2020). Similarly, PIWI proteins do not elute when the capture oligo has sufficient complementarity to bind MIWI piRISC.

See above

(Why doesn't PIWI behave in this assay? Is this a technical problem or does it have biological meaning?)

Alternatively, the authors could establish a more elegant and robust in vitro system based on soluble recombinant proteins (Anzelon and MacRae bioRxiv 2020.12.07.413112).

Our in vitro system is based on soluble recombinant protein. We established a stable mammalian cell line by three sequential cycles of transduction with lentivirus and FACS sorting, establishing a line over-expressing tagged, unloaded MIWI. Tagged MIWI was purified from the supernatant of a 100,000 × g spin to ensure that only

soluble protein was used for subsequent immunoaffinity purification. We note that in both Anzelon et al. (*Nature* 2021) and our study, the PIWI protein was purified in a single step using an engineered N-terminal tag and corresponds to a single band on a gel (see Reviewers' Figure R3).

See above

(It would be interesting to complement this system with structural studies. Such studies might be able to identify)

While, we certainly agree that the work of Anzelon et al. (*Nature* 2021) is elegant, one cannot study the mammalian piRNA pathway using sponge PIWI. Moreover, in their original *bioRxiv* preprint, Anzelon et al. note that despite considerable efforts, they were not able to produce empty, loadable Piwi protein from any vertebrate, including mouse.

We previously used recombinant mouse AGO2—produced by over-expression in cultured cells and purified via an epitope tag—successfully in both ensemble and single-molecule studies (Wee et al., *Cell* 2012; Salomon et al., *Cell* 2015). In our published studies of fly Ago2, mouse AGO2, and bacterial TtAgo, we determined the percent active protein for every preparation using either burst kinetics or stoichiometric binding titrations. None of those proteins were 100% active. This was not an impediment to our detailed thermodynamic and kinetic experiments because we measured the percent active protein (as well as the percent active RNA in binding studies). These same analytical methods were used for every preparation of MIWI and MILI in this manuscript.

Nonetheless, the question as to whether GTSF1 can potentiate EfPiwi—the sponge protein studied by Anzelon et al.—is important, since it speaks to the evolutionary origins of GTSF1 as an activator of the Piwi endonuclease. Although the *Ephydatia fluviatilis* genome remains to be sequenced, a high quality genome assembly of its sister species, *Ephydatia muelleri*, is available. In our revised manuscript, we have now purified EmGtsf1 and performed target cleavage reactions using EfPiwi piRISC, which was loaded with a synthetic piRNA and purified by the methods described by Anzelon et al. (*Nature* 2021). Adding EmGtsf1 to EfPiwi increased the rate (k_{obs}) of target-cleavage by EfPiwi ~100-fold in physiological Mg^{2+} and >6-fold in supraphysiological Mn^{2+} (2 mM; revised Figure 3g and Extended Data Fig. 4c). Mn^{2+} , whose ionic radius is larger than that of Mg^{2+} , often enhances catalysis by Argonaute proteins. The larger effect of EmGtsf1 on EfPiwi in Mg^{2+} than Mn^{2+} is consistent with Gtsf1 increasing the time Piwi proteins spend in their catalytically active conformation.

Porifera (sponges) were the first phylum to branch off the evolutionary tree from the last common ancestor of all animals; all other animals are members of the clade Eumetazoa (Diploblasts). Our finding that EmGtsf1 stimulates target cleavage by EfPiwi is strong evidence that (1) the ancestral function of GTSF1 proteins was to stimulate the catalytic activity of Piwi proteins and (2) the ancestral Piwi protein was a sluggish enzyme. Together, our results demonstrate activation of a PIWI by GTSF1 can be reconstituted

from entirely recombinant components for proteins from mouse and from sponges, animals whose last common ancestor lived ~900 Mya.

The addition of EfPiwi experiments supports the in vitro findings, but the conclusion that ...Gtsf1 increasing the time Piwi proteins spend in their catalytically active conformation... remains a speculation without further mechanistic studies. The biological relevance remains unknown.

The datapoints in Fig. 3f are highly variable and the curve seems overfitted. Datapoints and standard deviations should be depicted as in 3d.

The data points in Figure 3f are *rates* for different concentrations of GTSF1 for three independent replicates. All of the 48 individual experiments (240 individual gel lanes) used to determine the rates were presented in Supplementary Data Figure 2. As described in the legend to Figure 3f, the curve was not fitted (and therefore was not overfitted), but rather corresponded to the mean K_D and k_{pot} of three independent experiments, each fitted independently (χ^2 for goodness of fit: 0.01, 0.02, 0.46 for the wild-type replicates; 0.02, 0.04, 0.02 for the GTSF1^{W98A,W107A,W112A} mutant). The data for all three replicates were shown in the figure, but the curve was not obtained by fitting to them. We sought to provide the most data-rich presentation possible, rather than provide only the mean and S.D. We apologize that this proved to be confusing rather than informative.

We have prepared a new figure (revised Figure 3f and Reviewers' Figure R2) using the more standard method of data presentation, showing rates as mean \pm S.D., fit to the equation,

$$k_{burst} = k_{pot} [GTSF1] / (K_D + [GTSF1]). \text{ (Equation R1)}$$

The data are now displayed on a \log_{10} x-axis for improved visualization. We have also added a second x-axis showing the molar ratio of GTSF1:piRISC. The goodness-of-fit χ^2 ranged from 0.021–0.023 ($p \sim 4 \times 10^{-6}$).

Finally, to test that we have fit the appropriate model to our data (i.e., to ensure that the data are not overfitted), we performed a lack-of-fit F -test. The F -test takes as the null hypothesis that the relationship assumed by the model is reasonable, i.e., there is no lack of fit. For our model (Equation R1), the test statistic is $F^* = 0.032$. We compared our F^* to an F -distribution with 5 numerator degrees of freedom and 14 denominator degrees of freedom. At a significance level of $\alpha = 0.05$, the test statistic F^* is less than the critical value $F(0.95, df_1 = 5, df_2 = 14) = 2.96$. Therefore we cannot reject the null hypothesis (Equation R1), indicating that the model appropriately fits the data.

Replicates should be shown in Fig 4b,c.

We have performed the additional replicates and added them to the figures. We would like to note that the experiments required to add replicates to Figure 4b alone required eight independent piRISC preps, 384 time points, and 12 gels.

The authors speculate on effects of PAZ-piRNA interactions (Fig 5c), but there are no experiments addressing this point.

We respectfully disagree. The published literature and our experiments do address this point. Computational (Jung et al., *J Am Chem Soc* 2013), biochemical (Hur et al., *J Biol Chem* 2013), and structural studies (Schirle et al. *Science* 2014; Schirle et al. *eLife* 2015) of Argonaute proteins and structural studies of lepidopteran (Matsumoto et al., *Cell* 2016) and sponge (Anzelon et al., *Nature* 2021) PIWI show that only three regions of the protein contact the guide: (1) the 5' phosphate and the g1 base reside in the MID domain; (2) the MID and PIWI domains display all or part of the seed sequence in a stacked helical conformation; (3) and the 3' end and the last two piRNA nucleotides reside in the PAZ domain. In our experiments, the identity of the scissile phosphate—the site of target cleavage—is unaltered, demonstrating that shortening the piRNA from 30 nt to 16 nt preserved the interaction of the protein with the piRNA 5' phosphate and seed sequence. The only interpretation of our results consistent with all known properties of Argonaute proteins is that the 3' end of the piRNA spends less time in the PAZ domain as the piRNA is shortened.

There are still no experiments addressing the PAZ-piRNA interaction in the experimental system. Yes, the speculation is justified. It should be presented as interpretation but not to be confused with an observation. A discussion about this detail should not distract from the lack of mechanistic insight and biological relevance.

These speculations should be tested using PAZ mutants.

Unfortunately, the experiment cannot be performed, since PAZ mutations in PIWI proteins prevent them from stably loading with piRNAs (Matsumoto et al., *Cell* 2016).

The model in EDF 8 is a theoretical depiction and not based on actual data. It contains variables that can't be assessed in the presented assays. The authors should clarify the purely speculative nature of this depiction.

We respectfully disagree. The model incorporates all available data from the field, including those in Anzelon et al. (2021) and our manuscript. Of course, by definition models are speculative; they are not intended solely to summarize what is known. A good model should provide testable predictions for future work. We would also like to note that the —two-state model for Argonaute function was first proposed by us 16 years ago (Tomari and Zamore, *Genes Dev* 2005). All structural and kinetic data published to date support the model.

The model could include alternative explanations to a direct impact of Gtsf1 on the catalytic activity of Piwi. Gtsf1 could impact the presentation of the substrate RNA,

induce crowding, or stabilize the measured products. A discussion about the details of this model and possible alternatives should not distract from the lack of mechanistic insight and biological relevance.

Requirement for zinc? These assays are not clear. (1) EDTA is expected to inhibit piRISC activity because Argonaute proteins are Mg-dependent nucleases. This argument cannot be used as line of evidence for Zn.

We respectfully disagree. The Mg²⁺ ions associated with Argonaute proteins are freely exchangeable with those in solvent (Schwarz et al., *Curr Biol* 2004; Yuan et al., *Mol Cell* 2005). Argonaute proteins, including MIWI, can be stripped of Mg²⁺ by EDTA, but are readily rescued when an excess of divalent cation (compared to EDTA) is added back, because the metal is not structural. Extended Data Figure 3 shows that excess Mg²⁺ rescues EDTA for the basal MIWI piRISC cleavage activity but not for testis-lysate-stimulated activity. These experiments demonstrate that (1) a divalent cation—distinct from the Mg²⁺ bound to MIWI—is required for testis-lysate to potentiate target cleavage and (2) excess divalent cation cannot restore potentiation activity, i.e., the metal bound to the protein is not in free equilibrium with divalent cation in solution. Zn-finger proteins are typically irreversibly denatured by chelating the zinc.

I am confused. This answer seems to agree that EDTA cannot be used as line of evidence for Zn.

(2) The NEM treatment seems to either degrade or precipitate the reaction so that neither the substrate nor the products are visible (EDF 3c).

Neither the substrate nor piRISC was exposed to unreacted NEM; all NEM was inactivated by adding excess DTT before adding target or piRISC. The mock and treated samples differ only in the order of addition of NEM and DTT to the testis lysate. The NEM treated lysate degrades the substrate at long time points, probably because a ribonuclease inhibitor protein such as RNH1 or the testis-specific RNH2 (Miyamoto & Hasuike *J Assist Reprod Genet* 2002) is inactivated. As much as 7% of the amino acids in ribonuclease inhibitors can be cysteine, and ribonuclease inhibitors are known to be inactivated by NEM (Dickson et al., *Prog Nucleic Acid Res Mol Biol* 2005). To avoid target degradation by nucleases in the NEM-treated testis lysate, we have repeated the experiment using a shorter time course (Reviewers' Figure R1). We have substituted these data for the original longer time course in Extended Data Figure 3c.

(3) Phenanthroline seems to result in significant RNA degradation, which makes the interpretation of EDF 3g difficult.

We agree that treatment with phenanthroline increases nuclease activity in the lysate. Nonetheless, our interpretation, that Zn is required for the activity led us to discover that GTSF1 is the potentiating factor. Our original and new data show that GTSF1 is both necessary and sufficient for this, and the data explain all published observations about *Gtsf1* mouse mutants.

The answer does not address the problem that EDF3g has significant RNA degradation and is not interpretable.

(4) The fact that both Ni and Zn columns retain the activity (EDF 3h), could have various explanations. Do they also deplete the activity from the flow though?

Yes: 50% of the piRISC-stimulating activity flows through the Ni-IMAC, whereas just 9% flows through the Zn-IMAC, i.e., Zn-IMAC retains 91% of the activity. We have added this information to Extended Data Figure 3h.

The concentration of zinc used in the in vitro assay exceeds physiological concentrations by >10 000 fold and is thus irrelevant.

The intracellular concentration of zinc is ~0.3 mM in a eukaryotic cell (Maret, *Metallomics* 2015). We used 12.5 mM Zn²⁺ in the presence of 10 mM EDTA. Thus, the free zinc concentration was 2.5 mM. That would be ~8-fold higher (and not 10⁴) than physiological conditions in the presence of EDTA and ~40-fold higher than physiological without EDTA.

Does this experiment have any effect on the structural Zn in GTSF?

Interestingly, high concentrations of Mg (12.5 mM ~ 6x physiological concentration) seem to stimulate the reaction most efficiently (EDF 3f). The authors should test if the initially observed activator is indeed only magnesium, which might exist at higher concentrations in germ granules (speculation).

The addition of supraphysiological magnesium generally stimulates all Argonaute proteins because the bound divalent ion is freely exchangeable with those in solvent. This is expected generally for metal-dependent nucleases, which typically have $K_{D,Mg}$ ~0.5–2 mM. ($K_{D,Mg}$ is the Mg²⁺ concentration at which the rate of reaction, $v = 1/2 v_{max}$.) v will continue to increase asymptotically well beyond 10 mM Mg²⁺. By contrast, under physiological magnesium concentrations, testis lysate or rGTSF1 stimulate MIWI activity by two orders of magnitude.

It is highly unlikely that magnesium concentrations are different between germ granules and cytosol, given that only the macroviscosity is thought to be higher in macromolecular condensates; ions freely diffuse in and out of such materials.

How would changes in viscosity impact kinetics? The concern remains that there is no evidence for physiological relevance of the in vitro observations.

The authors should also evaluate the effect of other divalent cations (Anzelon and MacRae bioRxiv 2020.12.07.413112).

In our manuscript, we had tested Mg²⁺ and Zn²⁺. We have now tested adding a physiological concentration (~0.1 mM) of Mn²⁺: the addition of 0.1 mM Mn²⁺ stimulated

target cleavage ~4-fold, far less than the two-log stimulation by GTSF1. As described above, EmGTSF1 stimulates target cleavage by EfPiwi, even in the presence of Mn^{2+} .

The RNA binding activity of GTSF1: The authors show that the RNA binding activity of GTSF1 is required for its function(Fig. 3). Which RNAs do GTSF1 proteins bind in this system?

We are currently developing assays to determine whether GTSF1 interacts with the piRNA guide or the target or both, but answering this question may require a cryo-EM structure of the quaternary complex of MIWI, GTSF1, piRNA, and target. We hope we can answer this important question in a future study.

This is a crucial question and should be answered for consideration in a high-profile journal. Simple Cross-linking experiments might be sufficient. There are only two different RNA species in this minimal system. Identifying why mutations in the RNA binding domain obliterate the effect would be a first step toward a molecular mechanism.

Is the function related to the GTSF-bound tRNAs that co-purify with the protein (Ipsaro and Joshua-Tor, 2021)?

A UV scan of our purified rGTSF1 showed that the protein is free from contaminating nucleic acids. Like us, Ipsaro et al. found that GTSF1-bound RNA is removed by ion exchange chromatography (Supplementary Figure 1A, Ipsaro et al., 2021), which is one of the steps in our rGTSF1 purification protocol. We were unable to detect any change in piRNA-directed, MIWI-catalyzed target cleavage in the presence of GTSF1 when tRNA was added.

I am confused. Addition of tRNA inhibits the stimulating effect? If GTSF1 is bound to tRNA in cellulo (Ipsaro et al.), would this prevent any stimulating effect in vivo? This is important to clarify to address the biological relevance of the presented observations.

Does Gtsf1 bind either the piRNA or the substrate in the in vitro assay?

We do not currently know, but are working hard to determine whether GTSF1 binds the piRNA, substrate, or, most likely, both. We hope to report the results of our studies in a future paper.

This is a crucial question and should be answered for consideration in a high-profile journal. Simple Cross-linking experiments might be sufficient. There are only two different RNA species in this minimal system (three is you count the products). Identifying why mutations in the RNA binding domain obliterate the effect are a first step to elucidating a molecular mechanism.

RNA binding has only clearly been shown for the first zinc finger. The authors should use RNA binding mutants for each zinc-fingers separately to test which one is required and which binds RNA.

Ipsaro and Joshua-Tor (*Cell Reports* 2021; their Figure 1B) demonstrated that the second zinc finger does not participate in RNA binding.

Protein-protein interactions: GTSF1 interacts with PIWI complexes via aromatic residues in its central region (Donertas and Brennecke 2013; Yoshimura and Miyazaki, 2018). Mutant GTSF1 seems impaired in stimulating cleavage activity in vitro (Fig. 3F). However, the activator from testis extracts does not associate with MIWI (EDF 2). This discrepancy in the presented data needs clarification.

Our experiments showed that under the specific conditions of our protocol (extensive washing while MIWI is tethered to magnetic beads), the potentiating activity does not co-purify with MIWI. They do not speak to whether the activity associates with MIWI under more permissive conditions or in the presence of target RNA. We do not see any discrepancy.

I believe that investigating these differences between the presented in vitro observations and in vivo data are important to address biological relevance.

Perhaps GTSF1 is not the same factor that is responsible for the stimulating activity of the mouse testis extract. To resolve this question, the authors should test extract from

GTSF1 knockout testis or testis extract depleted of GTSF1 and compare its stimulative function with wild type extract.

This is an excellent suggestion, and we agree with the reviewer that testing testis extract depleted of GTSF1 is a critical experiment that was missing from our manuscript. To address the Reviewer's question, we used CRISPR/Cas9 to insert a 3XFLAG tag into the endogenous locus. These epitope-tagged GTSF1 knock-in mice are male fertile, demonstrating that the tagged protein functions normally in vivo. Because the stimulatory activity is greatest in secondary spermatocytes, we prepared lysate from FACS-purified germ cells isolated from the homozygous knock-in mice and, as a control, from C57BL/6 mice. We then used anti-FLAG antibody to immunodeplete lysate from purified secondary spermatocytes isolated from the homozygous knock-in mice (Reviewers' Figure R4a). In parallel, we performed the same experiment using testis lysate from C57BL/6 mice, in which GTSF1 is not tagged.

For both the knock-in strain and the untagged control, lysate prepared from secondary spermatocytes stimulated piRNA-directed target cleavage by MIWI. **In contrast, the stimulatory activity was dramatically reduced in the immunodepleted extract prepared from secondary spermatocytes from the tag-expressing, but not the C57BL/6 control, mice** (Reviewers' Figure R4b). Moreover, the stimulatory activity was eluted with 3XFLAG peptide from the anti-FLAG beads incubated with germ cell lysate

prepared from the tag-expressing, but not the control, mice. We conclude that the stimulatory activity present in the secondary spermatocytes corresponds to GTSF1. We have added these new data to the manuscript as Figure 2d–f, and discuss them on page 9 of the text.

This experiment is nice and required to draw a link between the stimulation by testis extract and GTSF1 in the first place.

It would also be interesting to know where within PIWI complexes GTSF1 binds, whether the binding is direct, and if not, which cofactor mediates the interaction. The authors should test the interaction of different GTSF proteins with different PIWI proteins and check if interaction strength correlates with activation of cleavage activity (EDF 5)

These experiments were in our original manuscript. Figure 3f showed that rGTSF1 binds directly to MIWI (GTSF1 and MIWI are the only two proteins in the reaction). In our original manuscript, we measured the K_D for wild-type GTSF1 and the GTSF1^{W98A,W107A,W112A} mutant (Figure 3f): the triple W mutation reduced the affinity of GTSF1 for MIWI >60-fold and can account for the difference in stimulatory activity of the wild-type and mutant proteins. The K_D for wild-type GTSF1 binding MIWI piRISC ~7 nM; **the concentration of piRISC in meiotic spermatocytes is ~5 μM (Gainetdinov et al., Mol Cell 2018)**. No additional protein should be required for GTSF1 to bind piRISC in vivo.

I am still confused about a potential mechanism. Does GTSF1 directly interact with Piwi or via an RNA to stimulate RISC activity? The molecular mechanism should be addressed. The authors are in a great position to test the molecular mechanism and its biological relevance.

PiRISC is supposed to catalyze multiple rounds of cleavage in vivo.

To the best of our knowledge, no published work has tested whether piRISC is multiple-turnover in vivo, and no PIWI protein has ever been shown to catalyze multiple rounds of cleavage in vitro. Work from the Pillai and Siomi labs suggests that the DEAD-box proteins Vasa and DDX43 facilitate transfer of sliced precursors between ping-pong partners, not that Vasa promotes multiple turnover catalysis. We note that the proposed mechanism for ping-pong amplification of piRNAs does not require PIWI proteins to catalyze multiple rounds of cleavage. The finding that, in mice, MIWI and MILI are nearly as abundant as ribosomes (Gainetdinov et al., Mol Cell 2018) may be a hint that cleavage is not multiple turnover in vivo.

The Zamore initially addressed the question of turn-over at the example of human Argonaute proteins (Science 2002, PMID: 12154197). The detailed differences of siRNA(Ago)RISC and piRNA(Piwi)RISC have recently been addressed in Anzelon et al. (2021). The entire manuscript deals with an in vitro stimulating activity of piRISC. Turnover should be addressed.

Target cleavage takes place in specialized cytoplasmic granules and turn-over is likely stimulated by the RNA helicase VASA/DDX4 (Xiol and Pillai, 2014). Identifying regulatory mechanisms of piRISC activity requires more effort to reconstitute a relevant environment in vitro and/or complementary assays in vivo.

No published study has shown that any function of the piRNA pathway occurs in nuage or other granules. Many piRNA pathway proteins localize to nuage, but whether they function there is unknown. No study has attempted to quantify the distribution of piRNA pathway proteins among cellular compartments. This is a common problem in cell biology: it is far easier to detect highly localized macromolecules than those that are more diffusely distributed. In a classic study of this phenomenon, Bergsten and Gavis (*Development* 1999) showed that the highly localized pool of *nanos* mRNA in polar granules—once thought to represent most if not all of the *nanos* mRNA in a fly embryo—actually corresponds to just 4% of the total.

This emphasizes the need for testing the biological relevance of the observations.

Drosophila GTSF1 localizes to the nucleus and associates with the piRISC that induces transcriptional silencing without target cleavage (Donertas and Brennecke, 2013). There is no report of cytoplasmic GTSF or any involvement in ping-pong. This existing literature does not fit with the proposed model.

We strongly disagree.

First, Yoshimura et al. (*EMBO Reports* 2018) showed that ping-pong is lost in *Gtsf1*^{-/-} homozygous mutant mouse fetal testes. (We discuss this in detail below in response to **the Reviewer's next comment.**)

Second, Chen et al. (*PLoS Genetics* 2020) showed that BmGtsf1 is required for both piRNA biogenesis and BmSiwi effector functions in silk moth. Silk moths, like other lepidoptera, have no nuclear PIWI protein, and piRNA production depends strongly on the ping-pong pathway. We reanalyzed the raw piRNA sequencing data from Chen et al. When normalized to miRNAs, the *BmGtsf1* mutant shows a ~5-fold reduction in piRNA abundance and a >7-fold decrease in ping-pong **Z₁₀-score in testis (Reviewers' Figure R5a)**. These data are fully consistent with our finding that BmGtsf1 is specific for BmSiwi (revised Figure 3f), which predicts that BmAgo3 homotypic ping-pong should persist in the *ΔBmGtsf1* mutant.

Finally, our hypothesis—that Gtsf1 stabilizes a specific conformation of PIWI proteins is entirely consistent with the *Drosophila* literature. Our data suggest that without *Asterix/DmGtsf1*, Piwi is unable to attain the protein conformation required to recruit the downstream factors required for transcriptional silencing. Of course, our data do not directly test this, nor, we believe, should we be required to test this in this manuscript. Fly Piwi is a recent, highly derived, adaptation of Brachycera, a suborder of Diptera (Lewis et al., *Genome Biol Evol* 2016). BmPiwi, MILI, and MIWI are all homologs of fly

Aub, and are thus more representative of the ancestral pathway that evolved into the modern piRNA pathways in mammals and most arthropods. Flies are a useful model, but many features of their piRNA pathways are unique and not found in other animals, including other non-Drosophilid Diptera.

Fly genetics should not be neglected because it does not fit a model based on in vitro data with unknown biological relevance.

Mouse Gtsf1 has been reported to localize to granules in both, nuclear and cytoplasmic compartments (Yoshimura 2018). While MIWI-1 piRNAs are reduced in the mutant, MILI-piRNAs, which are also ping-pong generated, are not. If GTSF1 stimulation was important for MILI-piRISC activity, one would expect a reduction of all ping-pong piRNAs. The discrepancy must be addressed.

We respectfully disagree. Yoshimura et al (*EMBO Reports*, 2018) clearly show that loss of GTSF1 in mouse testis disrupts the ping-pong pathway. **They reported that the g10A secondary piRNAs generated by MILI-MILI ping-pong are lost in Gtsf1^{-/-} mutants.** We reanalyzed the high-throughput sequencing data from Yoshimura et al. Consistent with what Yoshimura et al. originally described, MILI:MILI ping-pong—as measured by the abundance of ping-pong pairs and by the Z_{10} score (a measure of significance relative to background)—is completely lost in *Gtsf1^{-/-}* mutant testes. In *Gtsf1^{+/-}* heterozygous pre-natal mouse testes, 12.7% of piRNAs are in ping-pong pairs ($Z_{10} = 9.1$; i.e., $p < 0.00001$), whereas in *Gtsf1^{-/-}* homozygotes there is no significant ping-pong ($Z_{10} = -0.2$; i.e., $p > 0.8$) (Reviewers' Figure R5b).

Previously the Zamore lab suggested that all piRNA biogenesis requires ping-pong induction (PMID 30193099). Shouldn't all Mili-piRNAs be lost?

GTSF1 has been shown to interact with tRNAs. It is suggested to prevent transposition mutagenesis (Ipsaro and Joshua-Tor, 2021). How do the authors integrate the bound tRNA in their model?

Ipsaro et al. (*Cell Reports* 2021) reported that when GTSF1 was expressed in Sf9 insect cells or p19 teratoma cells, it co-purifies with tRNA. Although tRNA interaction provided a convenient way for Ipsaro and Joshua-Tor to map the RNA-binding domains of GTSF1, there is currently no evidence that the GTSF1- tRNA interaction plays a role in transposon silencing in animals. In contrast, we detected no RNA associated with our purified rGTSF1, which was produced in bacteria and purified using affinity and ion exchange chromatography.

I agree that the physiological relevance of the finding that GTSF1 associates with

tRNAs (Ipsaro et al. 2021) is not clear. However, the physiological relevance of the presented data is not clear either. The biological relevance of the observation as well as the molecular mechanism should be addressed for publication in a high-profile journal.

Overall, the proposed (*in vitro*) model does not fit with prior observations *in vivo*. These discrepancies are not sufficiently addressed.

We respectfully disagree, as we have noted above and in Reviewers' Table R1. Our model explains all prior observations in mammals and lepidoptera, but does not fully address GTSF1 function in *Drosophila*. As we noted above, the *Drosophila* piRNA pathway diverges considerably from the ancestral pathway in arthropods (Lewis et al., *Nature Ecol Evol* 2018), and Piwi itself is found only in Brachycera. Piwi is not even present in the Nematocera suborder of Diptera, which includes mosquitos (Lewis et al., *Genome Biol Evol.* 2016). Mouse MILI and MIWI are the equivalent of fly Aub, not Piwi. Unfortunately, Aub homologs are generally named —Piwi, causing considerable confusion in the field.

Fly genetics should not be neglected because it does not fit a model based on *in vitro* data with unknown biological relevance.

The revised version of the manuscript, though improved, fails to address the molecular mechanism and the biological relevance of the observations. I believe that it is crucial to address the molecular mechanism and the biological relevance for a high-impact journal, and the Zamore lab is in a perfect position to conduct these studies. The current work might be suited for a more specialized journal

Referee #3 (Remarks to the Author):

The authors have satisfactorily addressed most of my questions. In particular, the new *ex vivo* assay using tagged GTSF1 KI mice to prove GTSF1's PIWI potentiating activity is an important advance. The addition of the sponge *Gtsf1* data further strengthens the ancestral conserved function of the GTSF family. Overall, this revised manuscript is much improved, supporting the conclusion that GTSF1 is a novel auxiliary factor that potentiates the catalytic activity of PIWI proteins. This work will be of great interest to the small RNA community.

Suggestions:

For MIWI or MILI *in vitro* target cleavage assay, it would be important to include an successful example of endogenous piRNA (e.g. transposon piRNA or genic piRNA) slicing natural RNA target potentiated by GTSF1. This will strengthen the confidence in the broad applicability of this assay to study physiological piRNAs.

It would be helpful to show GTSF1 protein localization in different germ cell types in the adult testis and its colocalization with MIWI or MILI. This will be a straightforward experiment using the new Flag/HA-GTSF1 KI mice.

Title: GTSF1 (19kD) is not considered "tiny", suggest to remove "tiny" from the title or change it to "small".

Author Rebuttals to First Revision:

Responses to the Critiques of Reviewers 1 and 2

Editors' Concerns

We invited Reviewers 1 and 3 to comment on the remaining concerns of Reviewer 2.

Reviewer 1 commented that the request for insight into mechanism sounds reasonable - at least to answer whether GTSF1 directly interacts with Piwi or via an RNA. The results of EDFig2 suggest that GTSF1 does not interact strongly with Piwi, which is confusing. On the other hand, Reviewer 3 felt that further testing would be difficult in the absence of structural data.

The results of Extended Data Figure 2 show that GTSF1 does not interact stably with MIWI in the *absence of a target RNA*. The experiments in Figure 3e show a $K_D \sim 6$ nM for GTSF1 binding MIWI piRISC in the *presence* of a target RNA.

These experiments make specific mechanistic predictions that can be tested in the future by our lab or others, alleviating the concerns of Reviewer 2 regarding testable hypotheses. We hope this clarifies the editors' concerns.

Referee #1 (Remarks to the Author):

Summary

The authors adequately answered the requests from this referee. There are a few concerns, though.

Major concerns:

(Page 11) It is risky to discuss the impact of BmGtsf1-like on MIWI and MILI based on such small differences in their target cleavage in Extended Data Figure 5b. This referee also has the same concern in Fig. 3f. If it is challenging to optimize the assay condition further, the authors may want to explain the reason, etc.

We reexamined these data, evaluating statistical significance using ANOVA with Dunnett's post-hoc test to prevent type I errors from multiple hypothesis testing.

The addition of BmGtsf1-like increased the amount of target cleaved by MIWI by 2.6-fold (S.D. = 0.1; $p = 0.0027$), a significant increase. Perhaps the effect size seems small because MmGTSF1 increased target cleavage by MIWI by 17-fold (S.D. = 1; $p = 0.0002$)? BmGtsf1-like had no stimulatory effect for MILI. (All of these data have been moved to Extended Data Figure 6b to accommodate the addition of the experiments using endogenous piRNA sequences to Figure 3.)

(Fig. 4c) Is the rate of steady-state cleavage (k_{ss}) of MIWI loaded with the 16 nt guide, i.e., 0.03 nM/min, correct? The steady-state slope in the 21 nt graph is steeper than that in the 16 nt.

The burst and steady-state phases for the 21 nt guide are readily separable because they differ so dramatically. In contrast, the reaction rate for the 16 nt guide is a mixture of two rates of comparable magnitude. Consequently, one cannot infer the steepness of the steady-state slope from the graph. The best fit of our data to the burst-and-steady-state equation gives $k_{burst} \sim 0.04 \text{ nM min}^{-1}$ and $k_{ss} \sim 0.03 \text{ nM min}^{-1}$, consistent with a step other than product release being rate-determining for the 16 nt guide.

Other comments:

(Page 9) W98A, W107A, and W112A seem to be located in the middle region rather than in the carboxy-terminus, according to Fig. 2c.

We have harmonized our nomenclature to match that of Yoshimura et al. (2018) and now refer to this region as the “central region.”

(Page 9) The K_d values of the GTSF1 wild-type and mutant, 8 nM and 500 nM, are different from those in Fig. 3e.

As requested by the Reviewers during the first round of review, we performed additional replicates and changed the method of data display for Figure 3e. Unfortunately, we failed to update the text. We have changed 8 nM in the text to the correct value, 6 ± 1 nM, and 500 nM to the correct value, 600 ± 100 nM. We apologize for our error and thank the Reviewer for pointing it out.

(Page 10) Does ~900 Mya mean 900 million years ago?

Yes.

(Fig. 4b) It is hard to distinguish the colors in Fig. 4b. They need to be changed to noticeably different colors.

We have changed the colors.

Referee #2 (Remarks to the Author):

My initial concerns remain. The presented work is limited to an in vitro observation that does not integrate well with the current literature. The molecular mechanism remains unknown, and thus the physiological relevance cannot be tested at this stage. Further studies are required to elucidate the molecular mechanisms and formulate a testable hypothesis to probe the biological relevance in vivo. This project is premature for consideration in Nature.

Typically, biochemical studies of mechanism precede in vivo analysis. In our case, extensive in vivo data, including mutation of key residues with clear physiological relevance, were already published. Because mutations known to affect GTSF1 function in vivo similarly influence the biochemical activity we measured in vitro, we respectfully disagree with the assertion that our data do not integrate with the literature. We also disagree with assertion that the molecular mechanism remains unknown. Our data demonstrate that GTSF1 proteins from

diverse animal species, including species separated by ~900 million years of evolution, can recognize piRISC and activate the catalytic step in the target cleavage reaction. As we noted previously, the only piRNA mechanism not fully explained by our data is transcriptional silencing by *Drosophila* Piwi. Fly Piwi evolved in the fly lineage and is not found among animals more generally. Our data is consistent with the idea that to function in transcriptional silencing, fly Piwi must be in a conformation similar to the catalytically active conformation of other PIWI proteins and that GTSF1 helps stabilize this conformation. Given that our study examined PIWI proteins from sponges, lepidoptera, and mammals—all of whose PIWI proteins are orthologous—explaining the mechanism of an evolutionary outlier, no matter how interesting, lies outside the scope of our work.

In response to your original requests we: (1) created a knock-in FLAG-GTSF1 mouse line in order to demonstrate GTSF1 is the same factor that is responsible for the stimulating activity of the mouse testis extract; and (2) expanded our study to include a collaboration with the MacRae lab in order to test GTSF1 activity in their purified EfPIWI system. Both major efforts resulted in data unequivocally supporting our original model. Short of a cryo-EM structure, we do not know any more experiments we could perform that would meet your definition of elucidating the molecular mechanism.

Referee #3 (Remarks to the Author):

The authors have satisfactorily addressed most of my questions. In particular, the new ex vivo assay using tagged GTSF1 KI mice to prove GTSF1's PIWI potentiating activity is an important advance. The addition of the sponge Gtsf1 data further strengthens the ancestral conserved function of the GTSF family. Overall, this revised manuscript is much improved, supporting the conclusion that GTSF1 is a novel auxiliary factor that

potentiates the catalytic activity of PIWI proteins. This work will be of great interest to the small RNA community.

Thank you!

Suggestions:

For MIWI or MILI in vitro target cleavage assay, it would be important to include an successful example of endogenous piRNA (e.g. transposon piRNA or genic piRNA) slicing natural RNA target potentiated by GTSF1. This will strengthen the confidence in the broad applicability of this assay to study physiological piRNAs.

This was a great suggestion, and we have now examined the effect of mouse GTSF1 on two abundant pachytene piRNAs: one from the *pi6* cluster targeting *Scsep1* mRNA (Wu et al., *Nature Genetics* 2020) and one from *pi9* antisense to the L1MC transposon RNA. For *Scsep1*, we used the corresponding target site cleaved by the piRNA in vivo; for *L1MC*, we used a fully complementary target. GTSF1 accelerated the burst rate of MIWI-catalyzed target cleavage 19-fold for *L1MC* and 60-fold for *Scsep1*. The results are shown in Fig. 3c and Extended Data Fig. 5.

It would be helpful to show GTSF1 protein localization in different germ cell types in the adult testis and its colocalization with MIWI or MILI. This will be a straightforward experiment using the new Flag/HA-GTSF1 KI mice.

Yoshimura et al. (*EMBO Reports* 2018) showed by immunofluorescence that in prospermatogonia, GTSF1 colocalizes with MILI, TDRD1, MIWI2, and TDRD9. Interestingly, they found that the abundance of GTSF1, just like Asterix in flies, appears to be considerably less than that of its PIWI protein partners.

Title: GTSF1 (19kD) is not considered “tiny”, suggest to remove “tiny” from the title or change it to “small”.

We have removed the word “tiny” from the title.

Responses to the Third Set of Comments from Referee #2

Reviewer 2 poses three types of objections to publication of our study.

First, the Reviewer claims repeatedly that we have not demonstrated the mechanism by which GTSF1 enhances target cleavage by PIWI proteins. But we have, in fact, demonstrated the mechanism by which GTSF1 acts: it increases the pre-steady-state rate of cleavage of PIWI proteins. PIWI proteins find their targets at the speed of diffusion and once bound, release of the target RNA takes many, many hours. Thus, we can exclude all mechanisms that propose that GTSF1 either speeds target finding (no protein can accelerate diffusion) or delays target release (i.e., increases the affinity for substrate). Unless GTSF1 acts by a mechanism heretofore unknown in enzymology, it must either lower an energy barrier to PIWI proteins' achieving a catalytically active state or change the rate of the chemical step of phosphodiester bond breaking. I know of no examples of a protein that does the latter.

Second, the Reviewer insists that we confirm our experiments in vivo using methods that do not currently exist. We have, in fact, used mutations in GTSF1 that have already been tested in vivo, thereby correlating our kinetic studies with the in vivo consequences for piRNA production for both loss of GTSF1 and point mutation of key residues in GTSF1. But the Reviewer asks that we go further, and develop assays to study catalysis itself in vivo. Such experiments have never been performed for any Argonaute protein because no one knows how to do them.

Third, the Reviewer insists that we provide a level of molecular detail that could only come from a cryo-EM structure. This is clearly an unreasonable demand.

Specific Responses to the Reviewer's Comments

R2: Yes, the RNA binding mutant is indistinguishable from a control assay in vitro. Thus it is crucial to identify what the RNA binding domain interacts with in this assay, and then test the formulated hypothesis in vivo.

My laboratory has studied RNA silencing and Argonaute proteins for more than two decades, and have personally studied RNA binding proteins for 36 years. I know of no reliable method for detecting interactions between an RNA-binding protein and an RNA in vivo in mouse testis as the Referee requests. While we are most definitely working to determine whether the GTSF1 RNA-binding domain binds the piRNA, the target RNA, or both, all of our ongoing experiments are in vitro using purified components. Most importantly, we are working hard to solve the cryo-EM structure of the quaternary complex of MIWI, piRNA, target RNA, and GTSF1. It is unreasonable to ask authors not only to identify a novel and unanticipated phenomenon whose discovery explains nearly all available in vivo genetic data about GTSF1, but also to determine the precise molecular interactions of GTSF1 with RNA.

R2: Does [GTSF1] interact with the substrate or the small RNA? Does it really increase cleavage activity, or does it present the substrate, or stabilize the detected product? The mechanism remains unclear.

The mechanism by which GTSF1 accelerates the pre-steady-state rate of catalysis by piRISC is decidedly not unclear. Our evidence clearly distinguishes among the hypotheses suggested by the Reviewer. Anzelon et al. (*Nature* 2022) determined k_{off} and k_{on} for a target bound to EfPiwi piRISC: $k_{\text{off}} < 10^{-3} \text{ min}^{-1}$ and $k_{\text{on}} \sim 3.3 \times 10^9 \text{ min}^{-1}$ (i.e., at the diffusion limit). It is not possible that GTSF1 “presents the substrate” since association of target with piRISC is diffusion-limited, i.e., cannot be made faster in three dimensions. Given that mean lifetime of target-bound piRISC is >16 hours ($\tau = 1/k_{\text{off}}$), a time scale longer than our assays, GTSF1 cannot enhance binding of substrate to piRISC. Similarly, GTSF1 cannot stabilize the detected product, since we can account for all the RNA in our reactions: the product + uncleaved target = total; no product “is missing,” so none could be stabilized.

GTSF1 increases the pre-steady-state (burst) rate but does not increase the steady-state rate. To the best of our knowledge, the only known mechanisms for increasing the pre-steady-state rate of a reaction—i.e., the rate at which the very first substrate to encounter the enzyme is converted to product—is by reducing the energy barrier to adopting a catalytically competent conformation or by directly participating in catalysis itself. Given the general inaccessibility of the active site, the hypothesis that is most reasonable and most consistent with the known properties of Argonaute proteins is that GTSF1 lowers the energy barrier to PIWI proteins adopting a catalytically competent conformation.

R2: The effect on the catalytic activity is only shown in in vitro assays. Without further studies in vivo or at least in cellulo (single molecule imaging techniques might be able to address this in lepidopteran cell culture models like BmN4). The alteration of piRISC activity in vivo (in eukaryotes or procaryotes') remains speculative.

ALL of the known properties of Argonaute proteins have been deduced using in vitro experiments. **Catalysis by Argonaute proteins has never been studied by “single molecule imaging techniques” in vivo or in cultured cells for the simple reason that such methods do not exist for endonucleases.**

R2: I respectfully disagree. Ian MacRae's group recently established a robust model for piRNA targeting and piRISC function, and addressed mechanisms with elegant structure function studies (Nature 2021, PMID: 34471284).

The Reviewer inexplicably trusts the MacRae data, but objects to our kinetic studies using methods relied on in multiple published publications from our lab and others. These papers include our most recent kinetic study (Ober-Reynolds et al., *Mol Cell*, in press) of TtAgo, a protein that like MIWI, cannot be sequence affinity purified.

Despite the Reviewer's continued mischaracterization of the purity of our piRISC preparation, **we added to our revised manuscript experiments using EfPiwi loaded with a piRNA. These experiments not only employ the exact piRISC used in the MacRae structure paper, but they were performed by the first author of the 2021 *Nature* paper.** The result: sponge GTSF1 stimulates the pre-steady-state rate of target cleavage catalyzed by EfPiwi. It seems that even when we perform the experiments requested by the Reviewer, we receive no credit.

R2: It would also be interesting to know where within PIWI complexes GTSF1 binds, whether the binding is direct, and if not, which cofactor mediates the interaction. The authors should test the interaction of different GTSF proteins with different PIWI proteins and check if interaction strength correlates with activation of cleavage activity (EDF 5)

We agree. But answering this question will require a cryo-EM structure of the quaternary complex of MIWI, piRNA, target RNA, and GTSF1.

R2: Fly genetics should not be neglected because it does not fit a model based on in vitro data with unknown biological relevance.

The Reviewer crosses the line between rigorous and abusive with this comment. Fly genetics has not been "neglected." As we have clearly stated multiple times, we are interested in studying the mechanism of target cleavage by the PIWI proteins that share a common evolutionary descent. What GTSF1 does for fly Piwi, a protein unique to a very tiny set of flies (it is absent from most Diptera and all other animals), is interesting, but not particularly relevant for our study, which addresses a range of PIWI proteins that spans 900 million years: the pre-steady-state rates of target cleavage by mouse, silk moth, and sponge PIWI proteins are all accelerated by GTSF1 proteins.

R2: Previously the Zamore lab suggested that all piRNA biogenesis requires ping-pong induction (PMID 30193099). Shouldn't all Mili-piRNAs be lost?

The Reviewer again misrepresents the literature. In the cited paper (Gainetdinov et al., *Mol Cell* 2018) we wrote:

"Although the revised model explains the production of the majority of piRNAs in most animals, how piRNA biogenesis initiates in fetal germ cells in animals, such as mammals, whose germline is induced rather than predetermined, is not known. Moreover, the somatic ovarian follicle cells of *D. melanogaster* produce phased piRNAs, yet the catalytic activity of Piwi, the only PIWI protein expressed in these cells, is dispensable (Darricarrère et al., 2013). The absence of ping-pong in fly somatic follicle cells is a striking evolutionary exception to the ubiquitous presence of piRNA

amplification in the somatic tissues of a majority of arthropods (Lewis et al., 2018).”

piRNA biogenesis does not require ping-pong, as the reviewer well knows. And multiple papers have established that in the absence of canonical mechanisms of piRNA biogenesis, other pathways can compensate. But ping-pong requires target cleavage. In the absence of GTSF1 in vivo, ping-pong is dramatically reduced, despite the presence of additional, potentially compensatory GTSF1 paralogs. Thus, the in vivo observations are not only consistent with our experimental results, **they are explained by our findings.**

Reviewer Reports on the Second Revision:

Referees' comments:

Referee #1 (Remarks to the Author):

This reviewer is satisfied with the responses to the requests. The figure numbers of Extended Data are often shifted. The authors should check each of them carefully.

Referee #3 (Remarks to the Author):

Fig. 3C and Extended Fig. 5 is a great addition to show physiological relevance.

Yoshimura et al. (EMBO Reports 2018) showed GTSF1 is also co-expressed with MIWI2 (lacking slicer activity) in the nuclei of E17.5 mouse prospermatogonia. This suggests that GTSF1 might have cleavage-independent nuclear function associated with MIWI2 (e.g. transcriptional silencing). Although this is a potential role beyond the scope of this study, the authors could make the readers be aware in the discussion that potentiating PIWI catalytic activity may not be the sole function of GTSF1.

Overall, the authors have satisfactorily addressed my questions. This study opens new avenues for mechanistic inquiries into the regulation of piRNA biogenesis and function in various species.

Author Rebuttals to Second Revision:

Arif et al., *Nature* Manuscript 2021-05-07169C

Responses to the Final Set of Comments from Referees #1 and #3

Referee #1 (Remarks to the Author):

This reviewer is satisfied with the responses to the requests. The figure numbers of Extended Data are often shifted. The authors should check each of them carefully.

We have fixed the errors in numbering the Extended Data figures; we apologize for the inconvenience.

Referee #3 (Remarks to the Author):

Fig. 3C and Extended Fig. 5 is a great addition to show physiological relevance.

Thank you!

Yoshimura et al. (EMBO Reports 2018) showed GTSF1 is also co-expressed with MIWI2 (lacking slicer activity) in the nuclei of E17.5 mouse prospermatogonia. This suggests that GTSF1 might have cleavage-independent nuclear function associated with MIWI2 (e.g. transcriptional silencing). Although this is a potential role beyond the scope of this study, the authors could make the readers be aware in the discussion that potentiating PIWI catalytic activity may not be the sole function of GTSF1.

This is a great suggestion. We now note this in the discussion section.

Overall, the authors have satisfactorily addressed my questions. This study opens new avenues for mechanistic inquiries into the regulation of piRNA biogenesis and function in various species.

Thank you!